# LIKELIHOOD TRAINING OF SCHRÖDINGER BRIDGE USING FORWARD-BACKWARD SDES THEORY

**Tianrong Chen**[*], **Guan-Horng Liu**[*]**, Evangelos A. Theodorou**
Georgia Institute of Technology, USA
`{tianrong.chen, ghliu, evangelos.theodorou}@gatech.edu`

## ABSTRACT

Schrödinger Bridge (SB) is an entropy-regularized optimal transport problem that has received increasing attention in deep generative modeling for its mathematical flexibility compared to the Scored-based Generative Model (SGM). However, it remains unclear whether the optimization principle of SB relates to the modern training of deep generative models, which often rely on constructing log-likelihood objectives. This raises questions on the suitability of SB models as a principled alternative for generative applications. In this work, we present a novel computational framework for likelihood training of SB models grounded on *Forward-Backward Stochastic Differential Equations Theory* – a mathematical methodology appeared in stochastic optimal control that transforms the optimality condition of SB into a set of SDEs. Crucially, these SDEs can be used to construct the likelihood objectives for SB that, surprisingly, generalizes the ones for SGM as special cases. This leads to a new optimization principle that inherits the same SB optimality yet without losing applications of modern generative training techniques, and we show that the resulting training algorithm achieves comparable results on generating realistic images on MNIST, CelebA, and CIFAR10. Our code is available at `https://github.com/ghliu/SB-FBSDE`.

## 1 INTRODUCTION

Score-based Generative Model (SGM; Song et al. (2020)) is an emerging generative model class that has achieved remarkable results in synthesizing high-fidelity data (Song & Ermon, 2020; Kong et al., 2020a;b). Like many deep generative models, SGM seeks to find nonlinear functions that transform simple distributions (typically Gaussian) into complex, often intractable, data distributions. In SGM, this is done by first diffusing data to noise through a stochastic differential equation (SDE); then learning to *reverse* this diffusion process by regressing a network to the score function (*i.e.* the gradient of the log probability density) at each time step (Hyvärinen & Dayan, 2005). This reversed process thereby defines the generation (see Fig. 1).

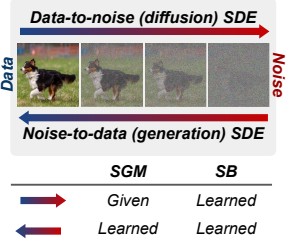

Figure 1: Both Score-based Generative Model (SGM) and Schrödinger Bridge (SB) transform between two distributions. While SGM requires pre-specifying the data-to-noise diffusion, SB instead *learns* the process.

Despite its empirical successes, SGM admits few limitations. First, the diffusion process has to obey a simple form (*e.g.* linear or degenerate drift) in order to compute the analytic score function for the regression purpose. Secondly, the diffusion process needs to run to sufficiently large time steps so that the end distribution is approximate Gaussian (Kong & Ping, 2021). For these reasons, SGM often takes a notoriously long time in generating data (Jolicoeur-Martineau et al., 2021), thereby limiting their practical usages compared to *e.g.* GANs or flow-based models (Ping et al., 2020; Karras et al., 2020b).

In the attempt to lift these restrictions, a line of recent works inspired by Schrödinger Bridge (SB; Schrödinger (1932)) has been proposed (De Bortoli et al., 2021; Wang et al., 2021; Vargas et al., 2021). SB – as an entropy-regularized optimal transport problem – seeks two optimal policies that transform back-and-forth between two *arbitrary* distributions in a *finite* horizon. The similarity between the

---

[*]Equal contribution. Order determined by coin flip. See Author Contributions section.

two problems (*i.e.* both involve transforming distributions) is evident, and the additional flexibility from SB is also attractive. To enable SB-inspired generative training, however, previous works require either ad-hoc multi-stage optimization or retreat to traditional SB algorithms, *e.g.* Iterative Proportional Fitting (IPF; Kullback (1968)). The underlying relation between the optimization principle of SB and modern generative training, in particular SGM, remains relatively unexplored, despite their intimately related problem formulations. More importantly, with the recent connection between SGM and log-likelihood computation (Song et al., 2021), it is crucial to explore whether there exists an alternative way of training SB that better respects, or perhaps generalizes, modern training of SGM, so as to solidify the suitability of SB as a principled generative model.

In this work, we present a fundamental connection between solving SB and training SGM. The difficulty arises immediately as one notices that the optimality condition of SB and the likelihood objective of SGM are represented by merely two distinct mathematical objects. While the former is characterized by two coupled partial differential equations (PDEs) (Léonard, 2013), the latter integrates over a notably complex SDE that resembles neither its diffusion nor reversed process (Song et al., 2021). Nevertheless, inspired by the recent advance on understanding deep learning through the optimal control perspective (Li & Hao, 2018; Liu et al., 2021a;b), we show that *Forward-Backward SDEs* – a mathematical methodology appearing in stochastic optimal control for solving nonlinear PDEs (Han et al., 2018) – paves an elegant way to connect the two objectives. The implication of our findings is nontrivial: It yields an exact log-likelihood expression of SB that precisely generalizes the one of SGM (Song et al., 2021) to fully nonlinear diffusion, thereby providing novel theoretical connections between the two model classes. Algorithmically, our framework suggests rich training procedures that resemble the joint optimization for diffusion flow-based models (Zhang & Chen, 2021) or more traditional IPF approaches (Kullback, 1968; De Bortoli et al., 2021). This allows one to marry the best of both worlds by improving the SB training with *e.g.* a SGM-inspired Langevin corrector (Song & Ermon, 2019). The resulting method, **SB-FBSDE**, generates encouraging images on MNIST, CelebA, and CIFAR10 and outperforms prior optimal transport models by a large margin.

Our method differs from the concurrent SB methods (De Bortoli et al., 2021; Vargas et al., 2021) in various aspects. First, while both prior methods rely on solving SB with *mean-matching* regression, our SB-FBSDE instead utilizes a *divergence-based* objectives (see §3.2). Secondly, neither of the prior methods focuses on log-likelihood training, which is the key finding in SB-FBSDE to bridge connections to SGM and adopt modern training improvements. Indeed, due to the difference in the underlying SDE classes,[1] their connections to SGM can only be made after time discretization by carefully choosing each step size (De Bortoli et al., 2021). In contrast, our theoretical connection is derived readily in continuous-time; hence unaffected by the choice of numerical discretization.

In summary, we present the following contributions.

- We present a novel computational framework, grounded on *Forward-Backward SDEs* theory, for computing the log-likelihood objectives of Schrödinger Bridge (SB) and solidifying their theoretical connections to Score-based Generative Model (SGM).

- Our framework suggests a new training principle that retains the mathematical flexibility from SB while enjoying advanced techniques from the modern generative training of SGM.

- We show that the resulting method – named **SB-FBSDE** – outperforms previous optimal transport-inspired baselines on synthesizing high-fidelity images and is comparable to other existing models.

**Notation.** We denote $p_t^{\mathrm{SDE}}(\mathbf{X}_t)$ as the marginal density driven by some SDE process $\mathbf{X}(t) \equiv \mathbf{X}_t$ until the time step $t \in [0, T]$. The time direction is aligned throughout this article such that $p_0$ and $p_T$ respectively correspond to the data and prior distributions. The gradient, divergence, and Hessian of a function $f(\boldsymbol{x})$, where $\boldsymbol{x} \in \mathbb{R}^n$, will be denoted as $\nabla_{\boldsymbol{x}} f \in \mathbb{R}^n$, $\nabla_{\boldsymbol{x}} \cdot f \in \mathbb{R}$, and $\nabla_{\boldsymbol{x}}^2 f \in \mathbb{R}^{n \times n}$.

## 2 PRELIMINARIES

### 2.1 SCORE-BASED GENERATIVE MODEL (SGM)

Given a data point $\mathbf{X}_0 \in \mathbb{R}^n$ sampled from an unknown data distribution $p_{\mathrm{data}}$, SGM first progressively diffuses the data towards random noise with the following forward SDE:

$$\mathrm{d}\mathbf{X}_t = f(t, \mathbf{X}_t)\mathrm{d}t + g(t)\mathrm{d}\mathbf{W}_t, \quad \mathbf{X}_0 \sim p_{\mathrm{data}}, \tag{1}$$

---

[1] We adopt the recent advance in SB theory (Caluya & Halder, 2021) that extends classical SB models (used in prior works) to the exact SDE class appearing in SGM. See Appendices A and C for more details.

where $f(\cdot, t) : \mathbb{R}^n \to \mathbb{R}^n$, $g(t) \in \mathbb{R}$, and $\mathbf{W}_t \in \mathbb{R}^n$ are the drift, diffusion, and standard Wiener process. Typically, $g(\cdot)$ is some monotonically increasing function such that for sufficiently large time steps, we have $p_T^{(1)} \approx p_{\text{prior}}$ resemble some prior distribution (*e.g.* Gaussian) at the terminal horizon $T$. Reversing (1) yields another SDE[2] that traverses backward in time (Anderson, 1982):

$$\mathrm{d}\mathbf{X}_t = [f - g^2 \, \nabla_{\boldsymbol{x}} \log p_t^{(1)}(\mathbf{X}_t)]\mathrm{d}t + g \, \mathrm{d}\mathbf{W}_t, \quad \mathbf{X}_T \sim p_T^{(1)}, \tag{2}$$

where $p_t^{(1)}$ corresponds to the marginal density of SDE (1) at time $t$, and $\nabla_{\boldsymbol{x}} \log p_t^{(1)}$ is known as the *score* function. These two stochastic processes are equivalent in the sense that their marginal densities are equal to each other throughout $t \in [0, T]$; in other words, $p_t^{(1)} \equiv p_t^{(2)}$.

When the drift $f$ is of simple structure, for instance linear (Ho et al., 2020) or simply degenerate (Song & Ermon, 2019), the conditional score function $\nabla_{\boldsymbol{x}} \log p_t^{(1)}(\mathbf{X}_t | \mathbf{X}_0 = \boldsymbol{x}_0) \equiv \nabla_{\boldsymbol{x}} \log p_{t|\boldsymbol{x}_0}$ admits an analytic solution at any time $t$. Hence, SGM proposes to train a parameterized score network $\mathbf{s}(t, \boldsymbol{x}; \theta)$ by regressing its outputs to the ground-truth values, *i.e.* $\mathbb{E}[\lambda(t) \| \mathbf{s}(t, \mathbf{X}_t; \theta) - \nabla_{\boldsymbol{x}} \log p_{t|\boldsymbol{x}_0} \|^2]$, where the expectation is taken over the SDE (1). In practice, $\lambda(t)$ is some hand-designed weighting function that largely affects the performance. Recent works (Song et al., 2021; Huang et al., 2021) have shown that the log-likelihood of SGM, despite being complex, can be lower-bounded as follows:

$$\log p_0^{\text{SGM}}(\boldsymbol{x}_0) \geq \mathcal{L}_{\text{SGM}}(\boldsymbol{x}_0; \theta) = \mathbb{E}\left[\log p_T(\mathbf{X}_T)\right] - \int_0^T \mathbb{E}\left[\frac{1}{2}g^2 \| \mathbf{s}_t \|^2 + \nabla_{\boldsymbol{x}} \cdot \left(g^2 \mathbf{s}_t - f\right)\right] \mathrm{d}t, \tag{3}$$

$$= \mathbb{E}\left[\log p_T(\mathbf{X}_T)\right] - \int_0^T \mathbb{E}\left[\frac{1}{2}g^2 \| \mathbf{s}_t - \nabla_{\boldsymbol{x}} \log p_{t|\boldsymbol{x}_0} \|^2 - \frac{1}{2}\| g\nabla_{\boldsymbol{x}} \log p_{t|\boldsymbol{x}_0} \|^2 - \nabla_{\boldsymbol{x}} \cdot f\right] \mathrm{d}t,$$

where $\mathbf{s}_t \equiv \mathbf{s}(t, \boldsymbol{x}; \theta)$ and the expectation is taken over the SDE (1) given a data point $\mathbf{X}_0 = \boldsymbol{x}_0$. This objective (3) suggests a principled choice of $\lambda(t) := g(t)^2$. After training, SGM simply substitutes the score function with the learned score network $\mathbf{s}(t, \boldsymbol{x}; \theta)$ to generate data from $p_{\text{prior}}$,

$$\mathrm{d}\mathbf{X}_t = [f - g^2 \, \mathbf{s}(t, \mathbf{X}_t; \theta)]\mathrm{d}t + g \, \mathrm{d}\mathbf{W}_t, \quad \mathbf{X}_T \sim p_{\text{prior}}. \tag{4}$$

It is important to notice that $p_{\text{prior}}$ needs *not* equal $p_T^{(1)}$ in practice, and the approximation is close only through a careful design of (1). Notably, designing the diffusion $g(t)$ can be particularly problematic as it affects both the approximation $p_T^{(1)} \approx p_{\text{prior}}$ and the training via the weighting $\lambda(t)$; hence can lead to unstable training (Nichol & Dhariwal, 2021). In contrast, Schrödinger Bridge considers a more flexible framework for designing the forward diffusion that requires minimal manipulation.

## 2.2 Schrödinger Bridge (SB)

Following the dynamic expression of SB (Pavon & Wakolbinger, 1991; Dai Pra, 1991), consider

$$\min_{\mathbb{Q} \in \mathcal{P}(p_{\text{data}}, p_{\text{prior}})} D_{\text{KL}}(\mathbb{Q} \,\|\, \mathbb{P}), \tag{5}$$

where $\mathbb{Q} \in \mathcal{P}(p_{\text{data}}, p_{\text{prior}})$ belongs to a set of path measure with $p_{\text{data}}$ and $p_{\text{prior}}$ as its marginal densities at $t = 0$ and $T$. On the other hand, $\mathbb{P}$ denotes a reference measure, which we will set to the path measure of (1) for later convenience. The optimality condition to (5) is characterized by two PDEs that are coupled through their boundary conditions. We summarize the related result below.

**Theorem 1** (SB optimality; Chen et al. (2021); Pavon & Wakolbinger (1991); Caluya & Halder (2021)). *Let $\Psi(t, \boldsymbol{x})$ and $\widehat{\Psi}(t, \boldsymbol{x})$ be the solutions to the following PDEs:*

$$\begin{cases} \frac{\partial \Psi}{\partial t} = -\nabla_{\boldsymbol{x}} \Psi^{\mathsf{T}} f - \frac{1}{2} \text{Tr}(g^2 \nabla_{\boldsymbol{x}}^2 \Psi) \\ \frac{\partial \widehat{\Psi}}{\partial t} = -\nabla_{\boldsymbol{x}} \cdot (\widehat{\Psi} f) + \frac{1}{2} \text{Tr}(g^2 \nabla_{\boldsymbol{x}}^2 \widehat{\Psi}) \end{cases} \quad s.t. \ \Psi(0, \cdot)\widehat{\Psi}(0, \cdot) = p_{\text{data}}, \ \Psi(T, \cdot)\widehat{\Psi}(T, \cdot) = p_{\text{prior}} \tag{6}$$

*Then, the solution to the optimization (5) can be expressed by the path measure of the following forward (7a), or equivalently backward (7b), SDE:*

$$\mathrm{d}\mathbf{X}_t = [f + g^2 \, \nabla_{\boldsymbol{x}} \log \Psi(t, \mathbf{X}_t)]\mathrm{d}t + g \, \mathrm{d}\mathbf{W}_t, \quad \mathbf{X}_0 \sim p_{\text{data}}, \tag{7a}$$

$$\mathrm{d}\mathbf{X}_t = [f - g^2 \, \nabla_{\boldsymbol{x}} \log \widehat{\Psi}(t, \mathbf{X}_t)]\mathrm{d}t + g \, \mathrm{d}\mathbf{W}_t, \quad \mathbf{X}_T \sim p_{\text{prior}}, \tag{7b}$$

*where $\nabla_{\boldsymbol{x}} \log \Psi(t, \mathbf{X}_t)$ and $\nabla_{\boldsymbol{x}} \log \widehat{\Psi}(t, \mathbf{X}_t)$ are the optimal forward and backward drifts for SB.*

---

[2]Hereafter, we will sometimes drop $f \equiv f(t, \mathbf{X}_t)$ and $g \equiv g(t)$ for brevity.

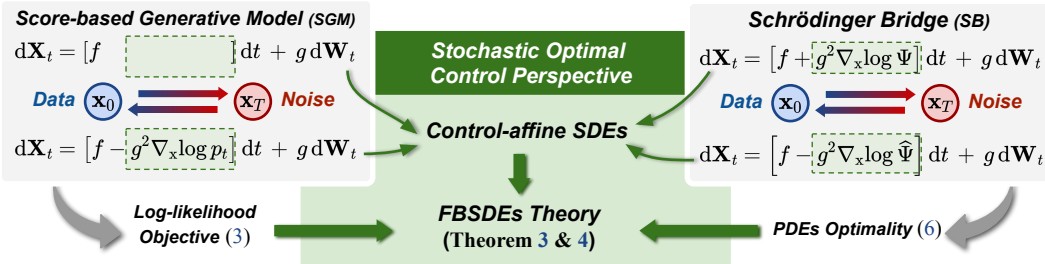

Figure 2: Schematic diagram of the our stochastic optimal control interpretation, and how it connects the objective of SGM (3) and optimality of SB (6) through Forward-Backward SDEs theory.

Similar to the forward/backward processes in SGM, the stochastic processes of SB in (7a) and (7b) are also equivalent in the sense that $\forall t \in [0, T]$, $p_t^{(7a)} \equiv p_t^{(7b)} \equiv p_t^{SB}$. In fact, its marginal density obeys a factorization principle: $p_t^{SB}(\mathbf{X}_t) = \Psi(t, \mathbf{X}_t)\widehat{\Psi}(t, \mathbf{X}_t)$.

To construct the generative pipeline from (7b), one requires solving the PDEs in (6) to obtain $\widehat{\Psi}$. Unfortunately, these PDEs are hard to solve even for low-dimensional systems (Renardy & Rogers, 2006); let alone for generative applications. Indeed, previous works either have to replace the original Schrödinger Bridge ($p_{\text{data}} \leftrightarrows p_{\text{prior}}$) with multiple stages, $p_{\text{data}} \leftrightarrows p_{\text{middle}} \leftrightarrows p_{\text{prior}}$, so that each segment admits an analytic solution (Wang et al., 2021), or consider the following half-bridge ($p_{\text{data}} \leftarrow p_{\text{prior}}$ *vs.* $p_{\text{data}} \rightarrow p_{\text{prior}}$) optimization (De Bortoli et al., 2021; Vargas et al., 2021),

$$\mathbb{Q}^{(1)} := \underset{\mathbb{Q} \in \mathcal{P}(\cdot, p_{\text{prior}})}{\arg\min} \ D_{\text{KL}}(\mathbb{Q} \, || \, \mathbb{Q}^{(0)}), \quad \mathbb{Q}^{(0)} := \underset{\mathbb{Q} \in \mathcal{P}(p_{\text{data}}, \cdot)}{\arg\min} \ D_{\text{KL}}(\mathbb{Q} \, || \, \mathbb{Q}^{(1)})$$

which can be solved with IPF algorithm (Kullback, 1968) starting from $\mathbb{Q}^{(0)} := \mathbb{P}$. In the following section, we will present a scalable computational framework for solving the optimality PDEs in (6) and show that it paves an elegant way connecting the optimality principle of SB (6) to the parameterized log-likelihood of SGM (3).

## 3 APPROACH

We motivate our approach starting from some control-theoretic observation (see Fig. 2). Notice that both SGM and SB consist of forward and backward SDEs with similar structures. From the stochastic control perspective, these SDEs belong to the class of *control-affine* SDEs with additive noise:

$$\text{d}\mathbf{X}_t = \boldsymbol{A}(t, \mathbf{X}_t)\text{d}t + \boldsymbol{B}(t, \mathbf{X}_t)\mathbf{u}(t, \mathbf{X}_t)\text{d}t + \boldsymbol{C}(t) \, \text{d}\mathbf{W}_t. \tag{8}$$

It is clear that the control-affine SDE (8) includes all SDEs (1,2,4,7) appearing in §2 by considering $(\boldsymbol{A}, \boldsymbol{B}, \boldsymbol{C}) := (f, \boldsymbol{I}, g)$ and different interpretations of the *control* variables $\mathbf{u}(t, \mathbf{X}_t)$. This implies that the optimization processes of both SGM and SB can be aligned through the lens of *stochastic optimal control* (SOC). Indeed, both problems can be interpreted as seeking some time-varying control policy, either the score function $\nabla_{\boldsymbol{x}} \log p_{t|\boldsymbol{x}_0}$ in SGM or $\nabla_{\boldsymbol{x}} \log \widehat{\Psi}$ in SB, that minimizes some objectives, (3) *vs.* (5), while subjected to some control-affine SDEs, (1,2) *vs.* (7). In what follows, we will show that a specific mathematical methodology in nonlinear SOC literature – called *Forward-Backward SDEs* theory (FBSDEs; see Ma et al. (1999)) – links the optimality condition of SB (6) to the log-likelihood objectives of SGM (3). All proofs are left to Appendix B.

### 3.1 FORWARD-BACKWARD SDEs (FBSDEs) REPRESENTATION FOR SB

The theory of FBSDEs establishes an innate connection between different classes of PDEs and forward-backward SDEs. Below we introduce the following connection related to our problem.

**Lemma 2** (Nonlinear Feynman-Kac;[3] Exarchos & Theodorou (2018)). *Consider the coupled SDEs*

$$\begin{cases} \text{d}\mathbf{X}_t = f(t, \mathbf{X}_t)\text{d}t + G(t, \mathbf{X}_t)\text{d}\mathbf{W}_t, & \mathbf{X}_0 = \boldsymbol{x}_0 \tag{9a} \\ \text{d}\mathbf{Y}_t = -h(t, \mathbf{X}_t, \mathbf{Y}_t, \mathbf{Z}_t)\text{d}t + \mathbf{Z}(t, \mathbf{X}_t)^{\mathsf{T}}\text{d}\mathbf{W}_t, & \mathbf{Y}_T = \varphi(\mathbf{X}_T) \tag{9b} \end{cases}$$

---

[3]Lemma 2 can be viewed as the nonlinear extension of the celebrated Feynman-Kac formula (Karatzas & Shreve, 2012), which characterizes the connection between linear PDEs and forward SDEs.

*where the functions $f$, $G$, $h$, and $\varphi$ satisfy proper regularity conditions[4] so that there exists a pair of unique strong solutions satisfying (9). Now, consider the following second-order parabolic PDE and suppose $v(t, \boldsymbol{x}) \equiv v$ is once continuously differentiable in $t$ and twice in $\boldsymbol{x}$, i.e. $v \in C^{1,2}$,*

$$\frac{\partial v}{\partial t} + \frac{1}{2} \operatorname{Tr}(\nabla_{\boldsymbol{x}}^2 v \, GG^{\mathsf{T}}) + \nabla_{\boldsymbol{x}} v^{\mathsf{T}} f + h(t, \boldsymbol{x}, v, G^{\mathsf{T}} \nabla_{\boldsymbol{x}} v) = 0, \quad v(T, \boldsymbol{x}) = \varphi(\boldsymbol{x}), \qquad (10)$$

*then the solution to (9) coincides with the solution to (10) along paths generated by the forward SDE (9a) almost surely, i.e., the following stochastic representation (known as the nonlinear Feynman-Kac relation) is valid:*

$$v(t, \mathbf{X}_t) = \mathbf{Y}_t \qquad and \qquad G(t, \mathbf{X}_t)^{\mathsf{T}} \nabla_{\boldsymbol{x}} v(t, \mathbf{X}_t) = \mathbf{Z}_t. \qquad (11)$$

Lemma 2 states that solutions to a certain class of nonlinear (via the function $h$ in (10)) PDEs can be represented by solutions to a set of forward-backward SDEs (9) through the transformation (11), and this relation can be extended to the viscosity case (Pardoux & Peng (1992); see also Appendix B). Note that $\mathbf{Y}_t$ is the solution to the backward SDE (9b) whose randomness is driven by the forward SDE (9a). Indeed, it is clear from (11) that $\mathbf{Y}_t$ (hence also $\mathbf{Z}_t$) is a time-varying function of $\mathbf{X}_t$. Since the $v$ appearing in the nonlinear Feynman-Kac relation (11) takes the random vector $\mathbf{X}_t$ as its argument, $v(t, \mathbf{X}_t)$ shall also be understood as a random variable. Finally, it is known that the original (deterministic) PDE solution $v(t, \boldsymbol{x})$ can be recovered by taking conditional expectation, *i.e.*

$$v(t, \boldsymbol{x}) = \mathbb{E}_{\mathbf{X}_t \sim (9a)}[\mathbf{Y}_t | \mathbf{X}_t = \boldsymbol{x}] \quad \text{and} \quad G(t, \boldsymbol{x})^{\mathsf{T}} \nabla_{\boldsymbol{x}} v(t, \boldsymbol{x}) = \mathbb{E}_{\mathbf{X}_t \sim (9a)}[\mathbf{Z}_t | \mathbf{X}_t = \boldsymbol{x}]. \qquad (12)$$

Since it is often computationally favorable to solve SDEs rather than PDEs, Lemma 2 has been widely used as a scalable method for solving high-dimensional PDEs (Han et al., 2018; Pereira et al., 2019). Take SOC applications for instance, their PDE optimality condition can be characterized by (11) under proper conditions, with the optimal control given in the form of $\mathbf{Z}_t$. Hence, one can adopt Lemma 2 to solve the underlying FBSDEs, rather than the original PDE optimality, for the optimal control. Despite seemingly attractive, whether these principles can be extended to SB, whose optimality conditions are given by *two coupled PDEs* in (6), remains unclear. Below we derive a similar FBSDEs representation for SB.

**Theorem 3** (FBSDEs to SB optimality (6)). *Consider the following set of coupled SDEs,*

$$\begin{cases} \mathrm{d}\mathbf{X}_t = (f + g\mathbf{Z}_t) \, \mathrm{d}t + g \mathrm{d}\mathbf{W}_t & (13a) \\[2mm] \mathrm{d}\mathbf{Y}_t = \frac{1}{2}\mathbf{Z}_t^{\mathsf{T}}\mathbf{Z}_t \mathrm{d}t + \mathbf{Z}_t^{\mathsf{T}} \mathrm{d}\mathbf{W}_t & (13b) \\[2mm] \mathrm{d}\widehat{\mathbf{Y}}_t = \left( \frac{1}{2}\widehat{\mathbf{Z}}_t^{\mathsf{T}}\widehat{\mathbf{Z}}_t + \nabla_{\boldsymbol{x}} \cdot (g\widehat{\mathbf{Z}}_t - f) + \widehat{\mathbf{Z}}_t^{\mathsf{T}}\mathbf{Z}_t \right) \mathrm{d}t + \widehat{\mathbf{Z}}_t^{\mathsf{T}} \mathrm{d}\mathbf{W}_t & (13c) \end{cases}$$

*where $f$ and $g$ satisfy the same regularity conditions in Lemma 2 (see Footnote 4), and the boundary conditions are given by $\mathbf{X}(0) = \boldsymbol{x}_0$ and $\mathbf{Y}_T + \widehat{\mathbf{Y}}_T = \log p_{\mathrm{prior}}(\mathbf{X}_T)$. Suppose $\Psi, \widehat{\Psi} \in C^{1,2}$, then the nonlinear Feynman-Kac relations between the FBSDEs (13) and PDEs (6) are given by*

$$\begin{aligned} \mathbf{Y}_t &\equiv \mathbf{Y}(t, \mathbf{X}_t) = \log \Psi(t, \mathbf{X}_t), & \mathbf{Z}_t &\equiv \mathbf{Z}(t, \mathbf{X}_t) = g\nabla_{\boldsymbol{x}} \log \Psi(t, \mathbf{X}_t), \\ \widehat{\mathbf{Y}}_t &\equiv \widehat{\mathbf{Y}}(t, \mathbf{X}_t) = \log \widehat{\Psi}(t, \mathbf{X}_t), & \widehat{\mathbf{Z}}_t &\equiv \widehat{\mathbf{Z}}(t, \mathbf{X}_t) = g\nabla_{\boldsymbol{x}} \log \widehat{\Psi}(t, \mathbf{X}_t). \end{aligned} \qquad (14)$$

*Furthermore, $(\mathbf{Y}_t, \widehat{\mathbf{Y}}_t)$ obey the following relation:*

$$\mathbf{Y}_t + \widehat{\mathbf{Y}}_t = \log p_t^{\mathrm{SB}}(\mathbf{X}_t).$$

The FBSDEs for SB (13) share a similar forward-backward structure as in (9), where (13a) and (13b,13c) respectively represent the forward and backward SDEs. One can verify that the forward SDE (13a) coincides with the *optimal* forward SDE (7a) with the substitution $\mathbf{Z}_t = g\nabla_{\boldsymbol{x}} \log \Psi$. In other words, these FBSDEs provide a *local* representation of $\log \Psi$ and $\log \widehat{\Psi}$ evaluated on the optimal path governed by (7a). Since $\mathbf{Z}_t$ and $\widehat{\mathbf{Z}}_t$ can be understood as the forward/backward policies, in a similar spirit of policy-based methods (Pereira et al., 2020; Schulman et al., 2015), that guide the SDE processes of SB, they sufficiently characterize the SB model. Hence, our next step is to derive a proper training objective to optimize these policies.

---

[4] Yong & Zhou (1999); Kobylanski (2000) require $f$, $G$, $h$, and $\varphi$ to be continuous, $f$ and $G$ to be uniformly Lipschitz in $\boldsymbol{x}$, and $h$ to satisfy quadratic growth condition in $\boldsymbol{z}$.

## 3.2 Log-likelihood Computation of SB

Theorem 3 has an important implication: It suggests that given a path sampled from the forward SDE (13a), the solutions to the backward SDEs (13b,13c) at $t = 0$ provide an unbiased estimation of the log-likelihood of the data point $\boldsymbol{x}_0$, *i.e.* $\mathbb{E}\left[\mathbf{Y}_0 + \widehat{\mathbf{Y}}_0|\mathbf{X}_0 = \boldsymbol{x}_0\right] = \log p_0^{\mathrm{SB}}(\boldsymbol{x}_0) = \log p_{\mathrm{data}}(\boldsymbol{x}_0)$, where $\mathbf{X}_t$ is sampled from (13a). We now state our main result, which makes this observation formal:

**Theorem 4** (Log-likelihood of SB model). *Given the solution satisfying the FBSDE system in (13), the log-likelihood of the SB model $(\mathbf{Z}_t, \widehat{\mathbf{Z}}_t)$, at a data point $\boldsymbol{x}_0$, can be expressed as*

$$\log p_0^{\mathrm{SB}}(\boldsymbol{x}_0) = \mathbb{E}\left[\log p_T(\mathbf{X}_T)\right] - \int_0^T \mathbb{E}\left[\frac{1}{2}\|\mathbf{Z}_t\|^2 + \frac{1}{2}\|\widehat{\mathbf{Z}}_t - g\nabla_{\boldsymbol{x}}\log p_t^{\mathrm{SB}} + \mathbf{Z}_t\|^2\right.$$
$$\left. -\frac{1}{2}\|g\nabla_{\boldsymbol{x}}\log p_t^{\mathrm{SB}} - \mathbf{Z}_t\|^2 - \nabla_{\boldsymbol{x}}\cdot f\right]\mathrm{d}t \qquad (15)$$

$$= \mathbb{E}\left[\log p_T(\mathbf{X}_T)\right] - \int_0^T \mathbb{E}\left[\frac{1}{2}\|\mathbf{Z}_t\|^2 + \frac{1}{2}\|\widehat{\mathbf{Z}}_t\|^2 + \nabla_{\boldsymbol{x}}\cdot(g\widehat{\mathbf{Z}}_t - f) + \widehat{\mathbf{Z}}_t^\mathsf{T}\mathbf{Z}_t\right]\mathrm{d}t, \quad (16)$$

*where the expectation is taken over the forward SDE (13a) with the initial condition $\mathbf{X}_0 = \boldsymbol{x}_0$.*

Similar to (3), Theorem 4 suggests a parameterized lower bound to the log-likelihoods, *i.e.* $\log p_0^{\mathrm{SB}}(\boldsymbol{x}_0) \geq \mathcal{L}_{\mathrm{SB}}(\boldsymbol{x}_0; \theta, \phi)$ where $\mathcal{L}_{\mathrm{SB}}(\boldsymbol{x}_0; \theta, \phi)$ shares the same expression in (16) except that $\mathbf{Z}_t \approx \mathbf{Z}(t, \boldsymbol{x}; \theta)$ and $\widehat{\mathbf{Z}}_t \approx \widehat{\mathbf{Z}}(t, \boldsymbol{x}; \phi)$ are approximated with some parameterized models (*e.g.* DNNs). Note that $\nabla_{\boldsymbol{x}}\log p_t^{\mathrm{SB}}$ is *intractable* in practice for any nontrivial $(\mathbf{Z}_t, \widehat{\mathbf{Z}}_t)$. Hence, we use the divergence-based objective in (16) as our training objective of both policies.

**Connection to score-based models.** Recall Fig. 2 and compare the parameterized log-likelihoods of SB (16) to SGM (3); one can verify that $\mathcal{L}_{\mathrm{SB}}$ collapses to $\mathcal{L}_{\mathrm{SGM}}$ when $(\mathbf{Z}_t, \widehat{\mathbf{Z}}_t) := (\mathbf{0}, g\,\mathbf{s}_t)$. From the SB perspective, this occurs only when $p_T^{(1)} = p_{\mathrm{prior}}$. Since no effort is required in the forward process to reach $p_{\mathrm{prior}}$, the optimal forward control $\mathbf{Z}_t$, by definition, degenerates; thereby making the backward control $\widehat{\mathbf{Z}}_t$ collapses to the score function. However, in any case when $p_T^{(1)} \neq p_{\mathrm{prior}}$, for instance when the diffusion SDEs are improperly designed, the forward policy $\mathbf{Z}_t$ steers the diffusion process back to $p_{\mathrm{prior}}$, while its backward counterpart $\widehat{\mathbf{Z}}_t$ compensates the reversed process accordingly. From this view, SB alleviates the problematic design in SGM by enlarging the class of diffusion processes to accept *nonlinear* drifts and providing an optimization principle on learning these processes. Moreover, Theorem 4 generalizes the log-likelihood training from SGM to SB.

**Connection to flow-based models.** Interestingly, the log-likelihood computation in Theorem 4, where we use a path $\{\mathbf{X}_t\}_{t\in[0,T]}$ sampled from a data point $\mathbf{X}_0$ to parameterize its log-likelihood, resembles modern training of (deterministic) flow-based models (Grathwohl et al., 2018), which have recently been shown to admit a close relation to SGM (Song et al., 2020; Gong & Li, 2021). The connection is built on the concept of *probability flow* – which suggests that the marginal density of an SDE can be evaluated through an ordinary differential equation (ODE). Below, we provide a similar flow representation for SB, further strengthening their connection to modern generative models.

**Corollary 5** (Probability flow for SB). *The following ODE characterizes the probability flow of the optimal processes of SB (7) in the sense that $\forall t,\ p_t^{(17)} \equiv p_t^{(7)} \equiv p_t^{\mathrm{SB}}$.*

$$\mathrm{d}\mathbf{X}_t = \left[f + g\mathbf{Z}(t, \mathbf{X}_t) - \frac{1}{2}g\left(\mathbf{Z}(t, \mathbf{X}_t) + \widehat{\mathbf{Z}}(t, \mathbf{X}_t)\right)\right]\mathrm{d}t \qquad (17)$$

One can verify (see Remark 10 in §B) that computing the log-likelihood of this ODE model (17) using flow-based training techniques indeed recovers the training objective of SB derived in (16).

## 3.3 Practical Implementation

In this section, we detail the implementation of our FBSDE-inspired SB model, named **SB-FBSDE**.

**Training process.** We treat the log-likelihood in (16) as our training objective, where the divergence can be can be estimated efficiently following Hutchinson (1989). This immediately distinguishes SB-FBSDE from prior SB models (De Bortoli et al., 2021; Vargas et al., 2021), which instead rely on

---

**Algorithm 1** Likelihood training of SB-FBSDE

---

**Input:** boundary distributions $p_{\text{data}}$ and $p_{\text{prior}}$,
  parameterized policies $\mathbf{Z}(\cdot, \cdot; \theta)$ and $\widehat{\mathbf{Z}}(\cdot, \cdot; \phi)$
**repeat**
  **if** memory resource is affordable **then**
    **run** Algorithm 2.
  **else**
    **run** Algorithm 3.
  **end if**
**until** converges

---

**Algorithm 2** Joint (diffusion flow-based) training

---

**for** $k = 1$ **to** $K$ **do**
  Sample $\mathbf{X}_{t \in [0,T]}$ from (13a) where $\boldsymbol{x}_0 \sim p_{\text{data}}$
  (computational graph retained).
  Compute $\mathcal{L}_{\text{SB}}(\boldsymbol{x}_0; \theta, \phi)$ with (16).
  Update $(\theta, \phi)$ with $\nabla_{\theta, \phi} \mathcal{L}_{\text{SB}}(\boldsymbol{x}_0; \theta, \phi)$.
**end for**

---

**Algorithm 3** Alternate (IPF-based) training

---

**Input:** Caching frequency $M$
**for** $k = 1$ **to** $K$ **do**
  **if** $k\%M == 0$ **then**
    Sample $\mathbf{X}_{t \in [0,T]}$ from (13a) where $\boldsymbol{x}_0 \sim p_{\text{data}}$
    (computational graph discarded).
  **end if**
  Compute $\widetilde{\mathcal{L}}_{\text{SB}}(\boldsymbol{x}_0; \phi)$ with (18).
  Update $\phi$ with gradient $\nabla_\phi \widetilde{\mathcal{L}}_{\text{SB}}(\boldsymbol{x}_0; \phi)$.
**end for**
**for** $k = 1$ **to** $K$ **do**
  **if** $k\%M == 0$ **then**
    Sample $\mathbf{X}_{t \in [0,T]}$ from (7b) where $\boldsymbol{x}_T \sim p_{\text{prior}}$
    (computational graph discarded).
  **end if**
  Compute $\mathcal{L}_{\text{SB}}(\boldsymbol{x}_T; \theta)$ with (19).
  Update $\theta$ with gradient $\nabla_\theta \widetilde{\mathcal{L}}_{\text{SB}}(\boldsymbol{x}_T; \theta)$.
**end for**

---

regression-based objectives.[5] For low-dimensional datasets, we simply perform joint optimization, $\max \mathcal{L}_{\text{SB}}(\boldsymbol{x}_0; \theta, \phi)$, to train the parameterized policies $\mathbf{Z}(\cdot, \cdot; \theta)$ and $\widehat{\mathbf{Z}}(\cdot, \cdot; \phi)$. For higher-dimensional (*e.g.* image) datasets, however, it can be prohibitively expensive to keep the entire computational graph. In these cases, we follow De Bortoli et al. (2021) by caching the sampled trajectories in a reply buffer and refreshing them in a lower frequency basis (around 1500 iterations). Although this implies that the gradient path w.r.t. $\theta$ will be discarded, we can leverage the symmetric structure of SB and re-derive the log-likelihood for the sampled noise, *i.e.* $\mathcal{L}_{\text{SB}}(\boldsymbol{x}_T)$, based on the backward trajectories sampled from (7b). We leave the derivation to Theorem 11 in §B due to space constraint. This results in an alternate training between the following two objectives after dropping all unrelated terms,

$$\widetilde{\mathcal{L}}_{\text{SB}}(\boldsymbol{x}_0; \phi) = -\int_0^T \mathbb{E}_{\mathbf{X}_t \sim (7a)} \left[ \frac{1}{2} \|\widehat{\mathbf{Z}}(t, \mathbf{X}_t; \phi)\|^2 + g \nabla_{\boldsymbol{x}} \cdot \widehat{\mathbf{Z}}(t, \mathbf{X}_t; \phi) + \mathbf{Z}_t^{\mathsf{T}} \widehat{\mathbf{Z}}(t, \mathbf{X}_t; \phi) \right] \mathrm{d}t, \quad (18)$$

$$\widetilde{\mathcal{L}}_{\text{SB}}(\boldsymbol{x}_T; \theta) = -\int_0^T \mathbb{E}_{\mathbf{X}_t \sim (7b)} \left[ \frac{1}{2} \|\mathbf{Z}(t, \mathbf{X}_t; \theta)\|^2 + g \nabla_{\boldsymbol{x}} \cdot \mathbf{Z}(t, \mathbf{X}_t; \theta) + \widehat{\mathbf{Z}}_t^{\mathsf{T}} \mathbf{Z}(t, \mathbf{X}_t; \theta) \right] \mathrm{d}t. \quad (19)$$

Our training process is summarized in Alg. 1. While the joint training scheme in Alg. 2 resembles recent diffusion flow-based models (Zhang & Chen, 2021), the alternate training in Alg. 3 relates to the classical IPF (De Bortoli et al., 2021), despite differing in the underlying objectives. Empirically, the joint training scheme can converge faster yet at the cost of introducing memory complexity. We highlight these flexible training procedures arising from the unified viewpoint provided in Theorem 4. Hereafter, we refer to each cycle, *i.e.* $2K$ training steps, in Alg. 3 as a *training stage* of SB-FBSDE.

**Generative process.** While the generative processes for SB can be performed as simply as propagating (7b) given the trained policy $\widehat{\mathbf{Z}}(\cdot, \cdot; \phi)$, it has been constantly observed that adopting Langevin sampling to the generative process greatly improves performance (Song et al., 2020). This procedure, often referred to as the *Langevin corrector*, requires knowing the score function $\nabla_{\boldsymbol{x}} \log p_t$. For SB, we can estimate its value by recalling (see §2.2) that $\mathbf{Z}_t + \widehat{\mathbf{Z}}_t = g \nabla_{\boldsymbol{x}} \log p_t^{\text{SB}}$. This results in the following predictor-corrector sampling procedure (see Alg. 4 in Appendix D for more details).

$$\text{Predict step:} \quad \mathbf{X}_t \leftarrow \mathbf{X}_t + g \, \widehat{\mathbf{Z}}_t \Delta t + \sqrt{g \Delta t} \, \epsilon \quad (20)$$

$$\text{Correct step:} \quad \mathbf{X}_t \leftarrow \mathbf{X}_t + \frac{\sigma_t}{g} (\mathbf{Z}_t + \widehat{\mathbf{Z}}_t) + \sqrt{2\sigma_t} \, \epsilon \quad (21)$$

where $\epsilon \sim \mathcal{N}(\mathbf{0}, \boldsymbol{I})$ and $\sigma_t$ is the pre-specified noise scales (see (59) in Appendix D).

**Limitations & efficiency.** The main computational burden of our method comes from the computation of the divergence and maintaining two distinct networks. Despite it typically increases the memory by 2~2.5 times compared to SGM, we empirically observe that the divergence-based training converges much faster per iteration than standard regression. As a result, SB-FBSDE admits comparable training time (+6.8% in our CIFAR10 experiment) compared to SGM, yet with a substantially fewer sampling time (-80%) due to adopting nonlinear SDEs.

---

[5]In fact, their regression targets may be recovered from (15) under proper transformation; see Appendix C.

# 4 EXPERIMENTS

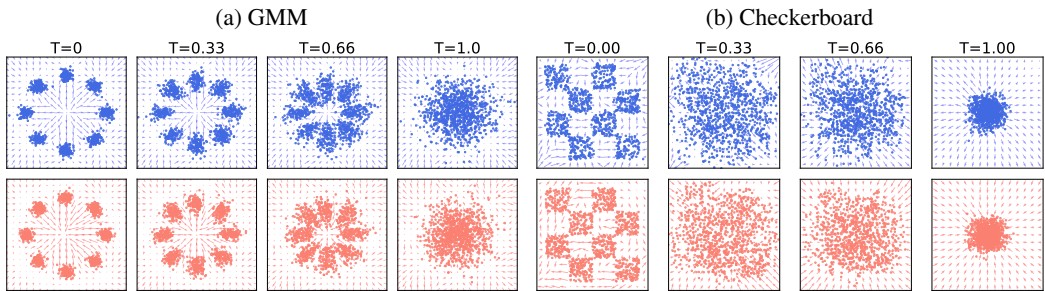

Figure 3: Validation of our SB-FBSDE model on two synthetic toy datasets that represent continuous and discontinuous distributions. *Upper*: Generation ($p_{\text{data}} \leftarrow p_{\text{prior}}$) process with the backward vector field $\widehat{\mathbf{Z}}(\cdot, \cdot; \phi)$. *Bottom*: Diffusion ($p_{\text{data}} \rightarrow p_{\text{prior}}$) process with the forward vector field $\mathbf{Z}(\cdot, \cdot; \theta)$.

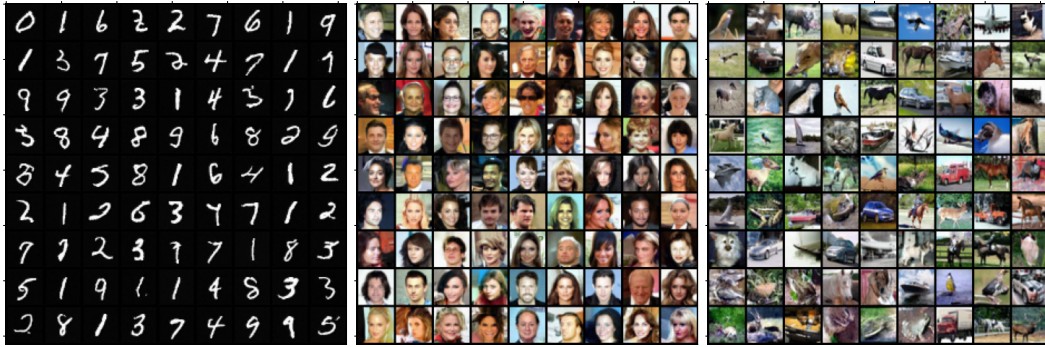

Figure 4: Uncurated samples from our SB-FBSDE models trained on MNIST (left), resized CelebA (middle) and CIFAR10 (right). More images can be found in Appendix E.

**Setups.** We testify SB-FBSDE on two toy datasets and three image datasets, *i.e.* MNIST, CelebA,[6] and CIFAR10. $p_{\text{prior}}$ is set to a zero-mean Gaussian whose variance varies for each task and can be computed according to Song & Ermon (2020). We parameterize $\mathbf{Z}(\cdot, \cdot; \theta)$ and $\widehat{\mathbf{Z}}(\cdot, \cdot; \phi)$ with residual-based networks for toy datasets and consider Unet (Ronneberger et al., 2015) and NCSN++ (Song et al., 2020) respectively for MNIST/CelebA and CIFAR10. All networks adopt position encoding and are trained with AdamW (Loshchilov & Hutter, 2017) on a TITAN RTX. We adopt VE-SDE (*i.e.* $f := \mathbf{0}$; see Song et al. (2020)) as our SDE backbone, which implies that in order to achieve reasonable performance, SB must *learn* a proper data-to-noise diffusion process. On all datasets, we set the horizon $T = 1.0$ and solve the SDEs via the Euler-Maruyama method. The interval $[0, T]$ is discretized into 200 steps for CIFAR10 and 100 steps for all other datasets, which are much fewer than the ones in SGM ($\geq$1000 steps). Other details are left in Appendix D.

**Toy datasets.** We first validate our joint optimization (*i.e.* Alg 2) on generating a mixture of Gaussian and checkerboard (adopted from Grathwohl et al. (2018)) as the representatives of continuous and discontinuous distributions. Figure 3 shows how the learned policies, *i.e.* $\mathbf{Z}(\cdot, \cdot; \theta)$ and $\widehat{\mathbf{Z}}(\cdot, \cdot; \phi)$, construct the vector fields that progressively transport samples back-and-forth between $p_{\text{prior}}$ and $p_{\text{data}}$. The vector fields can be highly nonlinear and dissimilar to each other. This resembles neither SGMs, whose forward vector field must obey linear structure, nor flow-based models, whose vector fields are simply with opposite directions. We highlight this as a distinct feature arising from SB models.

**Image datasets.** Next, we validate our alternate training (*i.e.* Alg 3) on high-dimensional image generation. The generated images for MNIST, CelebA, and CIFAR10 are presented in Fig. 4, which clearly suggest that our SB-FBSDE is able to synthesize high-fidelity images. More uncurated images can be founded in Appendix E. Regarding the quantitative evaluation, Table 1 summarizes the negative log-likelihood (NLL; measured in bits/dim) and the Fréchet Inception Distance score (FID; Heusel et al. (2017)) on CIFAR10. For our SB-FBSDE, we compute the NLL on the test set using Corollary 5, in a similar vein to SGMs and flow-based models, and report the FID over 50k

---

[6]We follow a similar setup of prior SB models (De Bortoli et al., 2021) and resize the image size to 32.

Table 1: CIFAR10 evaluation using negative log-likelihood (NLL; bits/dim) on the test set and sample quality (FID score) w.r.t. the training set. Our **SB-FBSDE** outperforms other optimal transport baselines by a large margin and is comparable to existing generative models.

| Model Class | Method | NLL ↓ | FID ↓ |
|---|---|---|---|
| Optimal Transport | **SB-FBSDE (ours)** | 2.96 | 3.01 |
| | DOT (Tanaka, 2019) | - | 15.78 |
| | Multi-stage SB (Wang et al., 2021) | - | 12.32 |
| | DGflow (Ansari et al., 2020) | - | 9.63 |
| SGMs | SDE (deep, sub-VP; Song et al. (2020)) | 2.99 | 2.92 |
| | ScoreFlow (Song et al., 2021) | 2.74 | 5.7 |
| | VDM (Kingma et al., 2021) | **2.49** | 4.00 |
| | LSGM(Vahdat et al., 2021) | 3.43 | **2.10** |

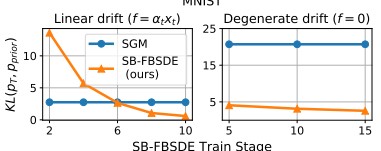

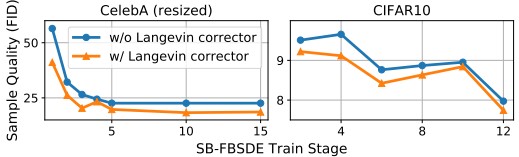

Figure 5: Validation of our SB-FBSDE on *learning* forward diffusions that are closer (in KL sense) to $p_{\text{prior}}$ compared to SGM.

Figure 6: Ablation analysis where we show that adding Langevin corrector to SB-FBSDE uniformly improves the FID scores on both CelebA and CIFAR10 training.

samples w.r.t the training set. Notably, our SB-FBSDE achieves 2.98 bits/dim and 3.18 FID score on CIFAR10, which is comparable to the top existing methods from other model classes (*e.g.* SGMs) and outperforms prior Optimal Transport (OT) methods (Wang et al., 2021; Tanaka, 2019) by a large margin in terms of the sample quality. More importantly, it enables log-likelihood computations that are otherwise infeasible in prior OT methods. We note that the quantitative comparisons on MNIST and CelebA are omitted as the scores on these two datasets are not widely reported and different pre-processing (*e.g.* resizing of CelebA) can lead to values that are not directly comparable.

**Validity of SB forward diffusion.** Our theoretical analysis in §3.2 suggests that the forward policy $\mathbf{Z}(\cdot,\cdot;\theta) \equiv \mathbf{Z}_\theta$ plays an essential role in governing samples towards $p_{\text{prior}}$. Here, we validate this conjecture by computing the KL divergence between the terminal distribution induced by $\mathbf{Z}_\theta$, *i.e.* $p_T^{(13a)}$, and the designated prior $p_{\text{prior}}$. We refer readers to Appendix D for the actual computation. Figure 5 reports these values over MNIST training. For both degenerate ($f := \mathbf{0}$) and linear ($f := \alpha_t \mathbf{X}_t$) base drifts, our SB-FBSDE generates terminal distributions that are much closer to $p_{\text{prior}}$. Note that the values of SGM remain unchanged throughout training since SGM relies on *pre-specified* diffusion. This is in contrast to our SB-FBSDE whose forward policy $\mathbf{Z}_\theta$ gradually shortens the KL gap to $p_{\text{prior}}$, thereby providing a better forward diffusion for training the backward policy.

**Effect of Langevin corrector.** In practice, we observe that the Langevin corrector greatly affects the generative performance. As shown in Fig. 6, including these corrector steps uniformly improves the sample quality (FID) on both CelebA and CIFAR10 throughout training. Since the SDEs are often solved via the Euler-Maruyama method, their propagation can be subjected to discretization errors accumulated over time. These Langevin steps thereby help re-distributing the samples at each time step $t$ towards the desired density $p_t^{\text{SB}}$. We emphasize this improvement as the benefit gained from applying modern generative training techniques based on the solid connection between SB and SGM.

## 5 CONCLUSION

In this work, we present a novel computational framework, grounded on Forward-Backward SDEs theory, for computing the log-likelihood of Schrödinger Bridge (SB) – a recently emerging model that adopts entropy-regularized optimal transport for generative modeling. Our findings provide new theoretical insights by generalizing previous theoretical results for Score-based Generative Model, and facilitate applications of modern generative training for SB. We validate our method on various image generative tasks, *e.g.* MNIST, CelebA, and CIFAR10, showing encouraging results in synthesizing high-fidelity samples while retaining the rigorous mathematical framework.

## ACKNOWLEDGMENTS

The authors would like to thank Ioannis Exarchos and Oswin So for their generous involvement and helpful supports during the rebuttal. The authors would also like to thank Marcus A Pereira and Ethan N Evans for their participation and kind discussion in the early stage of project exploration.

## AUTHOR CONTRIBUTIONS

The original idea of solving the PDE optimality of SB with FBSDEs theory was initiated by Tianrong. Later, Guan derived the main theories (*i.e.* Theorem 3, 4, 11, and Corollary 5) presented in Section 3.1, 3.2 and Appendix B with few helps from Tianrong. Tianrong designed the practical algorithms (*e.g.* stage-wise optimization and Langevin-corrector) in Section 3.3 and conducted most experiments with few helps from Guan. Guan wrote the main paper except for Section 4, which were written by both Tianrong and Guan. Both Guan and Tianrong contributed to code development.

## REPRODUCIBILITY STATEMENT

Our training algorithms are detailed in Alg. 1, 2, and 3, with the training objectives given in the same section (see (16, 18, 19)). Other implementation details (*e.g.* data pre-processing) are left in Appendix D. This shall provide sufficient information for readers of interests to reproduce our results. As we strongly believe in the merit of open sourcing, we intend to release our implementation upon publication. On the theoretical side, all proofs are left to Appendix B due to space constraint. We provide the assumptions in the same section.

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

## A  Introduction of Schrödinger Bridge

In this subsection, we provide a brief review for Schrödinger Bridge (SB) and some reasonings for Theorem 1. The SB problem, at its classical form, considers the following optimization (Dai Pra, 1991; Pavon & Wakolbinger, 1991),

$$
\min_{\mathbf{u} \in \mathcal{U}} \mathbb{E}\left[ \int_0^T \frac{1}{2} \|\mathbf{u}(t, \mathbf{X}_t)\|^2 \right]
$$

$$
\text{s.t. } \begin{cases} d\mathbf{X}_t = \mathbf{u}(t, \mathbf{X}_t)dt + \sqrt{2\epsilon}\, d\mathbf{W}_t \\ \mathbf{X}_0 \sim p_0(\mathbf{X}), \quad \mathbf{X}_T \sim p_T(\mathbf{X}) \end{cases}, \tag{22}
$$

where $\mathcal{U} := \{\mathbf{u} : [0, T] \times \mathbb{R}^n \mapsto \mathbb{R}^n | \langle \mathbf{u}, \mathbf{u} \rangle < \infty\}$. The optimization (22) characterizes a standard stochastic optimal control (SOC) programming with energy (*i.e.* $\frac{1}{2}\|\mathbf{u}\|^2$) minimization except with an additional terminal boundary condition. The optimality conditions to (22) are given by

$$
\begin{cases} \dfrac{\partial \psi}{\partial t} = -\dfrac{1}{2}\|\nabla_{\boldsymbol{x}} \psi\|^2 - \epsilon\, \Delta \psi, & \text{(23a)} \\[2mm] \dfrac{\partial p^*}{\partial t} = \nabla_{\boldsymbol{x}} \cdot (p^* \nabla_{\boldsymbol{x}} \psi) + \epsilon\, \Delta p^*, & \text{(23b)} \end{cases}
$$

where $\psi(t, \boldsymbol{x}) \in C^{1,2}$ is known as the *value* function in SOC literature and $p^*(t, \boldsymbol{x}) \in C^{1,2}$ is the associated optimal marginal density. $\Delta$ denotes the Laplace operator. Equations (23a) and (23b) are respectively the Kolmogorov's backward and forward PDEs, also known as Hamilton-Jacobi-Bellman and Fokker-Planck equations. The SB system can be obtained by applying the Hopf-Cole (Hopf, 1950; Cole, 1951) transformation $(\psi, p^*) \mapsto (\Psi, \widehat{\Psi})$,

$$
\begin{cases} \frac{\partial \Psi}{\partial t} = -\epsilon\, \Delta \Psi \\ \frac{\partial \widehat{\Psi}}{\partial t} = \epsilon\, \Delta \widehat{\Psi} \end{cases} \quad \text{s.t. } \Psi(0, \cdot)\widehat{\Psi}(0, \cdot) = p_0, \ \Psi(T, \cdot)\widehat{\Psi}(T, \cdot) = p_T. \tag{24}
$$

In this work, we consider a recent generalization of (22) and (24) to an SDE class with nonlinear drift, affine control, and time-varying diffusion. We synthesize their results below.

**Theorem 6** (SB optimality; Caluya & Halder (2021)). *Consider the following optimization*

$$
\min_{\mathbf{u}} \mathbb{E}\left[ \int_0^T \frac{1}{2} \|\mathbf{u}(t, \mathbf{X}_t)\|^2 \right]
$$

$$
\text{s.t. } \begin{cases} d\mathbf{X}_t = [f(t, \mathbf{X}_t) + g(t)\, \mathbf{u}(t, \mathbf{X}_t)]dt + \sqrt{2\epsilon}\, g(t)\, d\mathbf{W}_t \\ \mathbf{X}_0 \sim p_0(\mathbf{X}), \quad \mathbf{X}_T \sim p_T(\mathbf{X}) \end{cases}, \tag{25}
$$

*where $g(t)$ is uniformly lower-bounded and $f(t, \mathbf{X}_t)$ satisfies Lipschitz conditions with at most linear growth in $\boldsymbol{x}$. Then, the Hopf-Cole transformation to (25) becomes*

$$
\begin{cases} \frac{\partial \Psi}{\partial t} = -\nabla_{\boldsymbol{x}} \Psi^\mathsf{T} f - \epsilon\, \mathrm{Tr}(g^2 \nabla_{\boldsymbol{x}}^2 \Psi) \\ \frac{\partial \widehat{\Psi}}{\partial t} = -\nabla_{\boldsymbol{x}} \cdot (\widehat{\Psi} f) + \epsilon\, \mathrm{Tr}(g^2 \nabla_{\boldsymbol{x}}^2 \widehat{\Psi}) \end{cases}, \tag{26}
$$

*with the same boundary conditions $\Psi(0, \cdot)\widehat{\Psi}(0, \cdot) = p_0, \ \Psi(T, \cdot)\widehat{\Psi}(T, \cdot) = p_T$. The optimal control to (25) is thereby given by*

$$
\mathbf{u}^*(t, \mathbf{X}_t) = 2\epsilon g(t) \nabla_{\boldsymbol{x}} \log \Psi(t, \mathbf{X}_t). \tag{27}
$$

*Proof.* See Section III and Theorem 2 in f Caluya & Halder (2021). □

Theorem 6 is particularly attractive to us since its SDE corresponds exactly to the one appearing in score-based generative models. One can recover Theorem 1 by

*(i)* Following Pavon & Wakolbinger (1991), we know that the objective in (25) is equivalent to $D_{\mathrm{KL}}(\mathbb{Q} \,\|\, \mathbb{P})$ by an application of Girsanov's Theorem.

*(ii)* Equation (26) is exactly (6) with $\epsilon = \frac{1}{2}$. Furthermore, substituting the optimal control (27) to the stochastic process in (25) yields the optimal forward SDE in (7a).

*(iii)* Finally, reversing the SDE (7a) from forward to backward following Anderson (1982),

$$d\mathbf{X}_t = [f + g^2 \, \nabla_{\boldsymbol{x}} \log \Psi(t, \mathbf{X}_t) - g^2 \, \nabla_{\boldsymbol{x}} \log p_t^{\mathrm{SB}}] dt + g \, d\mathbf{W}_t, \tag{28}$$

and recalling the factorization principle, $\log p_t^{\mathrm{SB}}(\cdot) = \log \Psi(t, \cdot) + \log \widehat{\Psi}(t, \cdot)$, from Equation (4.15) in Chen et al. (2021) yield the optimal backward SDE in (7b).

## B  PROOFS AND REMARKS IN SECTION 3

In this section, we provide proofs for all of our theorems. We following the same notation by denoting $p_t^{\mathrm{SDE}}(\mathbf{X}_t)$ as the marginal density driven by some SDE process $\mathbf{X}(t) \equiv \mathbf{X}_t$ until the time step $t \in [0, T]$. Gradient and Hessian of a function $f(\boldsymbol{x})$, where $\boldsymbol{x} \in \mathbb{R}^n$, will respectively be denoted as $\nabla_{\boldsymbol{x}} f \in \mathbb{R}^n$ and $\nabla_{\boldsymbol{x}}^2 f \in \mathbb{R}^{n \times n}$. Divergence and Laplace operators will respectively be denoted as $\nabla \cdot$ and $\Delta$. Note that $\Delta = \nabla \cdot \nabla$. For notational brevity, we will only keep the subscript $\boldsymbol{x}$ for multivariate functions. Finally, $\mathrm{Tr}(\boldsymbol{A})$ denotes the trace of a square matrix $\boldsymbol{A}$.

We first restate the celebrated Itô lemma, which is known as the extension of the chain rule of ordinary calculus to the stochastic setting. It relies on the fact that $d\mathbf{W}_t^2$ and $dt$ are of the same scale and keeps the expansion up to $\mathcal{O}(dt)$.

**Lemma 7** (Itô formula; Itô (1951)). *Let $v \in C^{1,2}$ and let $\mathbf{X}_t$ be the stochastic process satisfying*

$$d\mathbf{X}_t = f(t, \mathbf{X}_t) dt + G(t, \mathbf{X}_t) d\mathbf{W}_t.$$

*Then, the stochastic process $v(t, \mathbf{X}_t)$ is also an Itô process satisfying*

$$\begin{aligned}
dv(t, \mathbf{X}_t) = \frac{\partial v(t, \mathbf{X}_t)}{\partial t} dt &+ \left[ \nabla_{\boldsymbol{x}} v(t, \mathbf{X}_t)^{\mathsf{T}} f + \frac{1}{2} \mathrm{Tr} \left[ G^{\mathsf{T}} \nabla_{\boldsymbol{x}}^2 v(t, \mathbf{X}_t) G \right] \right] dt \\
&+ \left[ \nabla_{\boldsymbol{x}} v(t, \mathbf{X}_t)^{\mathsf{T}} G(t, \mathbf{X}_t) \right] d\mathbf{W}_t.
\end{aligned} \tag{29}$$

Next, the following lemma will be useful in proving Theorem 3.

**Lemma 8.** *The following equality holds at any point $\boldsymbol{x} \in \mathbb{R}^n$ such that $p(\boldsymbol{x}) \neq 0$.*

$$\frac{1}{p(\boldsymbol{x})} \mathrm{Tr} \left( \nabla^2 p(\boldsymbol{x}) \right) = \| \nabla \log p(\boldsymbol{x}) \|^2 + \Delta \log p(\boldsymbol{x})$$

*Proof.*

$$\begin{aligned}
\mathrm{Tr} \left( \nabla^2 p(\boldsymbol{x}) \right) = \Delta p(\boldsymbol{x}) &= \nabla \cdot \nabla p(\boldsymbol{x}) \\
&= \nabla \cdot (p(\boldsymbol{x}) \nabla \log p(\boldsymbol{x})) \\
&= \nabla p(\boldsymbol{x})^{\mathsf{T}} \nabla \log p(\boldsymbol{x}) + p(\boldsymbol{x}) \Delta \log p(\boldsymbol{x}) \\
&= p(\boldsymbol{x}) \nabla \log p(\boldsymbol{x})^{\mathsf{T}} \nabla \log p(\boldsymbol{x}) + p(\boldsymbol{x}) \Delta \log p(\boldsymbol{x}) \\
&= p(\boldsymbol{x}) \left( \| \nabla \log p(\boldsymbol{x}) \|^2 + \Delta \log p(\boldsymbol{x}) \right)
\end{aligned}$$

$\square$

**Assumptions**  Before stating our proofs, we provide the assumptions used throughout the paper. These assumptions are adopted from stochastic analysis for SGM (Song et al., 2021; Yong & Zhou, 1999; Anderson, 1982), SB (Caluya & Halder, 2021), and FBSDE (Exarchos & Theodorou, 2018; Gorodetsky et al., 2015).

(i) $p_{\mathrm{prior}}, p_{\mathrm{data}} \in C^2$ with finite second-order moment.

(ii) $f$ and $g$ are continuous functions, and $|g(t)|^2 > 0$ is uniformly lower-bounded w.r.t. $t$.

(iii) $\forall t \in [0, T]$, we have $f(t, \boldsymbol{x}), \nabla_{\boldsymbol{x}} \log p_t(\boldsymbol{x}), \nabla_{\boldsymbol{x}} \log \Psi(t, \boldsymbol{x}), \nabla_{\boldsymbol{x}} \log \widehat{\Psi}(t, \boldsymbol{x}), \mathbf{Z}(t, \boldsymbol{x}; \theta)$, and $\widehat{\mathbf{Z}}(t, \boldsymbol{x}; \phi)$ Lipschitz and at most linear growth w.r.t. $\boldsymbol{x}$.

(iv) $\Psi, \widehat{\Psi} \in C^{1,2}$. $h$, and $\varphi$ are continuous functions. $h$ satisfies quadratic growth w.r.t. $\boldsymbol{x}$ uniformly in $t$.

(v) $\exists k > 0 : p_t^{\mathrm{SB}}(\boldsymbol{x}) = \mathcal{O}(\exp^{-\|\boldsymbol{x}\|_k^2})$ as $\boldsymbol{x} \to \infty$.

Assumptions (i) (ii) (iii) are standard conditions in stochastic analysis to ensure the existence-uniqueness of the SDEs; hence also appear in SGM analysis (Song et al., 2021). Assumption (iv) allows applications of Itô formula and properly defines the backward SDE in FBSDE theory. Finally, assumption (v) assures the exponential limiting behavior when performing integration by parts.

Now, let us begin the proofs of Theorem 3, 4, and Corollary 5.

**Theorem 3** (FBSDEs to SB optimality (6)). *Consider the following set of coupled SDEs,*

$$
\begin{cases}
\mathrm{d}\mathbf{X}_t = (f + g\mathbf{Z}_t)\,\mathrm{d}t + g\mathrm{d}\mathbf{W}_t & \text{(30a)} \\[2mm]
\mathrm{d}\mathbf{Y}_t = \dfrac{1}{2}\mathbf{Z}_t^{\mathsf{T}}\mathbf{Z}_t\mathrm{d}t + \mathbf{Z}_t^{\mathsf{T}}\mathrm{d}\mathbf{W}_t & \text{(30b)} \\[2mm]
\mathrm{d}\widehat{\mathbf{Y}}_t = \left(\dfrac{1}{2}\widehat{\mathbf{Z}}_t^{\mathsf{T}}\widehat{\mathbf{Z}}_t + \nabla_{\boldsymbol{x}} \cdot (g\widehat{\mathbf{Z}}_t - f) + \widehat{\mathbf{Z}}_t^{\mathsf{T}}\mathbf{Z}_t\right)\mathrm{d}t + \widehat{\mathbf{Z}}_t^{\mathsf{T}}\mathrm{d}\mathbf{W}_t & \text{(30c)}
\end{cases}
$$

*where $f$ and $g$ satisfy the same regularity conditions in Lemma 2 (see Footnote 4), and the boundary conditions are given by $\mathbf{X}(0) = \boldsymbol{x}_0$ and $\mathbf{Y}_T + \widehat{\mathbf{Y}}_T = \log p_{\mathrm{prior}}(\mathbf{X}_T)$. Suppose $\Psi, \widehat{\Psi} \in C^{1,2}$, then nonlinear Feynman-Kac relations between the FBSDEs (13) and PDEs (6) are given by*

$$
\begin{aligned}
\mathbf{Y}_t &\equiv \mathbf{Y}(t, \mathbf{X}_t) = \log \Psi(t, \mathbf{X}_t), & \mathbf{Z}_t &\equiv \mathbf{Z}(t, \mathbf{X}_t) = g\nabla_{\boldsymbol{x}} \log \Psi(t, \mathbf{X}_t), \\
\widehat{\mathbf{Y}}_t &\equiv \widehat{\mathbf{Y}}(t, \mathbf{X}_t) = \log \widehat{\Psi}(t, \mathbf{X}_t), & \widehat{\mathbf{Z}}_t &\equiv \widehat{\mathbf{Z}}(t, \mathbf{X}_t) = g\nabla_{\boldsymbol{x}} \log \widehat{\Psi}(t, \mathbf{X}_t).
\end{aligned} \tag{31}
$$

*Furthermore, $(\mathbf{Y}_t, \widehat{\mathbf{Y}}_t)$ obey the following relation:*

$$
\mathbf{Y}_t + \widehat{\mathbf{Y}}_t = \log p_t^{\mathrm{SB}}(\mathbf{X}_t).
$$

*Proof.* Similar to how the original nonlinear Feynman-Kac (*i.e.* Lemma 2) can be carried out by an application of Itô lemma (Ma et al., 1999). We can apply Itô lemma 7 to the stochastic process $\log \Psi(t, \mathbf{X}_t)$ w.r.t. the optimal forward SDE (7a).

$$
\mathrm{d}\log \Psi = \frac{\partial \log \Psi}{\partial t}\mathrm{d}t + \left[\nabla_{\boldsymbol{x}} \log \Psi^{\mathsf{T}}(f + g^2\nabla_{\boldsymbol{x}} \log \Psi) + \frac{1}{2}g^2 \operatorname{Tr}\left[\nabla_{\boldsymbol{x}}^2 \log \Psi\right]\right]\mathrm{d}t + \left[g\nabla_{\boldsymbol{x}} \log \Psi^{\mathsf{T}}\right]\mathrm{d}\mathbf{W}_t. \tag{32}
$$

From the PDE dynamics (6), we know that

$$
\begin{aligned}
\frac{\partial \log \Psi}{\partial t} &= \frac{1}{\Psi}\left(-\nabla_{\boldsymbol{x}}\Psi^{\mathsf{T}}f - \frac{1}{2}\operatorname{Tr}(g^2\nabla_{\boldsymbol{x}}^2\Psi)\right) \\
&= -\nabla_{\boldsymbol{x}} \log \Psi^{\mathsf{T}}f - \frac{1}{2}g^2 \operatorname{Tr}(\frac{1}{\Psi}\nabla_{\boldsymbol{x}}^2\Psi).
\end{aligned}
$$

The first term in the RHS can be readily canceled out with the related $f$-term in (32). The second term can also be canceled out using the fact that $\nabla_{\boldsymbol{x}}^2 \log \Psi = \frac{1}{\Psi}\nabla_{\boldsymbol{x}}^2\Psi - \frac{1}{\Psi^2}\nabla_{\boldsymbol{x}}\Psi\nabla_{\boldsymbol{x}}\Psi^{\mathsf{T}}$. Hence, we are left with

$$
\begin{aligned}
\mathrm{d}\log \Psi &= \left[\|g\nabla_{\boldsymbol{x}} \log \Psi\|^2 - \frac{1}{2}g^2 \operatorname{Tr}\left[\frac{1}{\Psi^2}\nabla_{\boldsymbol{x}}\Psi\nabla_{\boldsymbol{x}}\Psi^{\mathsf{T}}\right]\right]\mathrm{d}t + g\nabla_{\boldsymbol{x}} \log \Psi^{\mathsf{T}}\mathrm{d}\mathbf{W}_t \\
&= \frac{1}{2}\|g\nabla_{\boldsymbol{x}} \log \Psi\|^2\mathrm{d}t + g\nabla_{\boldsymbol{x}} \log \Psi^{\mathsf{T}}\mathrm{d}\mathbf{W}_t. \tag{33}
\end{aligned}
$$

Likewise, applying Itô lemma to $\log \widehat{\Psi}(t, \mathbf{X}_t)$, where $\mathbf{X}_t$ follows the SDE in (7a),

$$
\mathrm{d}\log \widehat{\Psi} = \frac{\partial \log \widehat{\Psi}}{\partial t}\mathrm{d}t + \left[\nabla_{\boldsymbol{x}} \log \widehat{\Psi}^{\mathsf{T}}(f + g^2\nabla_{\boldsymbol{x}} \log \Psi) + \frac{1}{2}g^2 \operatorname{Tr}\left[\nabla_{\boldsymbol{x}}^2 \log \widehat{\Psi}\right]\right]\mathrm{d}t + \left[g\nabla_{\boldsymbol{x}} \log \widehat{\Psi}^{\mathsf{T}}\right]\mathrm{d}\mathbf{W}_t, \tag{34}
$$

but now noticing that the dynamics of $\frac{\partial \log \widehat{\Psi}}{\partial t}$ become

$$\frac{\partial \log \widehat{\Psi}}{\partial t} = \frac{1}{\widehat{\Psi}}\left(-\nabla_{\boldsymbol{x}} \cdot (\widehat{\Psi}f) + \frac{1}{2}\operatorname{Tr}(g^2 \nabla_{\boldsymbol{x}}^2 \widehat{\Psi})\right)$$

$$= -\nabla_{\boldsymbol{x}} \log \widehat{\Psi}^{\mathsf{T}} f - \nabla_{\boldsymbol{x}} \cdot f + \frac{1}{2}g^2 \operatorname{Tr}(\frac{1}{\widehat{\Psi}}\nabla_{\boldsymbol{x}}^2 \widehat{\Psi}).$$

Only the first term in the RHS will be canceled out in (34). Hence, we are left with

$$\mathrm{d}\log \widehat{\Psi} = \left[-\nabla_{\boldsymbol{x}} \cdot f + \frac{1}{2}g^2 \operatorname{Tr}\left[\frac{1}{\widehat{\Psi}}\nabla_{\boldsymbol{x}}^2 \widehat{\Psi}\right]\right] \mathrm{d}t$$

$$+ \left[g^2 \nabla_{\boldsymbol{x}} \log \widehat{\Psi}^{\mathsf{T}} \nabla_{\boldsymbol{x}} \log \Psi + \frac{1}{2}g^2 \operatorname{Tr}\left[\nabla_{\boldsymbol{x}}^2 \log \widehat{\Psi}\right]\right] \mathrm{d}t + g\nabla_{\boldsymbol{x}} \log \widehat{\Psi}^{\mathsf{T}} \mathrm{d}\mathbf{W}_t. \tag{35}$$

Notice that the trace terms above can be simplified to

$$\frac{1}{2}\operatorname{Tr}\left[\frac{1}{\widehat{\Psi}}\nabla_{\boldsymbol{x}}^2 \widehat{\Psi} + \nabla_{\boldsymbol{x}}^2 \log \widehat{\Psi}\right] = \operatorname{Tr}\left[\frac{1}{\widehat{\Psi}}\nabla_{\boldsymbol{x}}^2 \widehat{\Psi}\right] - \frac{1}{2}\|\nabla_{\boldsymbol{x}} \log \widehat{\Psi}\|^2$$

$$= \frac{1}{2}\|\nabla_{\boldsymbol{x}} \log \widehat{\Psi}\|^2 + \Delta_{\boldsymbol{x}} \log \widehat{\Psi},$$

where the last equality follows by Lemma 8. Substituting this result back to (35), we get

$$\mathrm{d}\log \widehat{\Psi} = \left[-\nabla_{\boldsymbol{x}} \cdot f + \frac{1}{2}\|g\nabla_{\boldsymbol{x}} \log \widehat{\Psi}\|^2 + g^2 \Delta_{\boldsymbol{x}} \log \widehat{\Psi} + g^2 \nabla_{\boldsymbol{x}} \log \widehat{\Psi}^{\mathsf{T}} \nabla_{\boldsymbol{x}} \log \Psi\right] \mathrm{d}t + g\nabla_{\boldsymbol{x}} \log \widehat{\Psi}^{\mathsf{T}} \mathrm{d}\mathbf{W}_t$$

$$= \left[\nabla_{\boldsymbol{x}} \cdot \left(g^2 \nabla_{\boldsymbol{x}} \log \widehat{\Psi} - f\right) + \frac{1}{2}\|g\nabla_{\boldsymbol{x}} \log \widehat{\Psi}\|^2 + g^2 \nabla_{\boldsymbol{x}} \log \widehat{\Psi}^{\mathsf{T}} \nabla_{\boldsymbol{x}} \log \Psi\right] \mathrm{d}t + g\nabla_{\boldsymbol{x}} \log \widehat{\Psi}^{\mathsf{T}} \mathrm{d}\mathbf{W}_t \tag{36}$$

Finally, by rewriting (33) and (36) with the nonlinear Feynman-Kac in (31) yields

$$\boxed{\begin{aligned} \mathrm{d}\mathbf{X}_t &= (f + g\mathbf{Z}_t)\,\mathrm{d}t + g\mathrm{d}\mathbf{W}_t \\ \mathrm{d}\mathbf{Y}_t &= \frac{1}{2}\mathbf{Z}_t^{\mathsf{T}}\mathbf{Z}_t\mathrm{d}t + \mathbf{Z}_t^{\mathsf{T}}\mathrm{d}\mathbf{W}_t \\ \mathrm{d}\widehat{\mathbf{Y}}_t &= \left(\frac{1}{2}\widehat{\mathbf{Z}}_t^{\mathsf{T}}\widehat{\mathbf{Z}}_t + \nabla_{\boldsymbol{x}} \cdot (g\widehat{\mathbf{Z}}_t - f) + \widehat{\mathbf{Z}}_t^{\mathsf{T}}\mathbf{Z}_t\right)\mathrm{d}t + \widehat{\mathbf{Z}}_t^{\mathsf{T}}\mathrm{d}\mathbf{W}_t \end{aligned}}$$

This concludes the proof. $\qquad\square$

**Remark 9** (Viscosity solutions). These FBSDE results can be extended to viscosity solutions in the case when the classical solution does not exist (Pardoux & Peng, 1992). For the completeness, one shall understand them in the sense of $v(t, \boldsymbol{x}) = \lim_{\epsilon \to \infty} v_\epsilon(t, \boldsymbol{x})$ uniformly in $(t, \boldsymbol{x})$ over a compact set. Here $v^\epsilon(t, \boldsymbol{x})$ is the classical solution to (10) with $(f_\epsilon, G_\epsilon, h_\epsilon, \varphi_\epsilon)$ converge uniformly toward $(f, G, h, \varphi)$ over the compact set. We refer readers of interests to Exarchos & Theodorou (2018); Negyesi et al. (2021), and their references therein.

**Theorem 4** (Log-likelihood for SB models). *Given the solution satisfying the FBSDE system in (13), the log-likelihood of the SB model $(\mathbf{Z}_t, \widehat{\mathbf{Z}}_t)$, at a data point $\boldsymbol{x}_0$, can be expressed as*

$$\mathcal{L}_{\mathrm{SB}}(\boldsymbol{x}_0) = \mathbb{E}\left[\log p_T(\mathbf{X}_T)\right] - \int_0^T \mathbb{E}\left[\frac{1}{2}\|\mathbf{Z}_t\|^2 + \frac{1}{2}\|\widehat{\mathbf{Z}}_t - g\nabla_{\boldsymbol{x}} \log p_t^{\mathrm{SB}} + \mathbf{Z}_t\|^2\right.$$

$$\left.-\frac{1}{2}\|g\nabla_{\boldsymbol{x}} \log p_t^{\mathrm{SB}} - \mathbf{Z}_t\|^2 - \nabla_{\boldsymbol{x}} \cdot f\right]\mathrm{d}t \tag{38}$$

$$= \mathbb{E}\left[\log p_T(\mathbf{X}_T)\right] - \int_0^T \mathbb{E}\left[\frac{1}{2}\|\mathbf{Z}_t\|^2 + \frac{1}{2}\|\widehat{\mathbf{Z}}_t\|^2 + \nabla_{\boldsymbol{x}} \cdot (g\widehat{\mathbf{Z}}_t - f) + \widehat{\mathbf{Z}}_t^{\mathsf{T}}\mathbf{Z}_t\right]\mathrm{d}t, \tag{39}$$

*where the expectation is taken over the forward SDE (13a) with the initial condition $\mathbf{X}_0 = \boldsymbol{x}_0$.*

*Proof.*

$$\mathcal{L}_{\text{SB}}(\boldsymbol{x}_0)$$

$$=\mathbb{E}\left[\mathbf{Y}_0 + \widehat{\mathbf{Y}}_0 | \mathbf{X}_0 = \boldsymbol{x}_0\right]$$

$$=\mathbb{E}\left[\mathbf{Y}_T - \int_0^T \left(\frac{1}{2}\|\mathbf{Z}_t\|^2\right) \mathrm{d}t + \widehat{\mathbf{Y}}_T - \int_0^T \left(\frac{1}{2}\|\widehat{\mathbf{Z}}_t\|^2 + \nabla \cdot (g\widehat{\mathbf{Z}}_t - f) + \widehat{\mathbf{Z}}_t^\mathsf{T}\mathbf{Z}_t\right) \mathrm{d}t \Big| \mathbf{X}_0 = \boldsymbol{x}_0\right]$$

$$=\mathbb{E}\left[\mathbf{Y}_T + \widehat{\mathbf{Y}}_T | \mathbf{X}_0 = \boldsymbol{x}_0\right] - \int_0^T \mathbb{E}\left[\frac{1}{2}\|\mathbf{Z}_t\|^2 + \frac{1}{2}\|\widehat{\mathbf{Z}}_t\|^2 + \nabla \cdot (g\widehat{\mathbf{Z}}_t - f) + \widehat{\mathbf{Z}}_t^\mathsf{T}\mathbf{Z}_t \Big| \mathbf{X}_0 = \boldsymbol{x}_0\right] \mathrm{d}t$$

$$=\mathbb{E}[\log p_T(\mathbf{X}_T)] - \int_0^T \mathbb{E}\left[\frac{1}{2}\|\mathbf{Z}_t\|^2 + \frac{1}{2}\|\widehat{\mathbf{Z}}_t\|^2 + \nabla \cdot (g\widehat{\mathbf{Z}}_t - f) + \widehat{\mathbf{Z}}_t^\mathsf{T}\mathbf{Z}_t\right] \mathrm{d}t, \tag{40}$$

which recovers (39). Finally, notice that with integration by part, we have

$$\mathbb{E}_{\mathbf{X}_t \sim p_t^{\text{SB}}}\left[g\nabla \cdot \widehat{\mathbf{Z}}_t\right] = \int \left(g\nabla \cdot \widehat{\mathbf{Z}}_t\right) p_t^{\text{SB}} \, \mathrm{d}\mathbf{X}_t$$

$$= -\int g\widehat{\mathbf{Z}}_t^\mathsf{T}\nabla_{\boldsymbol{x}} p_t^{\text{SB}} \, \mathrm{d}\mathbf{X}_t$$

$$= -\int \left(g\widehat{\mathbf{Z}}_t^\mathsf{T}\nabla_{\boldsymbol{x}} \log p_t^{\text{SB}}\right) p_t^{\text{SB}} \, \mathrm{d}\mathbf{X}_t \tag{41}$$

$$= \mathbb{E}_{\mathbf{X}_t \sim p_t^{\text{SB}}}\left[-g\widehat{\mathbf{Z}}_t^\mathsf{T}\nabla_{\boldsymbol{x}} \log p_t^{\text{SB}}\right],$$

where we adopt common practice and assume the limiting behavior of $p_t^{\text{SB}}$; in other words, $\exists k > 0 : p_t^{\text{SB}}(\boldsymbol{x}) = \mathcal{O}(\exp^{-\|\boldsymbol{x}\|_k^2})$ as $\boldsymbol{x} \to \infty$. With (41), we can rewrite the related parts in (40) as

$$\mathbb{E}\left[\frac{1}{2}\|\widehat{\mathbf{Z}}_t\|^2 + g\nabla \cdot \widehat{\mathbf{Z}}_t + \widehat{\mathbf{Z}}_t^\mathsf{T}\mathbf{Z}_t\right]$$

$$=\mathbb{E}\left[\frac{1}{2}\|\widehat{\mathbf{Z}}_t\|^2 - \widehat{\mathbf{Z}}_t^\mathsf{T}\left(g\nabla \log p_t^{\text{SB}}\right) + \widehat{\mathbf{Z}}_t^\mathsf{T}\mathbf{Z}_t\right]$$

$$=\mathbb{E}\left[\frac{1}{2}\|\widehat{\mathbf{Z}}_t - g\nabla \log p_t^{\text{SB}} + \mathbf{Z}_t\|^2 - \frac{1}{2}\|g\nabla \log p_t^{\text{SB}} - \mathbf{Z}_t\|^2\right]. \tag{42}$$

Hence, we also recover (38). $\qquad\square$

**Corollary 5** (Probability flow for SB). *The following ODE characterizes the probability flow of the optimal processes of SB (7) in the sense that $\forall t$, $p_t^{(17)} \equiv p_t^{(7)} \equiv p_t^{\text{SB}}$.*

$$\mathrm{d}\mathbf{X}_t = \left[f + g\mathbf{Z}(t, \mathbf{X}_t) - \frac{1}{2}g\left(\mathbf{Z}(t, \mathbf{X}_t) + \widehat{\mathbf{Z}}(t, \mathbf{X}_t)\right)\right] \mathrm{d}t \tag{43}$$

*Proof.* The probability ODE flow (Song et al., 2020; Maoutsa et al., 2020) suggests that the equivalent ODE model for the SDE (1) is given by

$$\mathrm{d}\mathbf{X}_t = \left[f - \frac{1}{2}g^2\nabla_{\boldsymbol{x}} \log p_t^{(1)}\right] \mathrm{d}t.$$

We can adopt this result to the SDEs of SB (7a) by considering $f \leftarrow f + g\mathbf{Z}_t$ and $p_t^{(1)} \leftarrow p_t^{\text{SB}}$. This yields

$$\mathrm{d}\mathbf{X}_t = \left[f + g\mathbf{Z}_t - \frac{1}{2}g^2\nabla_{\boldsymbol{x}} \log p_t^{\text{SB}}\right] \mathrm{d}t. \tag{44}$$

Applying the the factorization principle (Chen et al., 2021) with $g\log p_t^{\text{SB}} = \mathbf{Z}_t + \widehat{\mathbf{Z}}_t$ concludes the proof. $\qquad\square$

**Remark 10** (Connection between SB-FBSDE and flow-based models). To demonstrate how applying flow-based training techniques to the probability ODE flow of SB (43) recovers the same log-likelihood objective in (39), recall that given an ODE $d\mathbf{X}_t = F(t, \mathbf{X}_t)dt$ with $\mathbf{X}_0 = \boldsymbol{x}_0 \sim p_{\text{data}}$, flow-based models compute the change in log-density using the instantaneous change of variables formula (Chen et al., 2018):

$$\frac{\partial \log p(\mathbf{X}_t)}{\partial t} = -\nabla_{\boldsymbol{x}} \cdot F,$$

which implies that the log-likelihood of $\boldsymbol{x}_0$ can be computed as

$$\log p(\mathbf{X}_T) = \log p(\boldsymbol{x}_0) - \int_0^T \nabla_{\boldsymbol{x}} \cdot F \, dt. \tag{45}$$

Now, consider the probability ODE flow of SB in (44),

$$F_{\text{SB}} := f + g\mathbf{Z}_t - \frac{1}{2}g(\mathbf{Z}_t + \widehat{\mathbf{Z}}_t) = f + \frac{1}{2}g(\mathbf{Z}_t - \widehat{\mathbf{Z}}_t).$$

Substituting this vector field $F_{\text{SB}}$ to (45) yields

$$\log p_T(\mathbf{X}_T) = \log p_0(\boldsymbol{x}_0) - \int_0^T \nabla_{\boldsymbol{x}} \cdot \left( f + \frac{1}{2}g(\mathbf{Z}_t - \widehat{\mathbf{Z}}_t) \right) dt$$

$$\Rightarrow \mathbb{E}\left[ \log p_0(\boldsymbol{x}_0) \right] = \mathbb{E}\left[ \log p_T(\mathbf{X}_T) \right] + \int_0^T \mathbb{E}\left[ \nabla_{\boldsymbol{x}} \cdot \left( f + \frac{1}{2}g(\mathbf{Z}_t - \widehat{\mathbf{Z}}_t) \right) \right] dt$$

$$= \mathbb{E}\left[ \log p_T(\mathbf{X}_T) \right] - \int_0^T \mathbb{E}\left[ \nabla_{\boldsymbol{x}} \cdot (g\widehat{\mathbf{Z}}_t - f) - \frac{1}{2}g\nabla_{\boldsymbol{x}} \cdot (\mathbf{Z}_t + \widehat{\mathbf{Z}}_t) \right] dt$$

$$\overset{(*)}{=} \mathbb{E}\left[ \log p_T(\mathbf{X}_T) \right] - \int_0^T \mathbb{E}\left[ \nabla_{\boldsymbol{x}} \cdot (g\widehat{\mathbf{Z}}_t - f) + \frac{1}{2}g(\mathbf{Z}_t + \widehat{\mathbf{Z}}_t)^{\mathsf{T}} \left( \nabla_{\boldsymbol{x}} \log p_t^{\text{SB}} \right) \right] dt$$

$$\overset{(**)}{=} \mathbb{E}\left[ \log p_T(\mathbf{X}_T) \right] - \int_0^T \mathbb{E}\left[ \nabla_{\boldsymbol{x}} \cdot (g\widehat{\mathbf{Z}}_t - f) + \frac{1}{2}(\mathbf{Z}_t + \widehat{\mathbf{Z}}_t)^2 \right] dt, \tag{46}$$

where (*) is due to integration by parts (recall (41)) and (**) again uses the factorization principle $\mathbf{Z}_t + \widehat{\mathbf{Z}}_t = g\nabla_{\boldsymbol{x}} \log p_t^{\text{SB}}$. One can verify that (46) indeed recovers (39).

**Theorem 11** (FBSDE computation for $\mathcal{L}_{\text{SB}}(\boldsymbol{x}_T)$ in SB models). *With the same regularity conditions in Theorem 3, the following FBSDEs also satisfy the nonlinear Feynman-Kac relations in (31).*

$$\begin{cases} d\mathbf{X}_t = \left( f - g\widehat{\mathbf{Z}}_t \right) dt + g d\mathbf{W}_t & \text{(47a)} \\[2mm] d\mathbf{Y}_t = -\left( \frac{1}{2}\mathbf{Z}_t^{\mathsf{T}}\mathbf{Z}_t + \nabla_{\boldsymbol{x}} \cdot (g\mathbf{Z}_t + f) + \mathbf{Z}_t^{\mathsf{T}}\widehat{\mathbf{Z}}_t \right) dt + \mathbf{Z}_t^{\mathsf{T}} d\mathbf{W}_t & \text{(47b)} \\[2mm] d\widehat{\mathbf{Y}}_t = -\frac{1}{2}\widehat{\mathbf{Z}}_t^{\mathsf{T}}\widehat{\mathbf{Z}}_t dt + \widehat{\mathbf{Z}}_t^{\mathsf{T}} d\mathbf{W}_t & \text{(47c)} \end{cases}$$

*Given a backward trajectory sampled from (47a), where $\mathbf{X}_T = \boldsymbol{x}_T$ and $\boldsymbol{x}_T \sim p_{\text{prior}}$, the log-likelihood of $\boldsymbol{x}_T$ is given by $\log p_{\text{prior}}(\boldsymbol{x}_T) = \mathbb{E}\left[ \mathbf{Y}_T + \widehat{\mathbf{Y}}_T | \mathbf{X}_T = \boldsymbol{x}_T \right] := \mathcal{L}_{\text{SB}}(\boldsymbol{x}_T)$. In particular,*

$$\mathcal{L}_{\text{SB}}(\boldsymbol{x}_T) = \mathbb{E}\left[ \log p_T(\mathbf{X}_0) \right] - \int_0^T \mathbb{E}\left[ \frac{1}{2}\|\widehat{\mathbf{Z}}_t\|^2 + \frac{1}{2}\|\mathbf{Z}_t\|^2 + \nabla_{\boldsymbol{x}} \cdot (g\mathbf{Z}_t + f) + \mathbf{Z}_t^{\mathsf{T}}\widehat{\mathbf{Z}}_t \right] dt, \tag{48}$$

*Proof.* Due to the symmetric structure of SB, we can consider a new time coordinate

$$s \triangleq T - t.$$

Under this transformation, the base reference $\mathbb{P}$ appearing in (5) is equivalent to

$$d\mathbf{X}_s = -f(s, \mathbf{X}_s)ds + g \, d\mathbf{W}_s.$$

The corresponding PDE optimality becomes

$$\begin{cases} \frac{\partial \Phi}{\partial s} = \nabla_{\boldsymbol{x}}\Phi^{\mathsf{T}} f - \frac{1}{2} \operatorname{Tr}(g^2 \nabla_{\boldsymbol{x}}^2 \Phi) \\[2mm] \frac{\partial \widehat{\Phi}}{\partial s} = \nabla_{\boldsymbol{x}} \cdot (\widehat{\Phi} f) + \frac{1}{2} \operatorname{Tr}(g^2 \nabla_{\boldsymbol{x}}^2 \widehat{\Phi}) \end{cases} \quad \text{s.t. } \Phi(0, \cdot)\widehat{\Phi}(0, \cdot) = p_{\text{prior}}, \ \Phi(T, \cdot)\widehat{\Phi}(T, \cdot) = p_{\text{data}}, \tag{49}$$

and the optimal forward/backward policies are given by

$$d\mathbf{X}_s = [-f + g^2 \, \nabla_{\boldsymbol{x}} \log \Phi(s, \mathbf{X}_s)]ds + g \, d\mathbf{W}_s, \quad \mathbf{X}_0 \sim p_{\text{prior}}, \tag{50a}$$

$$d\mathbf{X}_s = [-f - g^2 \, \nabla_{\boldsymbol{x}} \log \widehat{\Phi}(s, \mathbf{X}_s)]ds + g \, d\mathbf{W}_s, \quad \mathbf{X}_T \sim p_{\text{data}}. \tag{50b}$$

By comparing (50) with (7), one can notice that the new SB system $(\Phi, \widehat{\Phi})_s$ corresponds to the original system $(\Psi, \widehat{\Psi})_t$ via

$$\Phi(s, \mathbf{X}_s) = \widehat{\Psi}(T - t, \mathbf{X}_{T-t}) \qquad \text{and} \qquad \widehat{\Phi}(s, \mathbf{X}_s) = \Psi(T - t, \mathbf{X}_{T-t}). \tag{51}$$

Equation (51) shall be understood as the forward policy in $t$-coordinate system corresponds to the backward policy in $s$-coordinate system, and vise versa. Following similar derivations in the proof of Theorem 3, we can apply Itô lemma to expend the stochastic processes $d \log \Phi$ and $d \log \widehat{\Phi}$ w.r.t. (50a). This yields the following FBSDE system.

$$d\mathbf{X}_s = (g\mathbf{Z}'_s - f) \, ds + g d\mathbf{W}_s \tag{52a}$$

$$d\mathbf{Y}'_s = \frac{1}{2}\|\mathbf{Z}'_s\|^2 ds + {\mathbf{Z}'_s}^\mathsf{T} d\mathbf{W}_s \tag{52b}$$

$$d\widehat{\mathbf{Y}}'_s = \left( \frac{1}{2}\|\widehat{\mathbf{Z}}'_s\|^2 + \nabla_{\boldsymbol{x}} \cdot (g\widehat{\mathbf{Z}}'_s + f) + {\widehat{\mathbf{Z}}'_s}^\mathsf{T} \mathbf{Z}'_s \right) ds + {\widehat{\mathbf{Z}}'_s}^\mathsf{T} d\mathbf{W}_s \tag{52c}$$

Similar to (51), $(\mathbf{Y}'_s, \widehat{\mathbf{Y}}'_s, \mathbf{Z}'_s, \widehat{\mathbf{Z}}'_s)$ relate to the original FBSDE system (30) by

$$(\mathbf{Y}'_s, \widehat{\mathbf{Y}}'_s, \mathbf{Z}'_s, \widehat{\mathbf{Z}}'_s) = (\widehat{\mathbf{Y}}'_{T-t}, \mathbf{Y}'_{T-t}, \widehat{\mathbf{Z}}'_{T-t}, \mathbf{Z}'_{T-t}). \tag{53}$$

Changing the coordinate from $s$ to $t$ and applying (53) readily yield (47). Finally, the expression in (48) can be carried out similar to (40):

$$\mathcal{L}_{\text{SB}}(\boldsymbol{x}_T)$$

$$=\mathbb{E}\left[ \mathbf{Y}_T + \widehat{\mathbf{Y}}_T | \mathbf{X}_T = \boldsymbol{x}_T \right]$$

$$=\mathbb{E}\left[ \mathbf{Y}_0 - \int_0^T \left( \frac{1}{2}\|\widehat{\mathbf{Z}}_t\|^2 \right) dt + \widehat{\mathbf{Y}}_0 - \int_0^T \left( \frac{1}{2}\|\mathbf{Z}_t\|^2 + \nabla \cdot (g\mathbf{Z}_t + f) + \mathbf{Z}_t^\mathsf{T}\widehat{\mathbf{Z}}_t \right) dt \Big| \mathbf{X}_T = \boldsymbol{x}_T \right]$$

$$=\mathbb{E}\left[ \mathbf{Y}_0 + \widehat{\mathbf{Y}}_0 | \mathbf{X}_T = \boldsymbol{x}_T \right] - \int_0^T \mathbb{E}\left[ \frac{1}{2}\|\widehat{\mathbf{Z}}_t\|^2 + \frac{1}{2}\|\mathbf{Z}_t\|^2 + \nabla \cdot (g\mathbf{Z}_t + f) + \mathbf{Z}_t^\mathsf{T}\widehat{\mathbf{Z}}_t \Big| \mathbf{X}_T = \boldsymbol{x}_T \right] dt$$

$$=\mathbb{E}[\log p_0(\mathbf{X}_0)] - \int_0^T \mathbb{E}\left[ \frac{1}{2}\|\widehat{\mathbf{Z}}_t\|^2 + \frac{1}{2}\|\mathbf{Z}_t\|^2 + \nabla \cdot (g\mathbf{Z}_t + f) + \mathbf{Z}_t^\mathsf{T}\widehat{\mathbf{Z}}_t \right] dt. \tag{54}$$

We conclude the proof. $\qquad\square$

## C  COMPARISON WITH PRIOR SB WORKS

Our method is closely related to two concurrent SB models (De Bortoli et al., 2021; Vargas et al., 2021), yet differs in various aspects. Below we enumerate some of the differences.

**Training loss.** Both concurrent methods rely on solving SB *mean-matching* regression between the current drift and (estimated) optimal drift. This is in contrast to our SB-FBSDE, which instead utilizes a *divergence-based* objective (16). However, the regression objectives are in fact captured by Theorem 4. To see that, recall the forward and backward transition models considered in De Bortoli et al. (2021),

$$\mathbf{X}_{k+1} \sim \mathcal{N}(F_k(\mathbf{X}_k), 2\gamma_{k+1}\boldsymbol{I}), \quad \text{and} \quad \mathbf{X}_k \sim \mathcal{N}(B_{k+1}(\mathbf{X}_{k+1}), 2\gamma_{k+1}\boldsymbol{I}),$$

where $F_k(\boldsymbol{x}) := \boldsymbol{x} + \gamma_{k+1}f_k(\boldsymbol{x})$ and $B_{k+1}(\boldsymbol{x}) := \boldsymbol{x} + \gamma_{k+1}b_{k+1}(\boldsymbol{x})$ are solved alternately via

$$B_{k+1} \leftarrow \arg\min_{B_{k+1}} \mathbb{E}\left[ \|B_{k+1}(\mathbf{X}_{k+1}) - (\mathbf{X}_{k+1} + F_k(\mathbf{X}_k) - F_k(\mathbf{X}_{k+1}))\|^2 \right] \tag{55a}$$

$$F_k \leftarrow \arg\min_{F_k} \mathbb{E}\left[ \|F_k(\mathbf{X}_{k+1}) - (\mathbf{X}_k + B_{k+1}(\mathbf{X}_k) - B_{k+1}(\mathbf{X}_k))\|^2 \right]. \tag{55b}$$

In what follows, we focus mainly on the connection between (55a) and Theorem 4, yet similar analysis can be applied to (55b). Now, expanding (55a) with the definition of $(B_{k+1}, F_k)$

$$\|B_{k+1}(\mathbf{X}_{k+1}) - (\mathbf{X}_{k+1} + F_k(\mathbf{X}_k) - F_k(\mathbf{X}_{k+1}))\|^2$$
$$= \|(\mathbf{X}_{k+1} + \gamma_{k+1} b_{k+1}(\mathbf{X}_{k+1})) - (\mathbf{X}_{k+1} + \mathbf{X}_k + \gamma_{k+1} f_k(\mathbf{X}_k) - \mathbf{X}_{k+1} - \gamma_{k+1} f_k(\mathbf{X}_{k+1}))\|^2$$
$$= \|\underbrace{\gamma_{k+1} f_k(\mathbf{X}_{k+1})}_{\text{①}} + \underbrace{\gamma_{k+1} b_{k+1}(\mathbf{X}_{k+1})}_{\text{②}} - \underbrace{(\mathbf{X}_k + \gamma_{k+1} f_k(\mathbf{X}_k) - \mathbf{X}_{k+1})}_{\text{③}}\|^2, \tag{56}$$

which resembles the term $\|\mathbf{Z}_t + \widehat{\mathbf{Z}}_t - g\nabla_{\boldsymbol{x}} \log p_t^{\text{SB}}\|^2$ in (15). While ② indeed corresponds to our backward policy $\widehat{\mathbf{Z}}(k+1, \mathbf{X}_{k+1})$ after time discretization, ① slightly differs from $\mathbf{Z}(k+1, \mathbf{X}_{k+1})$ in how time is integrated, $\gamma_{k+1} f(k, \mathbf{X}_{k+1})$ *vs.* $\mathbf{Z}(k+1, \mathbf{X}_{k+1})$. On the other hand, ③ may be seen as an approximation of $g\nabla_{\boldsymbol{x}} \log p_t^{\text{SB}}$, which, crucially, is *not* utilized in SB-FBSDE training. Since $\nabla_{\boldsymbol{x}} \log p_t^{\text{SB}}$ is often intractable (nor does SB-FBSDE try to approximate it), SB-FBSDE instead uses the *divergence-based* objective (16), which does not appear in their practical training.

**SDE model class.** It is important to recognize that both concurrent methods are rooted in the *classical* SB formulation with the following SDE model,

$$d\mathbf{X}_t = f(t, \mathbf{X}_t)dt + \sqrt{2\gamma}\, d\mathbf{W}_t,$$

which, crucially, differs from the SDE concerned by both our SB-FBSDE and SGM,

$$d\mathbf{X}_t = f(t, \mathbf{X}_t)dt + g(t)d\mathbf{W}_t, \tag{57}$$

in that the diffusion $g(t)$ is a *time-varying* function. This implies that the connection between classical SB models and SGM can only be made in discrete-time after choosing proper step sizes. For instance, De Bortoli et al. (2021) considers the Euler-Maruyama discretization (see their §C.3),

$$\mathbf{X}_{k+1} = \mathbf{X}_k + \gamma_{k+1} f(k, \mathbf{X}_k) + \sqrt{2\gamma_{k+1}}\, \epsilon, \tag{58}$$

where $\epsilon \sim \mathcal{N}(\mathbf{0}, \boldsymbol{I})$. In order for (58) to match the discretization of (57), where $g(t)$ is often a monotonically increasing function, the step sizes $\{\gamma_k\}_{k=1}^N$ must also increase monotonically. However, since $\nabla_{\boldsymbol{x}} \log p_t^{\text{SB}}$ is approximated in De Bortoli et al. (2021) using the states from two consecutive steps (see (55)), this may also affect the accuracy of the regression targets.

In contrast, our SB-FBSDE is grounded on the recent SB theory (Caluya & Halder, 2021), which considers the *same* SDE model as in (57). As such, connection between SB-FBSDE and SGM is made directly in continuous-time (and can be extended to discrete-time flawlessly); hence unaffected by the choice of numerical discretization or step sizes.

**Model parametrization.** While Vargas et al. (2021) utilizes non-parametric models, *e.g.* Gaussian processes (hence are not directly comparable), both De Bortoli et al. (2021) and SB-FBSDE use DNNs to approximate the SB policies.

**Training algorithm and convergence.** Both concurrent methods rely on solving SB with IPF algorithm, which performs alternate training between the forward/backward policies. While SB-FBSDE can also be trained with IPF (see Alg. 3), we stress that it is also possible to train *both policies jointly* whenever the computational budget permits. Interestingly, this joint optimization – which is not presented in concurrent methods – resembles the training scheme of the recently-proposed diffusion flow-based model (Zhang & Chen, 2021). We highlight these flexible training procedures arising from the unified viewpoint provided in Theorem 4. Finally, with the close relation between Alg. 3 and IPF (despite with different objectives and SDE model classes), convergence analysis from classical IPF can be applied with few efforts. We leave it as a promising future work.

**Corrector Sampling.** While both De Bortoli et al. (2021) and SB-FBSDE implement corrector sampling, they corresponds to different quantities. Specifically, our SB-FBSDE relies on the same predictor-corrector scheme proposed in SGM (see Sec 4.2 in Song et al. (2020)), where the "*corrector*" part is made with a Langevin sampling using the desired optimal density "$\nabla \log p_t$". In SB, this term corresponds *exactly* to adding the outputs of our networks "$\mathbf{Z} + \widehat{\mathbf{Z}}$". This computation differs from the corrector sampling appearing in De Bortoli et al. (2021), which relies on single network (*i.e.* either $\mathbf{Z}$ or $\widehat{\mathbf{Z}}$). Crucially, this implies that the two methods is approaching different target distributions; hence leading to different training results. Notably, it has been reported in De Bortoli et al. (2021) that corrector sampling only gives negligible improvement (see §J.2 in De Bortoli et al. (2021)), yet in our case we observe major quantitative improvement (up to 4 FID).

## D    EXPERIMENT DETAILS

**Table 1 with other models.**

Table 2: CIFAR10 evaluation.

| Model Class | Method | NLL ↓ | FID ↓ |
|---|---|---|---|
| Optimal Transport | **SB-FBSDE (ours)** | 2.96 | 3.01 |
| | DOT (Tanaka, 2019) | - | 15.78 |
| | Multi-stage SB (Wang et al., 2021) | - | 12.32 |
| | DGflow (Ansari et al., 2020) | - | 9.63 |
| SGMs | SDE (deep, sub-VP; Song et al. (2020)) | 2.99 | 2.92 |
| | ScoreFlow (Song et al., 2021) | 2.74 | 5.7 |
| | VDM (Kingma et al., 2021) | **2.49** | 4.00 |
| | LSGM(Vahdat et al., 2021) | 3.43 | **2.10** |
| VAEs | VDVAE (Child, 2020) | 2.87 | - |
| | NVAE (Vahdat & Kautz, 2020) | 2.91 | 23.49 |
| | BIVA (Maaløe et al., 2019) | 3.08 | - |
| Flows | FFJORD (Grathwohl et al., 2018) | 3.40 | - |
| | VFlow (Chen et al., 2020) | 2.98 | - |
| | ANF (Huang et al., 2020) | 3.05 | - |
| GANs | AutoGAN (Gong et al., 2019) | - | 12.42 |
| | StyleGAN2-ADA (Karras et al., 2020a) | - | 2.92 |
| | LeCAM (Park & Kim, 2021) | - | 2.47 |

Figure 7: Training Hyper-parameters

| Dataset | learning rate | time steps | batch size | variance of $p_{\text{prior}}$ |
|---|---|---|---|---|
| Toy | 2e-4 | 100 | 400 | 1.0 |
| Mnist | 2e-4 | 100 | 200 | 1.0 |
| CelebA | 2e-4 | 100 | 200 | 900.0 |
| CIFAR10 | 1e-5 | 200 | 64 | 2500.0 |

Figure 8: Network Architectures

| Dataset | $\mathbf{Z}_t(\cdot, \cdot; \theta)$ and # of parameters | $\widehat{\mathbf{Z}}_t(\cdot, \cdot; \phi)$ and # of parameters |
|---|---|---|
| Toy | FC-ResNet (0.76M) | FC-ResNet (0.76M) |
| Mnist | reduced Unet (1.95M) | reduced Unet (1.95M) |
| CelebA | Unet (39.63M) | Unet (39.63M) |
| CIFAR10 | NCSN++ (62.69M) | Unet (39.63M) |

**Training.** We use Exponential Moving Average (EMA) with the decay rate of 0.99. Table 7 details the hyper-parameters used for each dataset. As mentioned in De Bortoli et al. (2021), the alternate training scheme may substantially accelerate the convergence under proper initialization. Specifically, when $\mathbf{Z}_t$ is initialized with degenerate outputs (*e.g.* by zeroing out its last layer), training $\widehat{\mathbf{Z}}_t$ at the first $K$ steps can be made in a similar SGM fashion since $p_t^{\text{SB}}$ now admits analytical expression. As for the proceeding stages, we resume to use (18, 19) since $(\mathbf{Z}_t, \widehat{\mathbf{Z}}_t)$ no longer have trivial outputs.

**Data pre-processing.** MNIST is padded from 28×28 to 32×32 to prevent degenerate feature maps through Unet. CelebA is resized to 3×32×32 to accelerate training. Both CelebA and CIFAR10 are augmented with random horizontal flips to enhance the diversity.

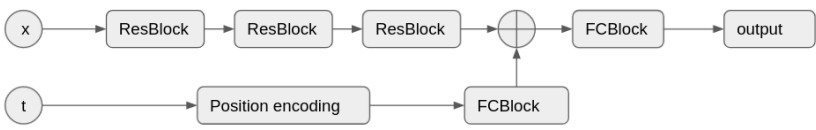

Figure 9: Network architecture for toy datasets.

**Sampling.** The sampling procedure is summarized in Alg. 4. Given some pre-defined signal-to-noise ratio $r$ (we set $r = 0.05$ for all experiments), the Langevin noise scale $\sigma_{t,i}$ at each time step $t$ and each corrector step $i$ is computed by

$$\sigma_{t,i} = \frac{2r^2 g^2 \|\epsilon_i\|^2}{\|(\mathbf{Z}(t, \mathbf{X}_{t,i}) + \widehat{\mathbf{Z}}(t, \mathbf{X}_{t,i}))\|^2}, \quad (59)$$

**Network architectures.** Table 8 summarizes the network architecture used for each dataset. For toy datasets, we parameterize $\mathbf{Z}(\cdot, \cdot; \theta)$ and $\widehat{\mathbf{Z}}(\cdot, \cdot; \phi)$ with the

---

**Algorithm 4** Generative Process of SB-FBSDE

**Input:** $p_{\text{prior}}$, policies $\mathbf{Z}(\cdot, \cdot; \theta)$ and $\widehat{\mathbf{Z}}(\cdot, \cdot; \phi)$
Sample $\mathbf{X}_T \sim p_{\text{prior}}$.
**for** $t = T$ **to** $\Delta t$ **do**
    Sample $\epsilon \sim \mathcal{N}(\mathbf{0}, \mathbf{I})$.
    Predict $\mathbf{X}_{t,1} \leftarrow \mathbf{X}_t + g\,\widehat{\mathbf{Z}}_t \Delta t + \sqrt{g\Delta t}\,\epsilon$.
    **for** $i = 1$ **to** $N$ **do**
        Sample $\epsilon_i \sim \mathcal{N}(\mathbf{0}, \mathbf{I})$.
        Compute $\nabla_{\boldsymbol{x}} \log p_{t,i}^{\text{SB}} \approx [\mathbf{Z}(t, \mathbf{X}_{t,i}) + \widehat{\mathbf{Z}}(t, \mathbf{X}_{t,i})]/g$.
        Compute $\sigma_{t,i}$ with (59).
        Correct $\mathbf{X}_{t,i+1} \leftarrow \mathbf{X}_{t,i} + \sigma_{t,i}\nabla_{\boldsymbol{x}} \log p_{t,i}^{\text{SB}} + \sqrt{2\sigma_{t,i}}\,\epsilon_i$.
    **end for**
    Propagate $\mathbf{X}_{t-\Delta t} \leftarrow \mathbf{X}_{t,N}$.
**end for**
**return** $\mathbf{X}_0$

---

architectures shown in Fig. 9. Specifically, *FCBlock* represents a fully connected layer followed by a swish nonlinear activation (Ramachandran et al., 2017). As for MNIST, we consider a smaller version of Unet (Ho et al., 2020) by reducing the numbers of residual block, attention heads, and channels respectively to 1, 2, and 32. Unet and NCSN++ respectively correspond to the architectures appeared in Ho et al. (2020) and Song et al. (2020).

**Remarks on Table 1.** We note that the values of our SB-FBSDE reported in Table 1 are computed *without* the Langevin corrector due to the computational constraint. For all other experiments, we adopt the Langevin corrector as it generally improves the performance (see Fig. 6). This implies that our results on CIFAR10, despite already being encouraging, may be further improved with the Langevin corrector.

**Remarks on Fig. 5.** To estimating $\text{KL}(p_T, p_{\text{prior}})$, we first compute the pixel-wise first and second moments given the generated samples $\mathbf{X}_T$ at the end of the forward diffusion. After fitting a diagonal Gaussian to $\{\mathbf{X}_T\}$, we can apply the analytic formula for computing the KL divergence between two multivariate Gaussians.

**Remarks on Fig. 6.** To accelerate the sampling process with the Langevin corrector, for this experiment we consider a reduced Unet (see Table 8) for CelebA. The FID scores on both datasets are computed with 10k samples. We stress, however, that the performance improvement using the Langevin corrector remains consistent across other (larger) architectures and if one increases the FID samples.

# E ADDITIONAL EXPERIMENTS

**Comparison to De Bortoli et al. (2021) under same setup.** To demonstrate the superior performance of our model, we conduct experiments with the exact same setup implemented in De Bortoli et al. (2021). Specifically, we adopt the same network architecture (reduced U-net), image preprocessing (center-cropping 140 pixel and resizing to $32 \times 32$), step sizes ($N$=50), and horizon (0.5 second) for fair comparison. Comparing our Fig. 10b to De Bortoli et al. (2021) (see their Fig. 6), it is clear that images generated by our model have higher diversity (*e.g.* color skin, facing angle, background, etc) and better visual quality. We conjecture that our performance difference may come from *(i)* the (in)sensitivity to numerical discretization between our divergence objectives and their mean-matching regression, and *(ii)* the foundational differences in how diffusion coefficients are designed.

(a) Ground Truth          (b) SB-FBSDE Generated Image

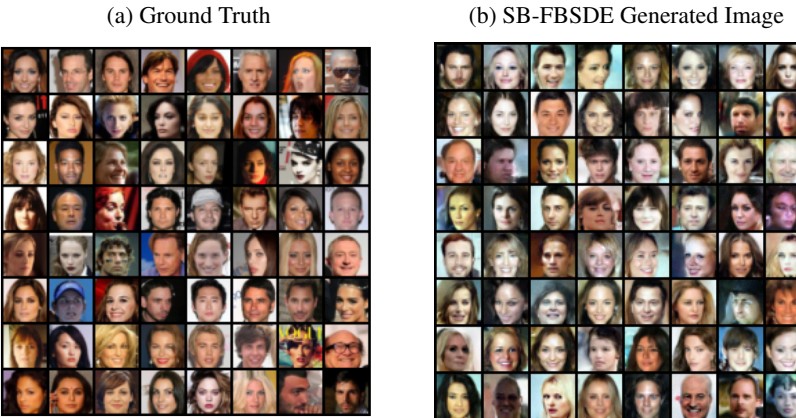

Figure 10: Comparison between images generated by ground truth and SB-FBSDE on reduced CelebA. Our SB-FBSDE is trained under the same data pre-processing, network architecture and stepsizes implemented in De Bortoli et al. (2021).

(a) SGM/50k          (b) SGM/50k + SB/b/5k

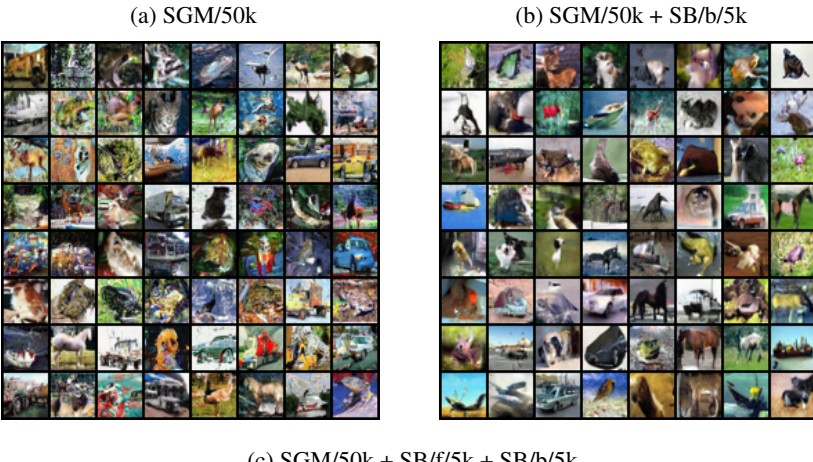

(c) SGM/50k + SB/f/5k + SB/b/5k

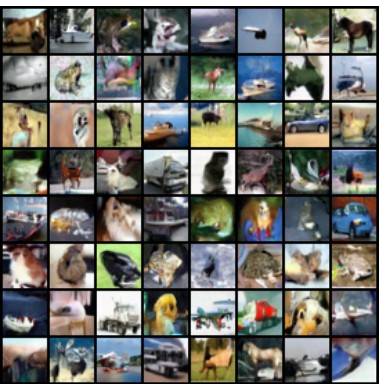

Figure 11: Qualitative results at the different stages of training. *(a)* Results after 50k training iterations using SGM's regression loss. *(b)* Refine the results of Fig. 11a by training the backward policy using (18) with 5k iterations. *(c)* Refine the results of Fig. 11a with a full SB-FBSDE stage using (18,19).

**SGM regression training + SB divergence-based training.** Table 3 reports the FID (using 10k samples, without corrector steps) at different stages of CIFAR10 training. We first train the backward policy with SGM's regression loss for a sufficient long iterations (50k) until the FID roughly converges. Then, we switch to our alternate training (Alg. 3) using the divergence-based objectives. Crucially,

Table 3: SGM regression training + SB divergence-based training. We denote "SGM/50k" as "training 50k steps using SGM loss", and "SB/{f,b}/5k" as "training forward/backward policy with 5k steps using our divergence loss", and etc.

| | initialization | SGM/10k | SGM/20k | SGM/50k | SGM/50k + SB/b/5k | SGM/50k + SB/f/5k + SB/b/5k |
|---|---|---|---|---|---|---|
| FID | 448 | 41.37 | 35.47 | 33.68 | 13.35 | 11.85 |

with only 5k iterations of our divergence-based training, we drop the FID dramatically down to 13.35 from 33.68. With a full stage of training (last column), the FID decreases even lower to 11.85. The qualitative results are provided in Fig. 11. Comparing Fig. 11a (corresponds to "SGM/50k" in Table 3) and Fig. 11b (corresponds to "SGM/50k + SB/b/5k" in Table 3), it can be seen that the visible flaw and noise have been substantially improved.

**Additional Figures**

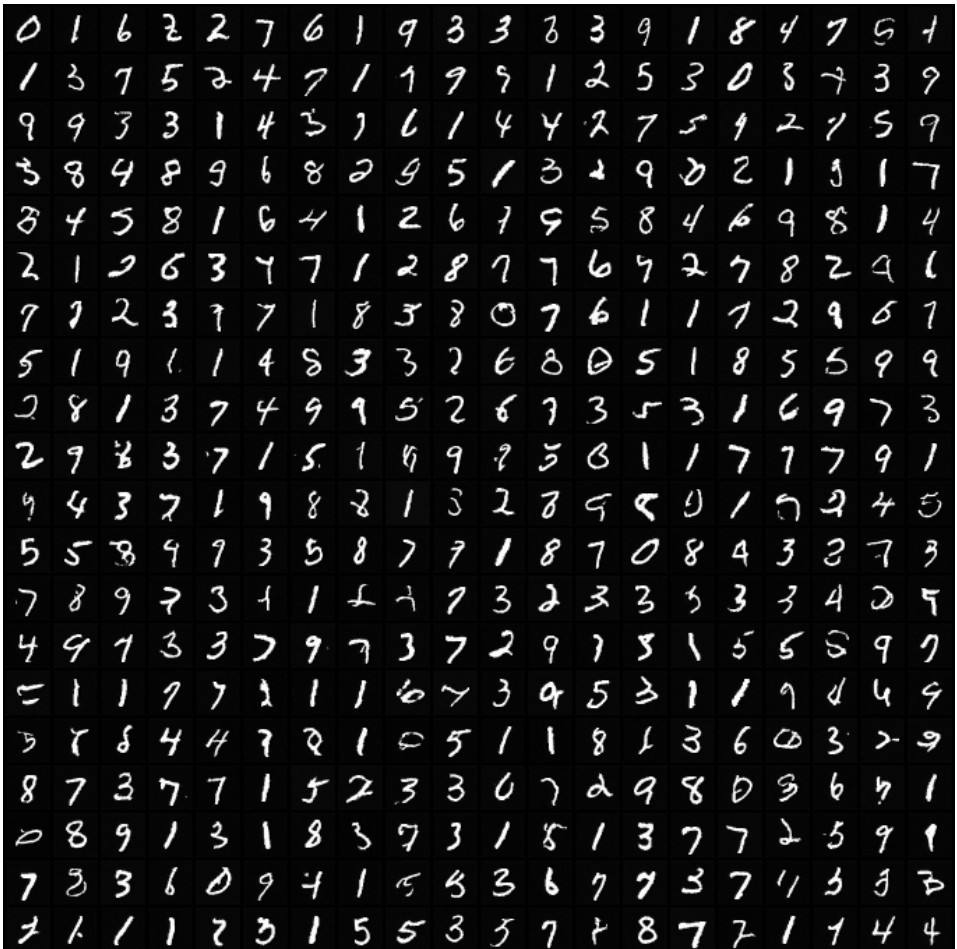

Figure 12: Uncurated samples generated by our SB-FBSDE on MNIST.

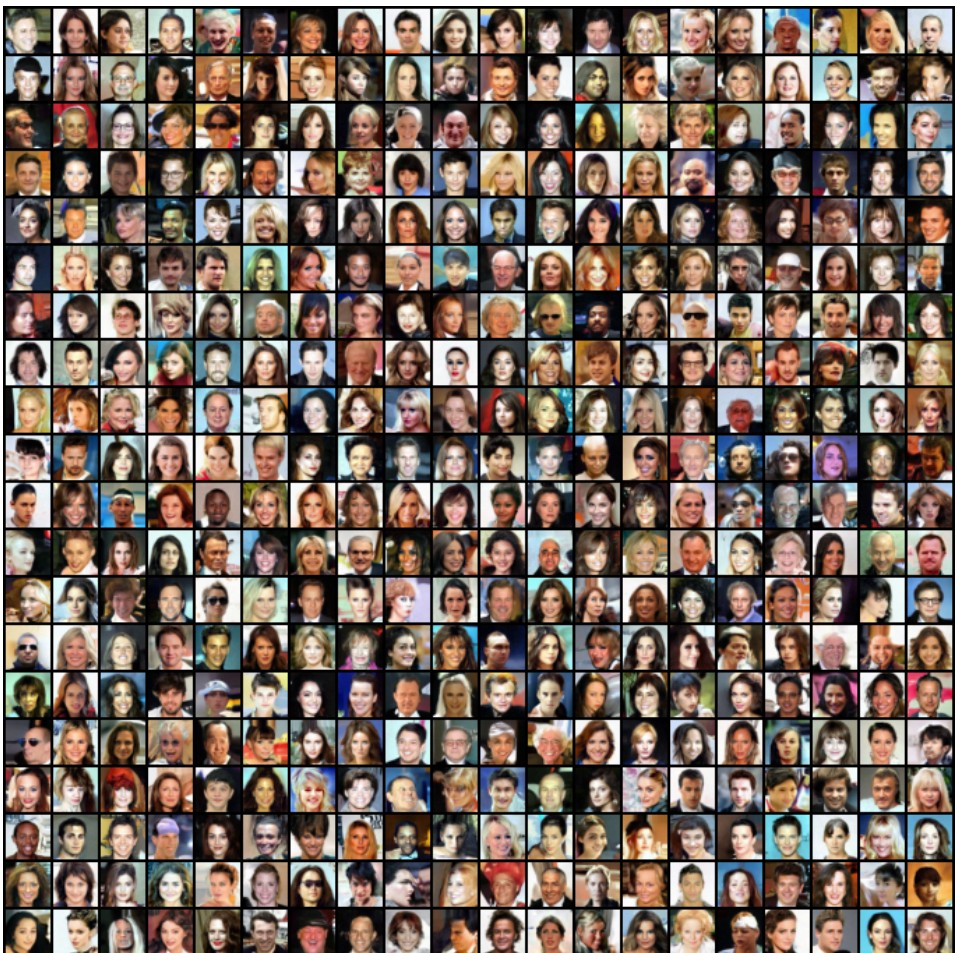

Figure 13: Uncurated samples generated by our SB-FBSDE on resized CelebA.

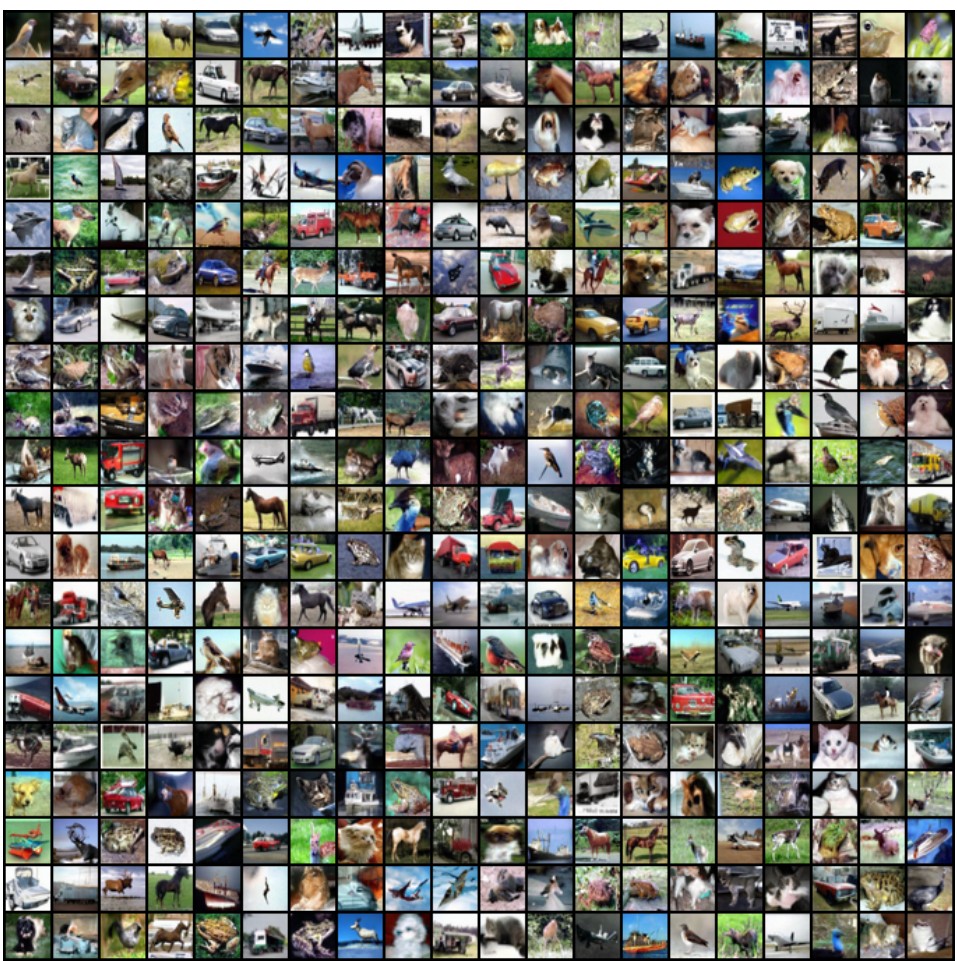

Figure 14: Uncurated samples generated by our SB-FBSDE on CIFAR10.

