# OpenReview forum: "Likelihood Training of Schrödinger Bridge using Forward-Backward SDEs Theory"
_ICLR.cc/2022/Conference — ICLR 2022 Poster_

### Official Review · Reviewer_dQMT · 2021-11-02

**Correctness:** 3
**Technical Novelty And Significance:** 3
**Empirical Novelty And Significance:** 2
**Recommendation:** 6
**Confidence:** 5

**Main Review:**

STRENGTHS:

-Overall the presentation of the paper is good with a clear presentation of the
score-based generative models and the Schrodinger bridge problem.

-The idea of using the FBSDE framework is new and original. I think that the
idea of using Feynman-Kac based representations to solve (non)linear PDEs is
definitely a good idea.

-The toy experiments and the generative modeling experiments are satisfactory.

WEAKNESSES:

-My first concern is with the formulation of the loss function that I find
opaque to say the least. Looking at (14) it turns out that the loss function is
in fact given by
$\int_0^T \mathbb{E}[\| \hat{z}_t - g \nabla \log p_t^{SB} + z_t \|^2]$ where
the expectation is taken over the path measure induced by the forward SDE. It is
not clear how $\nabla \log p_t^{SB}$ is computed. The classical ideas of [1,2]
cannot be used since the $\nabla \log p_t^{SB}$ is not given by a
Ornstein-Ulhenbeck process. Therefore the authors might rely on the idea used in
that approximate $\nabla \log p_t$ for general processes using their Langevin
discretization. The authors should clearly state what loss function they use in
practice (discretization of the time + approximation of $\nabla \log p_t$). Maybe they use the first part of (14) to compute the loss function but this is not clear either. The
pseudocode provided by the authors is not helpful and there is no anonymous
repository to check out the code.

-My second concern is a consequence of the first one. The authors claim that
their work is different from [3,4] but in practice Algorithm 1 is the same
training algorithm as the one derived in [3]. Indeed because the authors propose
an alternate minimization of the loss function they derive the loss function
they minimize at each step corresponds exactly to compute the IPF step
(Algorithm 1 in the current paper is almost the same as [3, Algorithm 1], see also [4, Algorithm 1], with the only difference being in how the loss is computed which is not clear from the current paper, see previous comment) . In this sense
the algorithm provided by the authors corresponds to a rewriting of [3,4]. Hence I
think the claims of the authors concerning the originality of the method is overstated and misleading. Similarly it is misleading to state that no
connection with Langevin based sampling as been established for Schrodinger
bridge models, see [3,4]. To summarize there is a clear overlap between the
method proposed by the authors and the work of [3,4] which is not acknowledged in
the paper.

-Another concern I have is related to the derivation of the algorithm. Given how
the authors derive Algorithm 1 there is no guarantee that 1) the procedure is
going to converge and 2) if it converges that the limiting bridge is indeed the
Schrodinger bridge. Both these facts can be obtained using the IPF approach, see
[3, Proposition 6]. As of now Theorem 4 is only valid at equilibrium, i.e. for
the optimal set of controls $(z_t, \hat{z}_t)$. For arbitrary controls
Theorem 4 might still be valid (or at least in a lower-bound form) but this is not
what is currently stated in the paper. The way the paper is written it looks like Algorithm 1 is motivated and justified by Theorem 4 which is not the case.


-I have a minor concern with the contribution of the authors to the FBSDE
theory. It seems that Lemma 2 and the current FBSDE is not enough to provide a
non-linear Feynman-Kac representation. I find that this contribution is not
clearly presented by the authors. Also the proof seems to be very heuristic, the
authors do not check the necessary regularity conditions to apply Ito
formulas. They refer to the ``same regularity conditions in Lemma 2'' but Lemma
2 is not informative because it is only stated that the functions ``G, f, h and
$\varphi$ satisfy mild regularity conditions''. The concept of viscosity
solution is not reintroduced in the paper (even in the supplementary material).

-Finally I have also concerns with the experiments. In particular there is no
comparison with [3,4] which are currently the concurrent for score-based
generative modeling using Schrodinger bridge. In particular, since Algorithm 1
and [3, Algorithm 1] appear almost identical it is unfortunate that the authors do not
provide any comparison with this approach in similar setting (same number of
steps, stepsize and without correcting step).

COMMENTS:

-It would be interesting to observe what is the generative model obtained after
one iteration of the algorithm, i.e. when the backward network is updated for
the first time. This should recover the generative model obtained using SGM and
further steps are a refinement of the method.

-It would be interesting to precisely quantify the influence of the corrector
(which seems to constitute the only difference between Algorithm 2 and the sampling
procedure proposed in [3]).

-``Poof'' and ``Remakrs'' in the title of the sections in the appendix.

-In Algorithm 1 do the authors really use a gradient descent algorithm? Or is a
more efficient like ADAM is used to train the neural network?


**Summary Of The Paper:**

In this paper the authors introduce a score-based generative model which relies
on a Schrodinger bridge formulation. More precisely the authors derive a loss
function for the control of a forward/backward SDE. By alternating the
minimization of the loss function between the forward control and the backward
control they obtain an approximation of the Schrodinger bridge. This loss
function is obtained using the forward backward SDE (FBSDE) framework to obtain
a non-linear Feynman-Kac representation of the evolution of the
log-potentials. This theoretical/methodological contribution is accompanied by a
experiments in generative modeling on the MNIST/CelebA/CIFAR-10 datasets.

**Summary Of The Review:**

To conclude I think that the idea of using FBSDE to derive the Schrodinger bridge
is original. However the method has a lot of overlap with the
works of [3,4] which is not acknowledged by the authors. The authors do not
precise which loss function they use in practice and because they do not give
access to the code (or at least to a detailed pseudo code) it is hard to
determine what is the original methodological contribution of the paper.
I think that the paper is not mature enough and that a true justification of Algorithm 2 with FBSDE is still missing. Experimentally speaking the authors did not compare their work with existing SB methods.
Based on these comments I recommend the rejection of the paper.

---

> ### Author Response · Authors · 2021-11-17
> **Author Response to Reviewer dQMT (Part 1/3)**
>
> **1. Comparison to [3,4]**
>
> We first stress that by the time of our submission, both concurrent works [3,4] were still under peer-review, and neither had released code nor provided sufficient information for reproducing. Further, there are no quantitative results (_e.g._ FID) on related datasets (Cifar10). In the initial submission, there was little we could do other than include them in the intro and provide judgment with our best knowledge. Below we provide clarifications on how our work differs from [3,4] in various aspects. These discussions are included in Sec 1, Sec 3.2, Sec 3.3, and Appendix D.2
>
> ---
>
> **1.1 Training loss**
>
> - **Our method does NOT rely on $\nabla \log p_t^\text{SB}$.** Since $\nabla \log p_t^\text{SB}$ is intractable for any nontrivial $(\mathbf{Z}, \widehat{\mathbf{Z}})$ (nor do we intend to approximate it), **all of our training processes rely on _divergence-based_ objectives**, which do not appear in their practical training. These objectives are now detailed in Eq (16,18,19) with Eq (16) being the objective for joint training (see Response **1.2**) and Eq (18,19) being the ones for alternate training. In the revision, we make this clear in both Sec 3.2 (paragraph below Theorem 4) and Sec 3.3 (Alg 1,2,3 and the paragraph below). We urge the reviewer to recognize the difference.
>
>
> - As conjectured by the reviewer, $\lVert \mathbf{Z}_t - \nabla \log p_t^\text{SB} + \widehat{\mathbf{Z}}_t \rVert^2$
> indeed relates to the _mean-matching regression_ in [3,4]. However, in Appendix D.2 we show that they are not exactly the same. We refer the reviewer to Eq (56,57) on page 25 for the derivation. When training $\widehat{\mathbf{Z}}_t$, the first term $\mathbf{Z}_t$ will differ in how time is integrated, and the second term $\nabla \log p_t^\text{SB}$ is approximated after time discretization. Only the last term $\widehat{\mathbf{Z}}_t$ coincides exactly. Crucially, that $\nabla \log p_t^\text{SB}$ is _approximated after discretization_ (using $\mathbf{X}$ from two consecutive time steps) implies that their training performance will be affected by the chosen step sizes and numerical solver. On the contrary, our divergence-based objectives are readily valid in continuous-time; hence less sensitive to numerical discretization.
>
>
> - The table below further highlights the performance difference between regression and divergence losses. Both models are trained from the _same_ parameter whose initial FID is 35.47. Despite having longer per-iteration runtime, our divergence loss leads to much lower FID (with 10k samples) after the same period of time. While the regression loss makes only minor progress (35.47 -> 33.68), our divergence loss reduces the FID by half (35.47 -> 16.08).
>
>     | | Total Runtime (second/iterations) | FID after 7 hours |
>     |---|---|---|
>     | Regression Loss | 1.41 | 33.68 |
>     | Divergence Loss (ours; Eq (18,19)) | 3.63 | 16.08 |
>
> ---
>
> **1.2 Training algorithms & SDE model**
>
> - We first stress that the alternate training is one of the _many_ training procedures one can derive from Theorem 4. For instance, the 2D toy datasets are solved with _joint_ optimization, _i.e._ we train both policies, $\mathbf{Z}(\cdot,\cdot; \theta)$ and $\widehat{\mathbf{Z}}(\cdot,\cdot; \phi)$, at the same time. For higher-dimensional (_e.g._ image) datasets where it can be too expensive to keep the entire computational graph, we retreat to the alternate training scheme. In practice, joint optimization can lead to faster convergence (since the computational graph provides rich gradient flow) yet at the cost of memory complexity. **This joint optimization is closer to diffusion flow-based models [5] and not presented in [3,4].**
>
>
> - The alternate training is indeed related to classical IPF algorithm [6]. However, aside from the alternate structure, which is shared across all proceeding SB methods [3,4,ours], each method still differs from another in what is being optimized (either exactly or approximately) and implemented. For instance, the originally intractable term $\nabla \log p_t^\text{SB}$ -- that [3] aims to _approximate_ -- is _replace_ in our framework with the divergence, which leads to dramatic performance difference as shown in Response **1**.
>
>
> - In additional to training loss, these methods also differ in their SDE models. Specifically, we adopt the recent advance in SB theory [7] that extends classical SB models (used in [3,4]) to the exact SDE model appearing in SGM. While the SDEs of [3,4] are based on $\mathrm{d}\mathbf{X}_t$=...+$\sqrt{2}\mathrm{d}\mathbf{W}_t$ with _time-invariant_ diffusion coefficient, ours (and SGM) feature _time-varying_ diffusion $g(t)\mathrm{d}\mathbf{W}_t$. This implies that the connection between [3,4] and SGM can only be made after time discretization by carefully choosing each step size (see Sec C.3 of [3]). In contrast, our theoretical connection is derived readily in continuous-time; hence unaffected by the choice of numerical discretization.

---

> > ### Author Response · Authors · 2021-11-17
> > **Author Response to Reviewer dQMT (Part 2/3)**
> >
> > **1.2 Training algorithms & SDE model (continue)**
> >
> > - From a more practical viewpoint, $g(k)$ (or equivalently their step sizes) plays a key role in training stability and generative performance [8,9]. For [3], however, these step sizes will also affect how the regression target is computed. Hence, instead of using the conventional increasing diffusion, $\cdots < g(k-1) < g(k) < \cdots$, [3] has to set the step sizes to first increasing then decreasing (essentially to ensure the regression targets at boundaries are well-approximated).
> >
> >
> > - With all these reasonings, we believe it is inaccurate to state "_our algorithm is a rewriting of [3,4]_" without examining the underlying mathematical formulation -- in the same way that it is inaccurate to treat [3,4,ours] as a rewriting of classical IPF [6]. We urge the reviewer to recognize the distinction.
> >
> >
> > - To avoid any future confusion, these discussions are included in both Sec 1 and Sec 3.3. The pseudocode for both joint and alternate training are detailed in Alg 2 and 3. Since Alg 3 itself closely resembles IPF (yet with different losses and models), we believe convergence analysis from classical IPF can be applied with few efforts. We leave it as a promising future work.
> >
> > ---
> >
> > **1.3 Langevin "corrector" sampling in Table 1**
> >
> > - Regarding "Langevin sampling" in Table 1, we indeed misused the term. Here, we refer specifically to the predictor-corrector scheme proposed in SGM (see Sec 4.2 in [10]), where the "corrector" part is made with a Langevin sampling using the desired optimal density "$\nabla \log p_t^\text{SB}$". In SB, this term corresponds exactly to $\mathbf{Z}+\widehat{\mathbf{Z}}$. This differs from the Langevin appearing in [3], which relies only on single network (either $\mathbf{Z}$ or $\widehat{\mathbf{Z}}$). Crucially, this implies the two methods are approaching different target distributions; hence leading to different training results. This may also explain why the corrector sampling in [3] was reported with negligible improvement (Sec J.2 in [3]), yet in our case we observe major quantitative improvement (up to 4 FID).
> >
> >
> > - We admit that Table 1 could have been stated more precisely. In the revision, we remove this table and replace it with a full paragraph so that it better reflects our clarifications so far.
> >
> > ---
> >
> > **1.4 Acknowledgment to [3]**
> >
> > - The only exception when $\nabla \log p_t^\text{SB}$ becomes tractable (at least in our case) is the first training stage. In which case, if we initialize $\mathbf{Z}:= \mathbf{0}$ then first train $\widehat{\mathbf{Z}}$, the term $\nabla \log p_t^\text{SB} = \nabla \log p_t^\text{(1)}$ can be computed just like SGM. As this is originally mentioned in [3], we have acknowledged them in the revision (see page 20).
> >
> >
> > - To improve practical efficiency, we follow similar advice from [3] by caching the sampled trajectories and refreshing them at a lower frequency, in a similar vein as the "reply buffer" adopted in Deep RL. We have acknowledged [3] in this regard (see Sec 3.3).
> >
> > ---
> >
> > **2. Parameterized lower bound & Theorem 4**
> >
> > - We update the log-likelihood expressions of both SGM (Eq (3)) and SB (Theorem 4) for clarification. Specifically, the value $\mathcal{L}_\text{SGM}(\boldsymbol{x};\theta)$ in Eq (3) serves as a parameterized lower bound of the SGM log-likelihood $\log p_0^\text{SGM}(\boldsymbol{x}) \ge \mathcal{L}_\text{SGM}(\boldsymbol{x};\theta)$, and the equality holds when the score network (parameterized by $\theta$) coincides with the true score function [10].
> >
> >
> > - Meanwhile, Theorem 4 now gives the exact log-likelihood of SB model, $\log p_0^\text{SB}(\boldsymbol{x})$, as a function of $\{\mathbf{X}_t,\mathbf{Z}_t,\widehat{\mathbf{Z}}_t\}$, which can be obtained by solving the FBSDE system (13). A similar parameterized lower bound $\log p_0^\text{SB}(\boldsymbol{x}) \ge \mathcal{L}_\text{SB}(\boldsymbol{x};\theta,\phi)$ can be obtained when $\mathbf{Z}_t \approx \mathbf{Z}(\cdot,\cdot; \theta)$ and $\widehat{\mathbf{Z}}_t \approx \widehat{\mathbf{Z}}(\cdot,\cdot; \phi)$ are approximated with some parameterized models. The equality holds when the parameterized models coincide with the values suggested by the FBSDE system.
> >
> >
> > - Just like how Eq (3) is used in [9] to justify the SGM training, Theorem 4 also motivates the training procedures in Alg 1,2,3. These results can be viewed as a nonlinear generalization of the SGM theory (also concurred by Reviewer 8tG7).
> >
> > ---
> >
> > [3] De Bortoli et al 2021, "Diffusion SB with SGM"
> > [4] Vargas et al 2021, "Solving SB via MLE"
> > [5] Zhang et al 2021, "Diffusion Normalizing flow"
> > [6] Kullback 1968, "Prob density..."
> > [7] Caluya et al 2021, "Wasserstein ... for SB ... nonlinear drift"
> > [8] Song et al 2019, "Generative Modeling ..."
> > [9] Song et al 2021, "MLE of SGM"
> > [10] Song et al 2020, "SGM through SDE"

---

> > > ### Author Response · Authors · 2021-11-17
> > > **Author Response to Reviewer dQMT (Part 3/3)**
> > >
> > > **3. Additional experiments**
> > >
> > > **3.1 Comparison to [3] under same setups**
> > >
> > > - Following the reviewer's suggestion, we run our method with the _same_ setups implemented in [3] on CelebA. We adopt the same network architecture (reduced U-net), number of steps (N=50), data-preprocessing, and without corrector steps. The qualitative results are provided in Fig 13b (see Appendix D.3; page 26). Comparing our Fig 13b to their results (see their Fig 6 in [3]), it is clear that images generated by our model have higher diversity (_e.g._ color skin, facing angle, background, etc) and better visual quality. We conjecture that our performance difference may come from _(i)_ the (in)sensitivity to numerical discretization between our divergence objectives and their mean-matching regression, and _(ii)_ the foundational differences in how diffusion coefficients are designed.
> > >
> > > - Unfortunately, [3] does not provide the numerical value of quantitative (FID) result, and we were unable to reproduce using their latest release of implementation [11]. Finally, we note that [4] utilizes non-parametric models, _e.g._ Gaussian processes, hence are not directly comparable.
> > >
> > >
> > > [11] Reproducibility seems to be an ongoing issue, see https://github.com/JTT94/diffusion_schrodinger_bridge/issues/1
> > >
> > > ---
> > >
> > > **3.2 SGM + SB refinement**
> > >
> > > - The below table reports the FID (using 10k samples) at different stages of Cifar10 training. We first train the backward policy with SGM's regression loss for a sufficient long iterations (50k) until the FID roughly converges. Then, we switch to the alternate training (Alg 3) using our divergence-based objectives. We denote "SGM/50k" as "training 50k steps using SGM loss", and "SB/{f,b}/5k" as "training forward/backward policy with 5k steps using our divergence loss", and etc.
> > >
> > >     |  | initialization | SGM/10k | SGM/20k | SGM/50k | SGM/50k + SB/b/5k | SGM/50k + SB/f/5k + SB/b/5k |
> > >     |---|---|---|---|---|---|---|
> > >     | FID | 448 | 41.37 | 35.47 | 33.68 | 13.35 | 11.85 |
> > >
> > > - Crucially, with only 5k iterations of our divergence-based training, we drop the FID dramatically down from 33.68 to 13.35. With a full stage of training (last column), the FID decreases even lower to 11.85. The qualitative results are provided in Fig 14 (see Appendix D.3; page 27). Comparing Fig 14a (corresponds to "SGM/50k") and Fig 14b (corresponds to "SGM/50k + SB/b/5k"), it can be noticed that the visible flaws and noises have been substantially improved.
> > >
> > > ---
> > >
> > > **4. Other clarifications**
> > >
> > > - In the updated revision, we restate Lemma 2, Theorem 3 & 4 with clearer notations, where we capitalize all random variables/vectors to better distinguish SDE, $\mathrm{d}\mathbf{X}_t$=..., and (deterministic) PDE, $\frac{\partial v(t,\boldsymbol{x})}{\partial t}$=..., dynamics. All related regularity conditions (_e.g._ the missing $v, \Psi, \widehat{\Psi} \in C^{1,2}$ to apply Ito Lemma)  have been properly mentioned or refereed to. Further, we synthesize all assumptions used throughout the proofs in Appendix A (see page 14), and include few remarks for viscosity solutions (see Remark 8; page 16).
> > >
> > >
> > > - In the updated Appendix A, we provide improvement and clarification by polishing all proofs. To avoid future confusion regarding which higher-order terms shall be dropped, we first restate Ito formula (provided in Lemma 6; see page 14). All later expansions follow Lemma 6, which essentially keeps the expansion up to first-order, _i.e._ all terms related to $\mathrm{d}t^2$ and $\mathrm{d}t\mathrm{d}\mathrm{W}_t$ are dropped.
> > >
> > >
> > > - We train the models with AdamW (as mentioned in the "Setups" paragraph in Sec 4), and restate Alg 1 2 3 in Sec 3.3 with a more generic form. We thank the reviewer for brining up this confusion.

---

> > > > ### Comment · Reviewer_dQMT · 2021-11-29
> > > > **Response**
> > > >
> > > > I thank the authors for their extended reply and clarifications.  The revised
> > > > manuscript is clearer and gives a much better account of the contribution.  I
> > > > have raised my score accordingly.
> > > >
> > > > I still have a few concerns and I hope that the authors will address them in a
> > > > future version:
> > > > 1) For me the correct justification of the methodology is
> > > > through the IPF theory and not likelihood training.  In particular, it is not
> > > > clear that by parameterizing the two different drifts we indeed recover an ELBO (see
> > > > my comment answer to 2. for more details).
> > > > 2) The authors should have released
> > > > an anonymous repository of the code or at least provided comparison with
> > > > concurrent approaches (even on toy examples).
> > > >
> > > >  I'll answer the rebuttal in detail below (each letter corresponds to
> > > > a paragraph in the given section of the rebuttal).
> > > >
> > > > 1. Comparison to [3,4]
> > > >
> > > > It is not true to say that "by the time of our
> > > > submission, both concurrent works [3,4] were still under peer-review, and
> > > > neither had released code nor provided sufficient information for reproducing".
> > > > [4] was published in MDPI Entropy on August 31st 2021 (see
> > > > https://www.mdpi.com/1099-4300/23/9/1134). It was available on Arxiv on June 3rd
> > > > 2021. The code was released at the same time as the paper at
> > > > https://github.com/franciscovargas/GP_Sinkhorn. [3] was accepted to NeurIPS 2021
> > > > (list of papers was released publicly on November 2nd, admittedly after the ICLR
> > > > 2022 deadline) however the paper was available on Arxiv on June 1st. The code
> > > > was available August 26th at
> > > > https://github.com/JTT94/diffusion_schrodinger_bridge.
> > > >
> > > > 1.1 Training loss
> > > >
> > > > I agree with the authors that the loss function is based on a divergence-based
> > > > objective and do not rely on an estimate of $\nabla \log p_t^{SB}$. I also thank
> > > > the authors for the detailed comparison between the regression loss and the
> > > > divergence loss.
> > > >
> > > > 1.2 Training algorithms \& SDE model
> > > >
> > > > a) b) I agree that the authors do not
> > > > restrict themselves to alternate policies. However, in practice for high
> > > > dimensional datasets, only the alternate policy is used. Doing so the difference
> > > > between [3,4] and the current work relies only on how the drift term in the
> > > > alternate diffusion is computed.
> > > >
> > > > c) This is a matter of taste but I don't find
> > > > the extension to time-varying diffusion to be especially relevant and to be a
> > > > good justification of the proposed method. In particular it is well-known that
> > > > time-reversal can be computed with time-dependent (or even state-dependent)
> > > > volatility matrices, see (Haussmann, Pardoux (1986) for instance). Most of the
> > > > results of [4] (and [3]) can be readily recovered by applying this change.
> > > >
> > > > e) I still think that the algorithm proposed in the current method (admittedly in its
> > > > alternative form) is extremely close to the ones proposed in [3,4]. I agree that
> > > > this is not a "rewriting" in the sense that the training loss is different
> > > > (divergence instead of regression). However, I want to point out that this is
> > > > not what was advertised in the first version of the paper (see the end of
> > > > Section 2 in the first version of the paper). In this respect, I thank the
> > > > authors for their clarifications and their acknowledgements of the tight links
> > > > between the approaches of [3,4] and theirs.
> > > >
> > > > 2. Parameterized lower bound \& Theorem 4
> > > >
> > > > I am not convinced by the explanation
> > > > here. It is true that in SGM parameterizing the score leads to a lower-bound on
> > > > the log-likelihood. However, I don't see how Theorem 4 (I also tried to find
> > > > this information in the Appendix but it does not seem to contain much more details)
> > > > tells us that the parameterized bound is a lower-bound of the log-likelihood. In
> > > > my opinion, the correct current justification of Algorithm 3 is through the IPF
> > > > theory and there is no clear guarantee that the joint optimization procedure
> > > > (Algorithm 2) will recover the Schr\"odinger bridge.  In this sense, I still
> > > > think that there is no real "likelihood training'' of the Schr\"odinger bridge
> > > > in that the parameterized bound is not a proper ELBO.
> > > >
> > > > 3.1 Comparison to [3] under same setups
> > > >
> > > > The experimental results do appear
> > > > better than the ones reported in [3]. I really wish that the authors had
> > > > released an anonymous repository of the code. This would have made clearer the
> > > > differences of implementation between the works of [3] and the present
> > > > contribution.  Also I think the authors could have compared their approach to
> > > > [4] at least in the low-dimensional setting.
> > > >
> > > > Additional comment
> > > >
> > > > I still think it would have been interesting to observe what is the generative
> > > > model obtained after one iteration of the algorithm, i.e. when the backward
> > > > network is updated for the first time. This should recover the generative model
> > > > obtained using SGM and further steps are a refinement of the method. In particular, depending on the number of how large T is this
> > > > might be close to divergence-based SGM.

---

> > > > > ### Author Response · Authors · 2021-11-30
> > > > > **Author Response to Reviewer dQMT (Part 1/2)**
> > > > >
> > > > > We thank the reviewer for the reply. We are pleased that the reviewer acknowledged our extensive clarification, thorough comparison/acknowledgment to [3,4], and the better performance of our approach over [3]. We greatly appreciate the reviewer's willingness to raise the score. Additional clarifications are provided below.
> > > > >
> > > > > ---
> > > > >
> > > > > **1. & 3. Comparison to [3,4]**
> > > > >
> > > > > - Regarding the code, we would like to kindly point out that the _initial commit time_ does NOT necessarily coincide with the _publicly releasing time_. At least for [3] -- which we have been tracking closely --, their repository was not made publicly available until mid October (the exact date is, unfortunately, visible only to their authors). By the time of our submission, we did try to implement their algorithms (following the instructions in their Appendix J.2) yet the results were nowhere close to reportable. As of now, we are still unable to reproduce using their latest release of implementation. As attached in our prior response, this issue had also been observed and raised by others [11].
> > > > >
> > > > >
> > > > > - We admit that an anonymous repository shall provide further clarifications. Unfortunately, given the [recent](https://www.reddit.com/r/MachineLearning/comments/p59pzp/d_imitation_is_the_sincerest_form_of_flattery/?utm_source=share&utm_medium=web2x&context=3) plagiarism alarms in ML & CV conferences, we are hesitated to release our implementation at this moment. To accommodate, however, we can confidently _assure_ that we will release our code upon publication, as we strongly believe in the merit of open sourcing. The updated Alg 1 & 3 in the revision shall also make it easier for anyone to reproduce our result, and we provide the reproducibility statement (as suggested by Reviewer Sm2V) on page 10.
> > > > >
> > > > >
> > > > > - We will include the comparison to [4] on toy datasets in the revision, and we thank the reviewer for the suggestion. Omitting the fact that we are unable to update revision at this point, both methods success in reaching the designed distributions. However, joint optimization with our Alg 1 & 2 admits faster convergence compared to alternate training (as mentioned on page 7).
> > > > >
> > > > >
> > > > > - Regarding 1.2(c), we admit that time-reversal can be applied to more complex diffusions (also mentioned in SGM; see Appendix A in [10]), and we have not argued that [3,4] cannot be extended to these classes. However, given the fact that [3,4] approximate $\nabla \log p_t$ using consecutive time steps $\mathbf{X}(k)$ and $\mathbf{X}(k+1)$, extending [3,4] to time-varying diffusion does amplify the differences (in terms of training loss) between [3,4] and our approach. For instance, increasing $g(t)$ in [3,4] shall readily enlarge the variance of the $\nabla \log p_t$ estimator. Hence, despite that the benefit of monotonically increasing $g(t)$ has been studied thoroughly [8,12] and adopted almost by default in diffusion models [2,8-10], it can be hard (or at least not as straightforward as ours) for [3,4] to adopt similar improvements.
> > > > >
> > > > > ---
> > > > >
> > > > > **2. Parameterized ELBO & Theorem 4**
> > > > >
> > > > > - Given the validity of Eq (3) as a proper ELBO on SGM's log-likelihood (as concurred by the reviewer), we note that one can arrive at Theorem 4 directly from Eq (3) by considering the substitutions: $f=f+g\mathbf{Z}(\cdot,\cdot;\theta)$ and $g\mathbf{s}(\cdot,\cdot;\theta)=\mathbf{Z}(\cdot,\cdot;\theta)+\widehat{\mathbf{Z}}(\cdot,\cdot;\phi)$. In other words, when we _(i)_ lift SGM to accept nonlinear forward drift of the form $f+g\mathbf{Z}$ and _(ii)_ plug in its associated score function, $g\nabla \log p_t^\text{SB}=\mathbf{Z}+\widehat{\mathbf{Z}}$, as suggested nontrivially by the SB theory, the SGM's ELBO coincides _exactly_ with the parametrized lower bound implied by Theorem 4. These substitutions are valid since the derivation of SGM's ELBO (see Theorem 1,3,4,5 in [9]) does _not_ rely on the forms $f$ and $\mathbf{s}(\cdot,\cdot;\theta)$, nor impose assumptions on them. This is also recognized by Reviewer 8tG7.
> > > > >
> > > > >
> > > > > - As such, one may view Theorem 4 as a generalization of SGM's ELBO to the nonlinear SDEs in SB; hence it is a valid ELBO for the log-likelihood of nonlinear SDEs in SB. This interpretation, as concurred by Reviewer 4oR9 and Sm2V, is nontrivial: Prior to our work, SB is often considered as a motivated (as an optimal transport model) yet separate model to SGM, and to what extent do the two models relate to each other remains unclear until we derive Theorem 4. From a more practical viewpoint, the SGM theory in Eq (3), to our best knowledge, has seldom been employed in modern image synthesis tasks, but rather serves as a principle to select hyper-parameters (_e.g._ likelihood weighting in [9]). On the contrary, our finding between SGM and SB allows one to draw inspirations from classical alternate SB algorithm, yielding a computationally efficient training procedure for nonlinear SDEs that may otherwise remain unexplored in standard SGM framework.

---

> > > > > > ### Author Response · Authors · 2021-11-30
> > > > > > **Author Response to Reviewer dQMT (Part 2/2)**
> > > > > >
> > > > > > **Additional comment (SGM + SB refinement)**
> > > > > >
> > > > > > - At the first stage of training, since the forward network $\mathbf{Z}(\cdot,\cdot;\theta) = \mathbf{0}$ is frozen with degenerate outputs, training the backward network $\widehat{\mathbf{Z}}(\cdot,\cdot;\phi)$ with regression/divergence objectives collapses to denoising/sliced score matching [13,14]. It has been reported in [14] that the divergence-based objective admits longer per-iteration runtime yet can achieve better performance, which aligns with our observation in Response **1.1**. On the other hand, the differences between divergence-based SGM (equivalently SSM [14]) and fully SB can be examined in Response **3.2**. Specifically, the last two columns of the Table represent "_SSM training_" and "_SB refinement + SSM training_" when both initialize with the _same_ pre-trained backward network ("SGM/50k"). Having SB refinement during training helps reduce the FID from 13.25 to 11.85.
> > > > > >
> > > > > > ---
> > > > > >
> > > > > > [12] Song & Ermon 2020, "Improved Techniques for SGM"
> > > > > > [13] Hyvarinen 2005, "Estimation of ... by SM"
> > > > > > [14] Song et al 2019, "SSM"

---

> > > > > > ### Public Comment · ~James_Thornton1 · 2022-04-06
> > > > > > **code release**
> > > > > >
> > > > > > I have only just seen this. For the record, I would like to kindly point out that I released the inital DSB code for [3] on August 27th 2021, not October as claimed.
> > > > > >
> > > > > > Thanks,
> > > > > > James

---

### Official Review · Reviewer_Sm2V · 2021-11-03

**Correctness:** 3
**Technical Novelty And Significance:** 4
**Empirical Novelty And Significance:** 3
**Recommendation:** 8
**Confidence:** 4

**Main Review:**

[Novelty and Significance]

The paper presents an elegant framework for likelihood training of Schrödinger bridge (SB)-based generative models. SB generative modeling is a developing area in the field of diffusion generative models and this paper makes a significant contribution to the area. It draws interesting connections to (and generalizes) the framework of previous diffusion-based generative models (e.g., SBGM).

[Writing and Clarity]

The paper is written fairly well, albeit for the expert reader. There exist some clarity issues:

- For Theorem 1, please provide a proof-sketch and/or cite the exact theorem number. In particular, it is unclear to me how to arrive at 7(a) and 7(b) from the Kolmogorov equations in (6). Is this after time-reversal?
- For the proofs, please include a brief description of each step. In the current state, some steps of the proofs are unclear.
    - In the proof of Theorem 3, why do you begin with (1) as the reference measure $\mathbb{P}$ and not 7(a)?
    - In several steps, $dt^2$ and $dtd\mathbf{w}_t$ terms have been dropped. Please mention this in the proof text for clarity.
    - In Eq. (20), I do not understand how you go from $\nabla \cdot(g \hat{z}_t-f)$ to $-\hat{z}_t^{\top}(g \nabla \log p_t^{\mathrm{SB}}) - \nabla \cdot f$.

Corrections:

- Eq. (15) is not an ODE and looks like a typographical mistake — the $g\mathrm{d}\mathbf{w}_t$ term should not be there.
- "... which can be _probability_ expensive on high-dimensional datasets" — *prohibitively*.
- Section A.1: ito —> Ito.
- Section A.1 before Eq. (17). What is $b$? I believe it should be $f$.

[Empirical Evaluation]

The empirical evaluation, although not extensive, is reasonable.

One thing that is unclear from Table 2 are the primary baselines; in my opinion, they should be Multi-stage SB and SGMs. It's good that the authors have reported results from several previous works; however, most of the results are not directly comparable, so it is important to specify what the main baselines are. A particular example is DOT which is not a generative model in itself but a sample improvement technique that operates on a pretrained GAN.

The authors are missing the following work in Table 2 in the optimal transport model class:

Ansari, Abdul Fatir, Ming Liang Ang, and Harold Soh. "Refining deep generative models via discriminator gradient flow." *arXiv preprint arXiv:2012.00780* (ICLR 2021).

On reproducibility: From the paper, it appears that the practical implementation has several moving parts and is not straightforward. The paper does not contain a reproducibility statement and also does not provide enough details for an expert reader to reproduce the results. I encourage the authors to release their code as supplementary material and include further details of the practical implementation in the Appendix for the benefit of the research community.

[Questions]

- Can you elaborate what exactly is meant by "... SGM by enlarging the class of diffusion processes to accept *nonlinear* drifts ..."?
- How is $\nabla\log p^\mathrm{SB}_t$ computed?
- What is the form of the function $f$ in your practical framework/implementation?

[Suggestions]

- It may be better to move the paragraph on "Connection to flow-based models" before "In practice, we parameterize the forward...". The current structure breaks the flow of the reader.
- To make the paper more accessible to readers unfamiliar with Schrödinger bridges, it would be helpful if you provide a brief review in the Appendix.

**Summary Of The Paper:**

This paper presents a framework for likelihood-based training of Schrödinger bridge-based generative models using the theory of Forward-Backward SDEs. In doing so, the paper draws relations to score-based generative models (SBGMs) and shows that the proposed framework is a generalization of the SBGM framework. The proposed framework also provides additional control to the forward SDE to reach the prior distribution unlike the case of SBGM. The authors propose a practical training algorithm which alternates between likelihood training of the forward and backward controls. Experiments on multiple image generation benchmarks demonstrate that the proposed SB-FBSDE performs well as a generative model of the data.

**Summary Of The Review:**

The paper presents a novel framework for likelihood training of Schrödinger bridge (SB)-based generative models. The technical contribution of this paper is significant: it generalizes the SBGM framework and provides a practical algorithm for likelihood training of SB models. Although the paper has some clarity issues, I believe these can be fixed in the revision. Overall, I think this is a good paper and I recommend acceptance.

---

> ### Author Response · Authors · 2021-11-17
> **Author Response to Reviewer Sm2V**
>
> **1. Clarification in the main paper**
>
> - $\nabla \log p_t^\text{SB}$ is typically intractable except for the first training stage. In which case, if we initialize $\mathbf{Z}:= \mathbf{0}$ then first train $\widehat{\mathbf{Z}}$, the term $\nabla \log p_t^\text{SB} = \nabla \log p_t^\text{(1)}$ can be computed just like SGM. However, all afterwards training relies on the divergence-based objectives Eq (16,18,19) as $(\mathbf{Z}, \widehat{\mathbf{Z}})$ no longer output trivial values. We note that the non-divergence expression in Theorem 4 is presented only for the completeness purpose (as it is more intuitive to understand and compare with SGM (3)) rather than deriving training algorithms. In the revision we provide additional clarification below Theorem 4. These divergence objectives are also properly refereed in Alg 1,2,3.
>
>
> - We use degenerate base drift $f := \mathbf{0}$ for all our experiments (see "Setups" paragraph in Sec 4). This implies that in order to achieve reasonable performance, our model must learn the entire data-to-noise diffusion process from scratch. The learned policies/drifts, as shown in the 2D examples, yield highly nonlinear processes. This is in contrast to SGM, which _fixes_ a _linear_ diffusion process throughout training. In this vein, our SB _enlarges_ the SDE model class of SGM by allowing one to learn nonlinear drifts.
>
>
> - We now include the reproducibility statement on page 10. Specifically, Alg 1 is now refined to Alg 1, 2, & 3, where we clarify the training pipeline and practical objectives. The generation process (original Alg 2) is left to Alg 4 in Appendix due to space constraint. As we strongly believe in the merit of open sourcing, we intend to release our implementation upon publication.
>
>
> - We include the missing reference [1] in the Cifar10 comparison table. Following the reviewer's suggestion (also due to space constraint), the table now only include OT and SGM methods. The full table is moved to Appendix B (see Table 2).
>
>
> - The additional stochastic term "$g\mathrm{d}\mathbf{w}_t$" in Corollary 5 was indeed a typo and has been corrected in the revision. We thank the reviewer for the meticulous reading.
>
> ---
>
> **2. Clarification in Appendix**
>
>
> - In Appendix D.1, we provide a brief introduction on Schrodinger Bridge (SB) theory and the proof-sketch of Theorem 1. Specifically, Theorem 1 is mainly adopted from the recent advance in SB theory [2] that extends the classical SB formulation (Eq (49); also used in prior SB works) to accept the SDE class appearing in SGM. Their results (provided in Theorem 11; see page 24) directly connects the Kolmogorov equation (6) to (7a). Then, as conjectured by the reviewer, (7b) can be derived by reversing (7a) then applying the relation $\log p_t^\text{SB}(\cdot) = \log\Psi(t,\cdot) + \log\Psi(t,\cdot)$ from [3].
>
>
> - In the updated Appendix A, we provide improvement and clarification by polishing all proofs. To avoid future confusion regarding which higher-order terms shall be dropped, we first restate Ito formula (provided in Lemma 6; see page 14). All later expansions follow Lemma 6, which essentially keeps the terms up to first-order, _i.e._ all terms related to $\mathrm{d}t^2$ and $\mathrm{d}t\mathrm{d}\mathrm{W}_t$ are dropped. Additionally, the assumptions used throughout the proofs are synthesized in the same section (see page 14).
>
>
> - The proof of Theorem 3 originally follows a standard stochastic optimal control analysis, where the FBSDE system is first derived from the "_uncontrolled_" path measure (_i.e._ $\mathbb{P}$ in our case) and then modified with important sampling (or change of measure) [4]. Yet, as suggested by the reviewer, one can directly base FBSDE system on (7a), which will yield an equivalent (yet more compact) way to conclude the same proof. We have updated the proof accordingly (with the typo being corrected), and we thank the reviewer for pointing this out.
>
>
> - The derivation from "$\nabla \cdot (g\widehat{\mathbf{Z}}_t)$" to "$-\widehat{\mathbf{Z}}_t^\mathsf{T} (g\nabla\log p_t^\text{SB})$" is detailed in Eq (34) on page 17. This is due to integration by part and the limit behavior of $p_t^\text{SB}$, which we adopt from original SGM theory [5].
>
> ---
>
> [1] Ansari, et al 2021, "Refining DGM via D-GF"
> [2] Caluya et al 2021, "Wasserstein ... for SB ... nonlinear drift"
> [3] Chen et al 2021, "Stochastic control liaisons:..."
> [4] Exarchos et al 2018, "SOC via FBSDE"
> [5] Song et al 2021, "MLE of SGM"

---

> > ### Comment · Reviewer_Sm2V · 2021-11-28
> > **Thanks for the clarifications**
> >
> > Thank you to the authors for their answers and clarifications. I appreciate that the authors have clarified the objective and training details in the revision and the readability of the paper has improved significantly. My overall opinion about this paper has not changed; I think this is an interesting piece of work that makes a worthy contribution to the area of SB-based generative models. Finally, I hope that the authors will release their code for the benefit of the research community at large.

---

### Official Review · Reviewer_4oR9 · 2021-11-04

**Correctness:** 3
**Technical Novelty And Significance:** 4
**Empirical Novelty And Significance:** 2
**Recommendation:** 5
**Confidence:** 3

**Main Review:**

This paper introduced a new way to look at the SB problem via a non-linear Feynman-Kac representation of a PDE that defines the SB optimal path measure. This overall framework seems to provide significant new insight on how SB can be solved practically, which is a key merit of the work, and I applaud the authors for the finding.
The strengths of the paper are:
* Proposing an interesting framework to approach the SB problem, via solving the SB optimization problem using a PDE to define a pair of equivalent forward and backward SDEs whose path measures correspond to the SB solution, and then solving this SDE using a Feynman-Kac type of representation.
* Making a connection with prior work, i.e. generalizing the likelihood-based training of score-based generative models.

Novelty aside, I do find the paper requires some major improvement in terms of clarity and a clearer discussion and analysis of limitations and the similarity/difference with prior work. Below are the weaknesses of the paper.
* Clarity needs work: the narrative at times is not very clear, which makes it harder to grasp what is being solved. (See more detailed questions below)
* Differences with some prior work (such as iterative proportional fitting) are not not well explained, even though the proposed likelihood training of the forward SDE and the backward SDE seems very similar.
* Limitations are not properly discussed: training the inference process is done at the cost of an increased cost of computation, which is arguably one of the most important features of SGM. Convergence property of the proposed method not discussed, i.e. whether or not the proposed likelihood training stage will lead to convergence towards the SB solution. A thorough discussion of these limitations and perhaps an experiment on compute cost analysis would be needed.

Some more detailed comments and questions:
* Throughout the introduction, it is not very clear what “computing the log-likelihood objective of Schrodinger Bridge” means. The narrative needs more work. More precisely, it wasn’t very clear to me what the “model” is until I finished reading section 3, and it took me multiple passes. For example theorem 4 refers to the log-likelihood of SB, it’s the log-likelihood of which model exactly? Fixing the inference SDE (i.e. $z_t$) and looking at the marginal likelihood induced by $\hat{z}_t$? Is it a likelihood or a lower bound?  The (bold) $z$ used in the previous theorem seems to not depend on any parametric form (as it solves the SB problem), but here it suddenly becomes parametric.
* End of 2.1, it is not clear why having a more flexible framework will help mitigate the instability problem just mentioned.
* Is the order of the arguments of $h$ in (11) incorrect?
* The presentation of Lemma 2 is a bit confusing, is it correct to say that despite the randomness induced by the Brownian motion used in the ito integral, the solution of y (an SDE) will still be a deterministic and smooth function, as it is after all a solution of the PDE? What’s confusing is that v is a solution of a PDE and y is a solution of an SDE, which is random by nature.
* Presentation of theorem 3 also needs work: is the bold $z$ related to the regular $z$ used in Lemma 2? The notation hasn’t been introduced.
* Last paragraph of page 5: what does the new interpretation of optimal control bring to us? Is there any practical benefit of it?
* Last paragraph of 4.1, I am not sure how meaningful the comparison is with some other OT methods, especially since the SB problem is connected to “entropy-regularized” OT, and is not exactly OT.

Minor points / typos:
* End of page 6, is it a maximization (of likelihood) instead of min?
* The last sentence of page 7: can be founded -> found


**Summary Of The Paper:**

Inspired by stochastic control and forward-backward SDE theory, this paper proposed an iterative algorithm to approach the Schrodinger bridge (SB) problem, generalizing variational likelihood-based training of score-based generative models (SGM). The proposed method is then used to train both the generative process and the inference process of SGMs, recasting the generative modeling problem as the problem of SB. The author then performed experiments on standard image datasets.


**Summary Of The Review:**

The paper introduces a new framework for likelihood-based training of the forward and backward SDEs that are inspired by the Schrodinger bridge problem, which is quite novel. But the paper is not very well written and the practical limitations of the proposed method are not sufficiently addressed. Therefore I do not vote for acceptance given its current form.

---

> ### Author Response · Authors · 2021-11-17
> **Author Response to Reviewer 4oR9 (Part 1/3)**
>
> **1. Clarification on Lemma 2 and Theorem 3**
>
> - In the updated revision, we restate Lemma 2 and Theorem 3 (pages 4-5) with clearer notations, where we capitalize all random variables/vectors to better distinguish SDE, $\mathrm{d}\mathbf{X}_t$=..., and (deterministic) PDE, $\frac{\partial v(t,\boldsymbol{x})}{\partial t}$=..., dynamics.
>
>
> - Regarding Lemma 2, $v(t,\boldsymbol{x})$ is indeed a _deterministic_ temporal-spatial function that obeys the PDE in (10). The stochasticity of $v(\cdot,\cdot)$ appears only when we substitute the spatial argument with the solution to an SDE, _e.g._ $v(t,\mathbf{X}_t)$ where $\mathbf{X}_t$ is from the forward SDE (9a). This shall be understood as applying Ito formula (provided in Appendix A; see Lemma 6): Since $\mathbf{X}_t$ is random by nature, the value of $v(t,\mathbf{X}_t)$ evolving through time is also random (typically denoted as $\mathbf{Y}_t$ in FBSDE theory). The merit of FBSDE theory [1] is then to characterize how $\mathbf{Y}_t = v(t,\mathbf{X}_t)$ evolves as a backward SDE (9b). This expression (9b) holds _almost surely_ along the path generated from the forward SDE.
>
>
> - The relation between $\boldsymbol{z}$ and $\mathbf{Z}_t$ shares a similar spirit. In Lemma 2, $\boldsymbol{z} := G^\mathsf{T}\nabla v$ corresponds to the (deterministic) _spatial derivative_ that contributes to the nonlinear dynamics of a PDE (via the function $h$). Substituting the spatial argument with the random process $\mathbf{X}_t$ makes $\mathbf{Z}_t := G(t,\mathbf{X}_t)\nabla v(t,\mathbf{X}_t) = G(t,\mathbf{X}_t)\nabla \mathbf{Y}_t$ a random vector. The $\mathbf{Z}_t$ appearing in Theorem 3 can be defined similarly, except its $\mathbf{Y}_t$ corresponds to a different PDE and $\mathbf{X}_t$ obeys a different forward SDE.
>
>
> - For the completeness, we note that the original (deterministic) PDE solution can be recovered by taking conditional expectation, _i.e._, $v(t,\boldsymbol{x}) = \mathbb{E}[\mathbf{Y}_t|\mathbf{X}_t=\boldsymbol{x}]$ where the expectation is taken over the forward SDE (9a). This is the key underlying our Theorem 4, where we utilize the fact that $\log p_\text{data}(\boldsymbol{x}) = \mathbb{E}[\mathbf{Y}_0 + \widehat{\mathbf{Y}}_0|\mathbf{X}_0=\boldsymbol{x}]$, and from which we can substitute the results from Theorem 3.
>
>
> - These clarifications are now included in Sec 3.1 (marked blue in page 5). The order of arguments of $h$ is also corrected; we thank the reviewer for the meticulous reading.
>
> ---
>
> **2. Clarification on log-likelihood expression and SB model**
>
>
> - We have updated the log-likelihood expressions of both SGM (Eq (3)) and SB (Theorem 4). Specifically, the value $\mathcal{L}_\text{SGM}(\boldsymbol{x};\theta)$ in Eq (3) serves as a parameterized lower bound of the SGM log-likelihood $\log p_0^\text{SGM}(\boldsymbol{x}) \ge \mathcal{L}_\text{SGM}(\boldsymbol{x};\theta)$, and the equality holds when the score network (parameterized by $\theta$) coincides with the true score function [2]. Meanwhile, Theorem 4 (page 6) now gives the exact log-likelihood of SB model, $\log p_0^\text{SB}(\boldsymbol{x})$, as a function of $\{\mathbf{X}_t,\mathbf{Z}_t,\widehat{\mathbf{Z}}_t\}$, which can be obtained by solving the FBSDE system (13). A similar parameterized lower bound $\log p_0^\text{SB}(\boldsymbol{x}) \ge \mathcal{L}_\text{SB}(\boldsymbol{x};\theta,\phi)$ can be obtained when $\mathbf{Z}_t \approx \mathbf{Z}(\cdot,\cdot; \theta)$ and $\widehat{\mathbf{Z}}_t \approx \widehat{\mathbf{Z}}(\cdot,\cdot; \phi)$ are approximated with some parameterized models (in our case, DNNs). The equality holds when the parameterized models coincide with the values suggested by the FBSDE system. These discussions are added in Sec 3.2 (below Theorem 4, marked blue on page 6).
>
>
> - The clarification above suggests that one shall refer the SB model to $\mathbf{Z}(\cdot,\cdot; \theta)$ and $\widehat{\mathbf{Z}}(\cdot,\cdot; \phi)$, which correspond to the parameterized forward/backward policies. As discussed in Sec 3.2, $\widehat{\mathbf{Z}}_t$ shares a similar role as the score network in SGM, whereas $\mathbf{Z}$ is a distinct _nonlinear_ component originated from SB (hence absent in SGM). Crucially, _both_ are needed to compute the log-likelihood (implied in Theorem 4 & Corollary 5), despite that the sampling process only requires one of which. This is a distinct feature arising from SB model and resembles neither score-based or flow-based models. In the revision, we clarify the relation between "_SB model_" and $(\mathbf{Z}, \widehat{\mathbf{Z}})$ at the end of Sec 3.1 (page 5, marked as blue), after all related technical details have been properly introduced. Theorem 4 in Sec 3.2 is also updated to re-emphasize that. We thank the reviewer for raising these confusions.
>
> ---
>
> [1] Exarchos et al 2018, "SOC via FBSDE"
> [2] Song et al 2021, "MLE of SGM"

---

> > ### Author Response · Authors · 2021-11-17
> > **Author Response to Reviewer 4oR9 (Part 2/3)**
> >
> > **3. Comparison to prior works based on Iterative Proportional Fitting (IPF)**
> >
> > - As conjectured by the reviewer, our alternate training closely relates to prior SB works that rely on IPF [3,4]. However, we first note that the alternate training is one of the _many_ training procedures one can derive from Theorem 4. For instance, the 2D toy datasets are solved with _joint_ optimization (Alg 2), _i.e._ we train both policies, $\mathbf{Z}(\cdot,\cdot; \theta)$ and $\widehat{\mathbf{Z}}(\cdot,\cdot; \phi)$, at the same time. For higher-dimensional (_e.g._ image) datasets where it can be too expensive to keep the entire computational graph, we retreat to the alternate training scheme (Alg 3). In practice, joint optimization can lead to faster convergence at the cost of memory complexity.
> >
> >
> > - In case of alternate training, our method still differs from [3,4] in various aspects. First, both prior methods formulate IPF as _mean-matching regression_. In Appendix D.2, we show that (see Eq (56,57) in page 25) these regression objectives in fact relate to the term $\lVert \mathbf{Z}_t + \widehat{\mathbf{Z}}_t -\nabla \log p_t^\text{SB} \rVert^2$ in Eq (15), where $\nabla \log p_t^\text{SB}$ is numerically approximated _after_ time discretization. Crucially, our method does _NOT_ involve in approximating $\nabla \log p_t^\text{SB}$. In practice, since $\nabla \log p_t^\text{SB}$ is intractable for any nontrivial $(\mathbf{Z}_t, \widehat{\mathbf{Z}}_t)$ (nor do we intend to approximate it), all of our training processes, including joint and alternate training, rely on the divergence-based objectives Eq (16,18,19), which are presented in neither of the prior methods.
> >
> >
> > - Secondly, these methods also differ in their SDE models. Specifically, we adopt the recent advance in SB theory [5] that extends classical SB models (used in [3,4]) to the exact SDE model appearing in SGM. While the SDEs of [3,4] are based on $\mathrm{d}\mathbf{X}_t$=...+$\sqrt{2}\mathrm{d}\mathbf{W}_t$ with _time-invariant_ diffusion coefficient, ours (and SGM) feature _time-varying_ diffusion $g(t)\mathrm{d}\mathbf{W}_t$. Aside from the fact that this $g(t)$ function plays a key role in training stability and generative performance [2,6], it also implies that the connection between [3,4] and SGM can only be made after time discretization by carefully choosing each step size (see Sec C.3 of [3]). In contrast, our theoretical connection is derived readily in continuous-time; hence unaffected by the choice of numerical discretization.
> >
> > - Finally, neither of the prior methods focuses on log-likelihood training, which is the key finding in our work to adopt modern training improvements.
> >
> > - To avoid future confusion, these discussions are now highlighted in both Sec 1 (page 2; marked as blue) and Sec 3.3 (page 6-7). We provide a thorough analysis in Appendix D.2. Meanwhile, we have acknowledged how the alternate training is related and motivated by prior IPF work [3].
> >
> > ---
> >
> > **4. Computational efficiency & limitation**
> >
> > - In practice, our method admits a comparable efficiency to SGM on Cifar10. The table below reports the values measured on the _same_ GPU (TITAN RTX) and network. Notably, our method requires similar training time (+6.8% compared to SGM) yet with a substantially fewer sampling time (-80%). This is made possible with 3 key components. First, we follow prior advice from [3] by caching the sampled trajectories in a reply buffer and refreshing them at a lower frequency (see the updated Alg 2). Secondly, adopting fully nonlinear SDEs greatly reduces the propagation steps (200 steps for ours vs 1000 steps for [2]). Finally, our divergence-based training losses typically converge faster per iteration (see next bullet point) comparing to regression objectives used in SGM.
> >
> >     |  | Total Train Time | Generation Time | Memory (batch=64) |
> >     |---|---|---|---|
> >     | SGM | 99.5 hours | 7 min 15 sec | 6.3 GB |
> >     | SB (ours) | 106.3 hours (+6.8%) | 1 min 27 sec (-80%) | 11.1 GB (+75%) |
> >
> >
> > - To highlight the benefit of our divergence losses, in the below table we report the training results using different losses. Both models are trained from with the _same_ parameter whose initial FID is 35.47. It is clear that despite having longer per-iteration runtime (roughly 2.5 times), our losses lead to much lower FID (with 10k samples) after the same period of time.
> >
> >     | | Total Runtime (second/iterations) | FID after 7 hours |
> >     |---|---|---|
> >     | Regression Loss (SGM) | 1.41 | 33.68 |
> >     | Divergence Loss (ours; Eq (18,19)) | 3.63 | 16.08 |
> >
> >
> > - The primary limitation of our method seems to be the memory, as maintaining 2 distinct networks (like all SB models) and computing the divergence terms can increase the memory by 60-80%. Finally, since Alg 3 is closely related to IPF (see Response **3**) despite with different losses/models, we believe convergence analysis from classical IPF can be applied with few efforts. We leave it as a promising future work.

---

> > > ### Author Response · Authors · 2021-11-17
> > > **Author Response to Reviewer 4oR9 (Part 3/3)**
> > >
> > > **5. Other clarifications**
> > >
> > > - The description at end of Sec 2.1 aimed to emphasize that SB provides a more flexible framework compared to SGM on "_designing the forward diffusion_", rather than mitigating the instability. We have clarified the description in the revision (mark as blue) to avoid confusion, and we thank the reviewer for pointing this out.
> > >
> > > - The description at end of Sec 3.1 aimed to emphasize the role of $\mathbf{Z}_t$ and $\widehat{\mathbf{Z}}_t$ as forward/backward policies so as to motivate why they shall be parameterized and learned (like policy-based methods [7,8]). We have clarified the description in the revision (mark as blue). We hope this shall also re-emphasize that the _SB model_ can be compactly represented by $(\mathbf{Z}_t,\widehat{\mathbf{Z}}_t)$.
> > >
> > > ---
> > >
> > > [3] De Bortoli et al 2021, "Diffusion SB with SGM"
> > > [4] Vargas et al 2021, "Solving SB via MLE"
> > > [5] Caluya et al 2021, "Wasserstein ... for SB ... nonlinear drift"
> > > [6] Song et al 2019, "Generative Modeling ..."
> > > [7] Pereira et al 2019, "DNN ... using nonlinear FK"
> > > [8] Janner et al 2019, "Model-based policy optimization"

---

> > > > ### Comment · Reviewer_4oR9 · 2021-11-30
> > > > **Response to the rebuttal**
> > > >
> > > > I thank the authors for putting in much work in the rebuttal. The clarity of the paper seems to be largely improved, and the additional discussion on limitations and the runtime analysis are very helpful. I am especially surprised that the training time of SB is only 6.8% higher. Is this for the same number of training iterations, or for reaching the same FID or likelihood? In this analysis, is SGM trained by sampling the entire trajectory of sample path like SB, or is the loss estimated via random time step without integration?

---

> > > > > ### Author Response · Authors · 2021-11-30
> > > > > **Author Response to Reviewer 4oR9**
> > > > >
> > > > > We thank the reviewer for the reply. We appreciate that the reviewer acknowledged our extensive clarification and discussion on computational efficiency. Additional clarifications are provided below.
> > > > >
> > > > > ---
> > > > >
> > > > > - "106.3 hours(+6.8%)" is the training time required for our SB model to reach the FID (3.18) reported in Sec 4 on Cifar10. It consists of total 12.5 SB training stages (recall that we refer a complete swipe of backward+forward training to 1 SB stage; see Alg 3). As mentioned on page 20 (or Sec 3.2 in the initial submission), the first 0.5 stage degenerates to SGM training when $\mathbf{Z}(\cdot,\cdot;\theta) = \mathbf{0}$ is frozen with degenerate outputs. This half stage consists of total 160k training iterations. For the later 12 stages, we train both networks alternately using the divergence-based losses in Eq (18,19) with total 18k training iterations. The training results at each SB stage are summarized below.
> > > > >
> > > > >     |  | Accumulated Train Time | Accumulated Train Iterations | FID |
> > > > >     |---|---|---|---|
> > > > >     | @ 0.5 SB stage | 99.5 hours | 160k | 6.02 |
> > > > >     | @ 12.5 SB stage | 106.3 hours | 178k | 3.18 |
> > > > >
> > > > >
> > > > > - Empirically, we observe that SGM training (_i.e._ regression loss) admits faster convergence at the beginning when comparing to the equivalent divergence loss (since the later admits longer per-iteration runtime). However, the FID of SGM training will later saturate and slow down after around 120k iterations (this aligns with prior observation; see Fig 5 & 15 in [9]). In these stages, SB's divergence loss will start to show its superior benefit by providing much richer training signal for _both_ networks. This is evidenced by the second Table in our Response **4**, where we show that starting from a warm-up parameter (FID=35.47), divergence losses are much effective in reducing FID (35.47 -> 16.08) given the same amount of time. Combining these empirical observations, our reported training pipeline best trades-off and leverages the innate connection between SB and SGM (at the first 0.5 SB stage) for training fully nonlinear SDE models.
> > > > >
> > > > >
> > > > > - As for SGM, we simply reuse the results from the first half of the SB stage. Hence, "99.5 hours" is the training time for 160k SGM training iterations to achieve FID of 6.02. Crucially, the SGM training does _NOT_ trained by sampling the entire trajectory as in SB. Rather, it follows standard procedure where at each training iteration we uniformly sample time steps from $[0,T]$ (hence without integration), apply analytic formula to batch data for computing the noise-corrupted states $\mathbf{X}_t$, then perform regression training. Lastly, we note that in order for the pure SGM training to reach the same FID as ours 3.18, one would require much longer training time than the reported 99.5 hours.
> > > > >
> > > > > ---
> > > > >
> > > > > [9] Song & Ermon 2020, "Improved Techniques for SGM"

---

> ### Author Response · Authors · 2021-11-29
> **Follow up before the discussion period closes**
>
> We thank the reviewer for the feedback. We have provided major improvements and clarifications in our updated revision (also concurred by Reviewer Sm2V and dQMT). As we are approaching the end of the discussion period, we would like to ask whether our responses clarify the questions raised in your initial reviews. We are happy to provide any further clarification and discussion.

---

### Official Review · Reviewer_8tG7 · 2021-11-08

**Correctness:** 3
**Technical Novelty And Significance:** 3
**Empirical Novelty And Significance:** 3
**Recommendation:** 8
**Confidence:** 3

**Main Review:**

## Strengths
1. The proposed method is motivated from optimal stochastic control in a principled way. It bridges the gap between $p_T$ and the prior distribution in existing score-based techniques.
2. Compared to other works based on the Schrodinger bridge, the approach described in this paper seems to be more scalable, without the need to hand design part of the transformation as in Wang et al., 2021, or iterative proportional filtering as in De Bortoli et al. 2021.

## Weaknesses
1. The efficiency of the proposed method is unclear in current writing. In Theorem 4, the loss function is described as integrals of expectations. However, it is unclear with respect to which random variable is the expectation taken. How do you estimate the expectation in the loss function? If the expectation is over $x_t$, don't you need to simulate the forward SDE in equation (12a) for each datapoint? Since equation (12a) depends on $z_t$, which is parameterized by a deep neural network, there is no easy way to sample $x_t$ without solving the SDE numerically. By contrast, in score-based generative modeling, $x_t$ is sampled as a noise-perturbed Gaussian with a closed form.

2. There are many errors in writing that affect reading and understanding. For example, in page 3, $p_t^{(2)}$ is never defined. In Lemma 2, I am not sure how to understand the expression $v(t, x) = y(t, x)$. As the solution to an SDE, shouldn't $y$ be a stochastic process? How can a stochastic process equal to $v(t, x)$, which is a deterministic function? What's the definition of $y(t, x)$? Similarly, in Theorem 3, how is $y_t \equiv y(t, x_t)$, what's the definition of $y(t, x_t)$, and how is $y_t$ a function of $x_t$? In Corollary 5, why does an ODE contain the stochastic term $g d w_t$? In Algorithm 1, why are there references to (23a) and (24)? Do you mean other equations?

3. Authors reported better performance compared to prior optimal transport methods like Wang et al., 2021. However, is this because the network architecture used in this paper (NCSN++) is better than those used in previous methods? Will it be more fair to compare with the same network architecture?

4. Isn't the framework proposed in this work exactly the same as score-based generative modeling? If you let the forward SDE be $d x_t = (f + gz_t )dt + gdw_t$, and parameterize the score network as $z_t + \hat{z}_t$, then the reverse SDE is the same as (7b). In this case, you can derive Theorem 4 and Corollary 5 automatically from the theory of score-based generative models.

**Summary Of The Paper:**

Inspired by recent work on score-based generative modeling, this paper proposes to solve Schrodinger bridges for generative modeling. Different from other Schrodinger bridge works, this paper connects the training to maximum likelihood, and provides a way to compute the log-likelihood of the model. The resulting method can shrink the gap between $p_T$ and the prior distribution, and produce high quality image samples and likelihood comparable to score-based generative models.

**Summary Of The Review:**

The paper proposes a useful Schroding bridge framework for generative modeling that is simpler than previous counterparts. However, there are concerns on efficiency issues, writing errors, and fairness in experimental comparison. It is also unclear how much difference is this method compared to the original formulation of score SDEs.

---

> ### Author Response · Authors · 2021-11-17
> **Author Response to Reviewer 8tG7 (Part 1/2)**
>
> **1. Practical training loss & computational efficiency**
>
> - In the updated Theorem 4 and Sec 3.3 (pages 6-7), we clarify our practical training losses in Eq (16,18,19), and detail which random variables their expectations are taken over. These are either denoted in the subscript, _e.g._ $\mathbb{E}_{\mathbf{X}_t\sim(\text{7a,b})}$ in Eq (18,19), or mentioned below the equation, _e.g._ Eq (16), due to space constraint. In practice, we jointly train both networks (Alg 2) using Eq (16) whenever the computational budget is affordable (_e.g._ for toy examples); otherwise we retreat to alternate training (Alg 3) using Eq (18,19).
>
>
> - As conjectured by the reviewer, the expectations in our losses require simulating the parameterized _nonlinear_ SDEs, which, unlike the linear SDE in score-based generative model (SGM), do not admit closed forms. We stress, however, that this is a common practice when dealing with nonlinear SDEs and has been mentioned briefly in SGM for potential nonlinear generalization (see Sec 3.3 and Appendix A in [1]). On the other hand, similar sampling procedures also appear in prior (nonlinear) Schrodinger Bridge (SB) models [2,3], and our work admits no exception.
>
>
> - In practice, our method admits a comparable efficiency to SGM on Cifar10. The table below reports the values measured on the _same_ GPU (TITAN RTX) and network. Notably, our method requires similar training time (+6.8% compared to SGM) yet with a substantially fewer sampling time (-80%). This is made possible with 3 key components. First, we follow prior advice from [2] by caching the sampled trajectories in a reply buffer and refreshing them at a lower frequency (see the updated Alg 2). Secondly, adopting fully nonlinear SDEs greatly reduces the propagation steps (200 steps for ours vs 1000 steps for [1]). Finally, our divergence-based training losses typically converge faster per iteration (see next bullet point) comparing to regression objectives used in SGM. Our primary bottleneck seems to be the memory, as we maintains 2 networks and need to compute the the divergence.
>
> 	|  | Total Train Time | Generation Time | Memory (batch=64) |
> 	|---|---|---|---|
> 	| SGM | 99.5 hours | 7 min 15 sec | 6.3 GB |
> 	| SB (ours) | 106.3 hours (+6.8%) | 1 min 27 sec (-80%) | 11.1 GB (+75%) |
>
>
> - To highlight the benefit of our divergence losses, in the table below we report the training results using different losses. Both models are trained from the _same_ parameter whose initial FID is 35.47. It is clear that despite having longer per-iteration runtime (roughly 2.5 times), our losses lead to much lower FID (with 10k samples) after the same period of time.
>
> 	| | Total Runtime (second/iterations) | FID after 7 hours |
> 	|---|---|---|
> 	| Regression Loss (SGM) | 1.41 | 33.68 |
> 	| Divergence Loss (ours; Eq (18,19)) | 3.63 | 16.08 |
>
> ---
>
> **2. Relation to score-based generative models (SGM)**
>
> - While one can indeed link Theorem 4 and Corollary 5 to the SGM theory, Eq (3), using the transformation ($f=f+g\mathbf{Z}$ and $s=\mathbf{Z}+\widehat{\mathbf{Z}}$) suggested by the reviewer, we stress that this nontrivial interpretation is made possible only _after_ examining our Theorem 4, where we present the exact log-likelihood of SB. Prior to our work, SB is often considered as a motivated (as an optimal transport model) yet separate model to SGM, and to what extent do the two models relate to each other remains unclear until we introduce the rigorous analysis of FBSDE theory in this work. This distinct theoretical contribution is also concurred by Reviewer 4oR9 and Sm2V.
>
>
> - From a more practical viewpoint, the SGM theory in Eq (3), to our best knowledge, has seldom been employed in modern image synthesis tasks, but rather served as a principle to select hyper-parameters (_e.g._ likelihood weighting in [4]). On the contrary, our theoretical finding between SGM and SB allows one to adopt new insights from classical SB algorithm, _e.g._ IPF [2,5] to improve training. This leads to a computationally efficient training procedure (see Response **1.**) for nonlinear SDE models, that may otherwise remain unexplored in standard SGM framework.
>
> ---
>
> [1] Song et al 2020, "SGM through SDE"
> [2] De Bortoli et al 2021, "Diffusion SB with SGM"
> [3] Vargas et al 2021, "Solving SB via MLE"
> [4] Song et al 2021, "MLE of SGM"
> [5] Kullback 1968, "Prob density..."

---

> > ### Author Response · Authors · 2021-11-17
> > **Author Response to Reviewer 8tG7 (Part 2/2)**
> >
> > **3.Clarification on Lemma 2, Theorem 3, and Corollary 5**
> >
> > - In the updated revision, we restate Lemma 2 and Theorem 3 (pages 4-5) with clearer notations, where we capitalize all random variables/vectors to better distinguish SDE, $\mathrm{d}\mathbf{X}_t$=..., and (deterministic) PDE, $\frac{\partial v(t,\boldsymbol{x})}{\partial t}$=..., dynamics.
> >
> >
> > - Regarding Lemma 2, $v(t,\boldsymbol{x})$ is indeed a _deterministic_ temporal-spatial function that obeys the PDE in (10). The stochasticity of $v(\cdot,\cdot)$ appears only when we substitute the spatial argument with the solution to an SDE, _e.g._ $v(t,\mathbf{X}_t)$ where $\mathbf{X}_t$ is from the forward SDE (9a). This shall be understood as applying Ito formula (provided in Appendix A; see Lemma 6): Since $\mathbf{X}_t$ is random by nature, the value of $v(t,\mathbf{X}_t)$ evolving through time is also random (typically denoted as $\mathbf{Y}_t$ in FBSDE theory). The merit of FBSDE theory [6,7] is then to characterize how $\mathbf{Y}_t = v(t,\mathbf{X}_t)$ evolves as a backward SDE (9b). This expression (9b) holds _almost surely_ along the path generated from the forward SDE.
> >
> >
> > - Similarly, in Theorem 3, $\Psi(t,\boldsymbol{x})$ is a deterministic function that obeys the PDE in (6). Substituting the spatial argument with the solution to the forward SDE (13a) introduces stochasticity as $\Psi(t,\mathbf{X}_t)$. In Theorem 3, its logarithmic transform $\log\Psi(t,\mathbf{X}_t)$ is denoted as the random variable $\mathbf{Y}_t$, which readily suggests that $\mathbf{Y}_t = \log\Psi(t,\mathbf{X}_t)$ is a function of $t$ and $\mathbf{X}_t$. For the completeness, we note that the original (deterministic) PDE solution can be recovered by taking conditional expectation; in other words, $\log\Psi(t,\boldsymbol{x}) = \mathbb{E}[\mathbf{Y}_t|\mathbf{X}_t=\boldsymbol{x}]$ where the expectation is taken over the forward SDE (13a). This is essentially the key underlying our Theorem 4, where we utilize the fact that $\log p_\text{data}(\boldsymbol{x}) = \mathbb{E}[\mathbf{Y}_0 + \widehat{\mathbf{Y}}_0|\mathbf{X}_0=\boldsymbol{x}]$, and from which we can substitute the results from Theorem 3.
> >
> >
> > - The additional stochastic term "$g\mathrm{d}\mathbf{w}_t$" in Corollary 5 was a typo and has been corrected in the revision. We thank the reviewer for the meticulous reading.
> >
> > ---
> >
> > **4.Comparison to Wang et al., 2021 [8] using the same network**
> >
> > - In the table below, we report the results using the _same_ network architecture from [8]. In this case, our method achieves **8.13** FID on Cifar10, which is still much lower than their reported **12.32**. As briefly mentioned in Sec 2, [8] relies on a two-stage (_i.e._ two SBs) optimization, where the first SB is solved with logistic regression and the second SB collapses exactly to the regression (_i.e._ denoising score matching) in [1]. On the contrary, our method directly optimizes the entire SB (without breaking it into 2 pieces) while utilizing divergence-based objectives.
> >
> > 	|  | Wang et al., 2021 [8] | Ours |
> > 	|---|---|---|
> > 	| FID | 12.32 | **8.13** |
> >
> > ---
> >
> > **5. Other Clarification**
> >
> > - The expression $p_t^{(*)}$, as mentioned in the notation paragraph (page 2), denotes the marginal density driven by an SDE ($\ast$) at time $t$. In the revision, we re-emphasize this expression when it is first introduced (see page 3, below Eq (2)).
> >
> >
> > - All expectations appearing in the main paper is now properly referred to their associated random variables.
> >
> >
> > - Alg 1, 2, 3 are updated so that all related equations appear in the main paper.
> >
> > ---
> >
> > [6] Exarchos et al 2018, "SOC via FBSDE"
> > [7] Negyesi et al 2021, "One Step Malliavin"

---

> ### Author Response · Authors · 2021-11-29
> **Follow up before the discussion period closes**
>
> We thank the reviewer for the feedback. We have provided major improvements and clarifications in our updated revision (also concurred by Reviewer Sm2V and dQMT). As we are approaching the end of the discussion period, we would like to ask whether our responses clarify the questions raised in your initial reviews. We are happy to provide any further clarification and discussion.

---

> ### Comment · Reviewer_8tG7 · 2021-11-30
> **Feedback after rebuttal**
>
> I would like to thank the authors for their careful response and revision. The new results on training speed and performance with the same model architecture are great. Please make sure to incorporate all of them in the final revision. Writing has been improved after changing the notations. It is still unclear here and there, and needs more polish to have a larger impact. For example, the notation $p^{(1)}_T$ is still not properly introduced before equation (2), and it is unclear what $Z_t$ and $Z(t, X_t)$ are in Lemma 2, and how equation (9b) can determine $Z_t$.
>
> Considering all the improvements, I would like to increase my score and recommend acceptance of this paper.

---

> > ### Author Response · Authors · 2021-11-30
> > **Author Response to Reviewer 8tG7**
> >
> > We thank the reviewer for the reply. We are pleased that the reviewer acknowledged our extensive clarification and additional experiment support. We greatly appreciate the reviewer's willingness to raise the score. Additional clarifications are provided below.
> >
> > ---
> >
> > - $\mathbf{Z}_t=\mathbf{Z}(t,\mathbf{X}_t)$ are meant to be the same variable (similar to Eq (14)). We will unify the notation to $\mathbf{Z}_t$ in the revision, so that it better aligns with the notations used in Lemma 2.
> >
> >
> > - Looking from the FBSDE system (9) alone, there is no specific procedure to decide what $\mathbf{Z}_t$ shall be purely from Eq (9b). Different choices of random process $\mathbf{Z}_t$ will yield different backward solution $\mathbf{Y}_t$; in other words, it can be problem-specific [8]. Hence, if we look from the entire Lemma 2, where the FBSDE (9) has been interpreted as the solution to PDE (10), $\mathbf{Z}_t$ indeed has a specific formula given by the nonlinear Feynman-Kac relation (11). Specifically, $\mathbf{Z}_t$ can be determined from $(\mathbf{X}_t,\mathbf{Y}_t)$ via $\mathbf{Z}_t = G(t,\mathbf{X}_t)\nabla_x\mathbf{Y}_t$. (Note that $\mathbf{Y}_t$ is a time-varying function of $\mathbf{X}_t$ as implied by Eq (11).)
> >
> >
> > - These discussions will be included in the revision (along with the clarification on $p_T^\text{(1)}$). Again, we always thank the reviewer for their precious time on providing the valuable feedback.
> >
> > ---
> >
> > [8] Ma & Yong 2007, "FBSDEs and their Applications"

---

> > > ### Comment · Reviewer_8tG7 · 2021-11-30
> > > **Thanks for the clarification**
> > >
> > > The authors' clarification on how equation (9) determines $Z_t$, and what equation (11) means is very helpful. Incorporating this explanation to the paper can make it much easier to understand.

---

### Author Response · Authors · 2021-11-17
**Author response to all reviewers**

We thank the reviewers for their valuable comments. We are excited that the reviewers identified the novelty of our technical contributions (Reviewer 8tG7, 4oR9, Sm2V, dQMT), appreciated the principled connection with prior fields (Reviewer 4oR9, Sm2V), acknowledged our experimental validations (Reviewer dQMT), and found the paper well-written (Reviewer Sm2V, dQMT). We believe our work takes a significant step toward principled optimization of Schrodinger bridge that is deeply intertwined with score-based models.

---

As _all_ reviewers recognized our technical novelty, the primary critics rather come from the insufficient clarification on presentation and computational efficiency (raised by Reviewer 8tG7 & 4oR9), and the limited comparison and acknowledgment to two concurrent works [1,2] (raised by Reviewer dQMT). We stress that by the time of our submission, both were still under peer-review, and neither had released their codes nor provided sufficient information for reproducing.


In our updated revision, we provide major improvements by clarifying _all_ raised questions. Notable changes in the main paper are enumerated below (marked blue in the revision). Additional clarifications & experiments are included in **Appendix D**.

- The notations are made clear throughout the paper. We capitalize all random variables/vectors so as to better distinguish SDE, $\mathrm{d}\mathbf{X}_t$=..., and (deterministic) PDE, $\frac{\partial v(t,\boldsymbol{x})}{\partial t}$=... dynamics. With these changes, we restate **Lemma 2** and **Theorem 3** (pages 4-5) and provide additional explanation to enhance reading experience.


- The presentation in Sec 3.2 has been rearranged, where we separate out **Sec 3.3 Practical Implementation** (pages 6-7) to clarify our training objectives Eq.(16,18,19), training procedures (Alg. 1,2,3), and discuss the computational efficiency of our method.


- The proofs of our Theorem 3,4 and Corollary 5 are refined in Appendix A. Regularity conditions and assumptions (concerned by Reviewer dQMT) are properly attached to each theory, and we synthesize them on page 14 for the completeness.


- In **Appendix D.2**, we provide thorough analysis in how concurrent works differ from ours in various aspects, including _(i)_ training loss, _(ii)_ SDE model class, _(iii)_ time discretization, and _(iv)_ connection to likelihood training. The algorithmic similarly to IPF has been acknowledged in Sec 1 & 3.3, and we replace **Table 1** with a full paragraph (page 2) for clarification.


- Notable typo ($g\mathrm{d}\mathbf{w}_t$) in Corollary 5  is corrected.


We try our best to resolve all raised concerns in individual response below. We sincerely hope Reviewer 8tG7 and 4oR9 will reconsider the rating and Reviewer dQMT will re-evaluate at an entirety.

---

[1] De Bortoli et al, 2021, "Diffusion SB with SGM"
[2] Vargas et al, 2021, "Solving SB via MLE"

---

> ### Public Comment · ~Neil_D_Lawrence1 · 2022-02-15
> **Code Availability Clarification for Vargas et al**
>
> Congrats on your really nice paper!
>
> Just a quick clarification on code for the papers you reference in this summary.
>
> Francisco was really good about making sure code was available for Vargas et al. It's been up since the time of the original ArXiv submission. In that code you can also find a NN parameterisation of the method allows recreation of De Bortoli et al. up to their dt^2 term (which Vargas et al doesn't have).
>
> You can find the code in the GitHub repo here:  https://github.com/franciscovargas/GP_Sinkhorn
>
> Also, I think Vargas et al was fully peer reviewed and published at the time of ICLR submission (https://www.mdpi.com/1099-4300/23/9/1134)
>
> Congrats again! So cool all this work in this space, and we're all grateful also for your foundational contributions to it!

---

> > ### Public Comment · ~Guan-Horng_Liu1 · 2022-02-15
> > **Thanks for your clarification!**
> >
> > Hi Neil,
> >
> > We very much thank your clarification! In our initial rebuttal **(11/17)**, we indeed missed to recognize the publication on Entropy and the code (we were largely misled by the AAAI format in the arxiv by then). This was pointed out by Reviewer dQMT during [post-rebuttal discussion](https://openreview.net/forum?id=nioAdKCEdXB&noteId=xM6SFYVTDxM) **(11/29)**, and we responded with preliminary comparisons later on the same day. These comparisons were, however, tested on the original algorithm described in Vargas et al (i.e., GP-based method). It'll be interesting to also compare against the latest DNN parametrization of Vargas et al. We'll try to include it later.
> >
> > Thanks again for your clarification (also sharing the code!) and contribution to the field!
> >
> > Bests,
> > Authors

---

> > > ### Public Comment · ~Francisco_Vargas1 · 2022-02-19
> > > **Thank you for the prompt response and preliminary comparisons !**
> > >
> > > To ease future comparisosn we have just added a notebook for the original GP based approach of Vargas-et al. with all the classical toy examples that are typically explored as baselines in these works https://github.com/franciscovargas/GP_Sinkhorn/blob/main/notebooks/2D%20Toy%20Data/2d_examples.ipynb. Our focus was on solving the SBP and not as much generative modelling so we were missing some of these tasks and the classical tunings for them.
> > >
> > > Apologies for the misleading AAAI format we went for the usual publication route of Workshop/Symposium + Journal/Conference. The AAAI format corresponds to the fall symposium on science guided AI, where we also presented the work, we have added a footnote to the arxiv version to clarify this. I agree its confusing, we felt the double collumn format suited the paper the most so we left the arxiv version with the workshop format.
> > >
> > > As for the DNN parametrisation, there is no need to include, since you already compare to DSB (De Bortoli et al.).
> > >
> > > Thank you again for the detailed discussion, responses, and the seminal contributions of your work.

---

> ### Public Comment · ~Francisco_Vargas1 · 2022-02-19
> **Discussion of similarities to the objective in Vargas 2021b**
>
> Dear Authors !
>
> Congratulations on this work it is an excellent contribution ! really looking forward to exploring and applying some of this work!
>
> I had one quick comment. The attempt to trade of regression based objectives with div(f(x,t)) was also initially explored in [Vargas 2021b]  resulting in similar objectives to Eq 15 of thrm 4. This was also released to the public as a technical report in June 2021, and originally submited as a masters thesis (University of Cambridge UK) in June 2020.
>
>  Specifically Equation 6.20 of Prop 1 Section 6.3.1 and Equation 6.24 in prop 2 that follows of [Vargas 2021b]. For quick reference 6.20 and 6.42 (where $f_i^-$ comes from a previous half bridge / prior):
>
> $$f_{i}^{+} = \mathrm{argmin}_{f^{+}} \mathbb{E}^+\left[-2\log p(X_T) +\int_0^T\left(\frac{1}{2}||f^{+} - f_i^{-} ||^2- \nabla \cdot f_i^{-}\right)dt\right] $$
>
> $$f_{i+1}^{-} = \mathrm{argmin}_{f^{-}} \mathbb{E}^-\left[-2\log p(X_0) +\int_0^T\left(\frac{1}{2}||f^{-} - f_i^{+} ||^2 + \nabla \cdot f_i^{+}\right)dt\right]$$
>
> The objective derived there (which is then applied in the IPF algorithm) has similar form to your likelihood training objectives proposed in Eqs 15 in Theorem 4.  The proofs/derivations are a bit similar too you will notice one of the intermediate forms before reaching 6.20 are even more akin to the likelihood training objectives in your work.
>
> Overall seem to be some differences as well as some similarities ! One key difference being is that without your novel Theorem 3 we had no way of efficiently estimating the log likelihood of a data point and had to resort to standard estimators which worked quite poorly for us.  As theorem 3 dictates somewhat the origin of these objectives most of the differences will arise from here.
>
> Another difference I can see is your formulation  allows for discarging the graph of the process to which the expectations are taken with respect to which removes a very large computational overhead. I believe this may also aid with mode collapse some discussion in 6.3.3 [Vargas 2021b].
>
> In short it might be worth a very brief mention/discussion to the objectives in [Vargas, F., 2021b] as you hugely overcome the issues (computation, colapse, data likelihood estimation) of the early divergence based training objectives for SBP that we proposed there.
>
> Again congratulations for this amazing work !
>
> [Vargas, F., 2021b] Vargas, F., June 2021. Machine-learning approaches for the empirical Schrödinger bridge problem (No. UCAM-CL-TR-958). University of Cambridge, Computer Laboratory.  link:  https://www.cl.cam.ac.uk/techreports/UCAM-CL-TR-958.html, pdf: https://www.cl.cam.ac.uk/techreports/UCAM-CL-TR-958.pdf .

---

> > ### Public Comment · ~Guan-Horng_Liu1 · 2022-02-19
> > **Clarification on the alternate/joint objectives (and thanks for the discussion!)**
> >
> > Hi Francisco,
> >
> > We appreciate these valuable discussions and will surely include [Vargas, F., 2021b] in the up-coming revision! Aside from all the detailed comments you've listed down, we would like to add two additional clarifications:
> >
> > - Adopting your notation in 6.20 and 6.24, the objectives of our alternate training (_i.e._, Eq 18 & 19) can be re-phrased as
> >     $${\color{red}f_{i}^{-}} = \mathrm{argmin}_{\color{red}f^{-}} \mathbb{E}^+\left[\int_0^T\left(\frac{1}{2}||f_i^{+} - f^{-} ||^2- \nabla \cdot f^{-}\right)dt\right] \qquad \text{(18)}$$
> >
> >     $${\color{red}f_{i+1}^{+}} = \mathrm{argmin}_{\color{red}f^{+}} \mathbb{E}^-\left[\int_0^T\left(\frac{1}{2}||f_i^{-} - f^{+} ||^2 + \nabla \cdot f^{+}\right)dt\right] \qquad \text{(19)}$$
> >    which, crucially, come from optimizing the same KL objectives (as appeared in many IPF-based SBs, _e.g._, [De Bortoli et al., 2021]) yet w.r.t. **_different_** variables, _e.g._, $f^+$ in your 6.20 _vs._ $f^-$ in our Eq 18. The optimizing variables in Eq 18 & 19 were not made artificially but rather arises naturally from MLE perspective (Theorem 3 & 4). It also leads to a few computational benefits, as you mentioned, in that _(i)_ the computational graph can be discarded and _(ii)_ we need not have to know the boundary densities $p_0$ and $p_T$; hence making the framework suitable for complex data, _e.g._, image, generation where $p_0 \equiv p_\text{data}$ is often unknown. This was also mentioned in [De Bortoli et al., 2021].
> >
> > - Our joint training (Alg 2) relies on the log-likelihood computation in Theorem 3, and, as you stated, was not captured in [Vargas, F., 2021b]. This joint objective generalizes ODE-based generative models to SDE (setting $g := 0$ in Theorem 4 will yield the same formula for computing log-likelihood in CNF); hence referred to as 'diffusion flow' in the revision.
> >
> > Thanks again for your discussion and attaching this very nice technical report!

---

> > > ### Public Comment · ~Francisco_Vargas1 · 2022-02-19
> > > **Great discussion and very helpful clariifcations , thanks again !**
> > >
> > > Open review should have an upvote for comments ! the above was very helpful. The MLE perspective is very helpful and insightful. Going from $argmin_{Q \in \mathcal{D}(\pi_1)} D_{KL}(dQ |||dP_i)$ to instead  $argmin_Q D_{KL}(dP^{\cdot|1}_i \frac{d\pi_1}{dp_i} \big|\big| dQ )$ (e.g. Obs 1+Theorem 1 Vargas 2021a) is of a similar flavour and also helped us a lot with the GP formulation allowing to detach the graph and such.
> > >
> > > In a way the connection of your alternation algorithm to IPF can also be made explicit via re-expressing  $argmin_Q D_{KL}(dP^{\cdot|1}_i \frac{d\pi_1}{dp_i} \big|\big| dQ )$   in terms of divergences similar to how 6.20 and 6.24 were derived  (i.e.applying your theorem 4 or our Props 1,2 directly after Girsanov+disintegration could potentially work I think). It does highlights the two (Eq 6.20, 6.24 vs 18,19) are the same if the argmins are switched  from +->- and viseversa (which is made very clear in the way you wrote them above).  This might be useful, although from the responses to reviewers it sounds like you might have another strategy to describe the equivalence to IPF for the camera ready version.
> > >
> > > Thanks again for the very helpful discussion ! its very reasuring and promsing to see these connections and similarities !
> > >
> > > [Vargas 2021a] Vargas, F., Thodoroff, P., Lamacraft, A. and Lawrence, N., 2021. Solving schrödinger bridges via maximum likelihood. Entropy, 23(9), p.1134.
> > >
> > > [Vargas 2021b] Vargas, F., June 2021. Machine-learning approaches for the empirical Schrödinger bridge problem (No. UCAM-CL-TR-958). University of Cambridge, Computer Laboratory.

---

### Decision · Program_Chairs · 2022-01-20

**Decision:**

Accept (Poster)

**Comment:**

The paper presents a new computational framework, grounded on Forward-Backward SDEs theory, for the log-likelihood training of Schrödinger Bridge and provides theoretical connections to score-based generative models. The presentation of the results is not satisfactory (the algorithm should be clarified in several places and the notation is not accurate which raises doubts about the soundness of the method). The paper is thus very hard to read for the non-experts on the subject. Furthermore, some reviewers raise concerns about the similarity of this method to other algorithms that were never cited in the paper. Finally, the empirical analysis, as of now, is limited.

In the rebuttal the authors carefully addressed lots of the comments. However paper's presentation still needs to be substantially improved (de-densification of the paper would be extremely important since now the main narrative is very convoluted). The authors made several changes in the manuscript, but detailed discussion regarding training time complexity still seems to be missing (main body and the Appendix) in the new version of the manuscript, even though this was one of the main raised concerns. Overall, the manuscript requires major rewriting. Since the comments regarding the content were successfully addressed (the reviewers are satisfied with detailed answers given by the authors), the paper satisfies the conference bar and can be accepted.